# Learning multivariate Gaussians with imperfect advice

**Arnab Bhattacharyya** [* 1]  **Davin Choo** [* 2]  **Philips George John** [* 3]  **Themis Gouleakis** [* 4]

## Abstract

We revisit the problem of distribution learning within the framework of learning-augmented algorithms. In this setting, we explore the scenario where a probability distribution is provided as potentially inaccurate advice on the true, unknown distribution. Our objective is to develop learning algorithms whose sample complexity decreases as the quality of the advice improves, thereby surpassing standard learning lower bounds when the advice is sufficiently accurate. Specifically, we demonstrate that this outcome is achievable for the problem of learning a multivariate Gaussian distribution $N(\boldsymbol{\mu}, \boldsymbol{\Sigma})$ in the PAC learning setting. Classically, in the advice-free setting, $\widetilde{\Theta}(d^2/\varepsilon^2)$ samples are sufficient and worst case necessary to learn $d$-dimensional Gaussians up to TV distance $\varepsilon$ with constant probability. When we are additionally given a parameter $\widetilde{\boldsymbol{\Sigma}}$ as advice, we show that $\widetilde{\mathcal{O}}(d^{2-\beta}/\varepsilon^2)$ samples suffice whenever $\|\widetilde{\boldsymbol{\Sigma}}^{-1/2}\boldsymbol{\Sigma}\widetilde{\boldsymbol{\Sigma}}^{-1/2} - \boldsymbol{I_d}\|_1 \leq \varepsilon d^{1-\beta}$ (where $\|\cdot\|_1$ denotes the entrywise $\ell_1$ norm) for any $\beta > 0$, yielding a polynomial improvement over the advice-free setting.

## 1. Introduction

The problem of approximating an underlying distribution from its observed samples is a fundamental scientific problem. The *distribution learning* problem has been studied for more than a century in statistics, and it is the underlying engine for much of applied machine learning. The emphasis in modern applications is on high-dimensional distributions, with the goal being to understand when one can escape the curse of dimensionality. The survey by (Diakonikolas, 2016) gives an excellent overview of classical and modern techniques for distribution learning, especially when there is some underlying structure to be exploited.

In this work, we investigate how to go beyond worst case sample complexities for learning distributions by considering situations where one is also given the aid of possibly imperfect advice regarding the input distribution. We position our study in the context of *algorithms with predictions*, where the usual problem input is supplemented by "predictions" or "advice" (potentially drawn from modern machine learning models). The algorithm's goal is to incorporate the advice in a way that improves performance if the advice is of high quality, but if the advice is inaccurate, there should not be degradation below the performance in the no-advice setting. Most previous works in this setting are in the context of online algorithms, e.g. for the ski-rental problem (Gollapudi & Panigrahi, 2019; Wang et al., 2020; Angelopoulos et al., 2020), non-clairvoyant scheduling (Purohit et al., 2018), scheduling (Lattanzi et al., 2020; Bamas et al., 2020a; Antoniadis et al., 2022), augmenting classical data structures with predictions (e.g. indexing (Kraska et al., 2018) and Bloom filters (Mitzenmacher, 2018)), online selection and matching problems (Antoniadis et al., 2020; Dütting et al., 2021; Choo et al., 2024), online TSP (Bernardini et al., 2022; Gouleakis et al., 2023), and a more general framework of online primal-dual algorithms (Bamas et al., 2020b). However, there have been some recent applications to other areas, e.g. graph algorithms (Chen et al., 2022; Dinitz et al., 2021), causal learning (Choo et al., 2023), and mechanism design (Gkatzelis et al., 2022; Agrawal et al., 2022).

We apply the algorithms with predictions perspective to the classical problem of learning high-dimensional Gaussian distributions. For a $d$-dimensional Gaussian $N(\boldsymbol{\mu}, \boldsymbol{\Sigma})$, it is known (e.g. see Appendix C of (Ashtiani et al., 2020)) that
(1) When $\boldsymbol{\Sigma} = \mathbf{I}_d$, $\Theta(d/\varepsilon^2)$ i.i.d. samples suffice to learn a $\widehat{\boldsymbol{\mu}} \in \mathbb{R}^d$ such that $\mathrm{d_{TV}}(N(\boldsymbol{\mu}, \mathbf{I}_d), N(\widehat{\boldsymbol{\mu}}, \mathbf{I}_d)) \leq \varepsilon$.
(2) In general, $\widetilde{\Theta}(d^2/\varepsilon^2)$ i.i.d. samples suffice to learn $\widehat{\boldsymbol{\mu}}$ and $\widehat{\boldsymbol{\Sigma}}$ such that $\mathrm{d_{TV}}(N(\boldsymbol{\mu}, \boldsymbol{\Sigma}), N(\widehat{\boldsymbol{\mu}}, \widehat{\boldsymbol{\Sigma}})) \leq \varepsilon$.
Here, $\mathrm{d_{TV}}$ denotes the *total variation distance*, and the algorithm for both cases is the most natural one: compute the empirical mean and empirical covariance. Meanwhile, note

---

[*]Equal contribution. Part of work done while the authors were affiliated with the National University of Singapore, Singapore. [1]University of Warwick, United Kingdom [2]Harvard University, United State of America [3]CNRS-CREATE & National University of Singapore, Singapore [4]Nanyang Technological University, Singapore. Correspondence to: Arnab Bhattacharyya <arnab.bhattacharyya@warwick.ac.uk>, Davin Choo <davinchoo@seas.harvard.edu>, Philips George John <philips.george.john@u.nus.edu>, Themis Gouleakis <themis.gouleakis@ntu.edu.sg>.

*Proceedings of the $42^{nd}$ International Conference on Machine Learning*, Vancouver, Canada. PMLR 267, 2025. Copyright 2025 by the author(s).

that if one is given as advice the correct mean $\widetilde{\boldsymbol{\mu}} = \boldsymbol{\mu}$, then using distribution testing, one can certify that $\|\widetilde{\boldsymbol{\mu}} - \boldsymbol{\mu}\|_2 \leq \varepsilon$ using only $\widetilde{\Theta}(\sqrt{d}/\varepsilon^2)$ samples, quadratically better than without advice; see (Diakonikolas et al., 2023) and Appendix C of (Diakonikolas et al., 2017). This observation motivates the object of our study.

**GAUSSIAN LEARNING WITH ADVICE**: Given samples from a Gaussian $N(\boldsymbol{\mu}, \boldsymbol{\Sigma})$, as well as advice $\widetilde{\boldsymbol{\mu}}$ and $\widetilde{\boldsymbol{\Sigma}}$, how many samples are required to recover $\widehat{\boldsymbol{\mu}}$ and $\widehat{\boldsymbol{\Sigma}}$ such that $\mathrm{d_{TV}}(N(\boldsymbol{\mu}, \boldsymbol{\Sigma}), N(\widehat{\boldsymbol{\mu}}, \widehat{\boldsymbol{\Sigma}}) \leq \varepsilon$ with probability at least $1 - \delta$? The sample complexity should be a function of the dimension, $\varepsilon, \delta$, as well as a measure of how close $\widetilde{\boldsymbol{\mu}}$ and $\widetilde{\boldsymbol{\Sigma}}$ are to $\boldsymbol{\mu}$ and $\boldsymbol{\Sigma}$ respectively.

**Notation.** We use *lowercase letters* for scalars, set elements, random variable instantiations, *uppercase letters* for random variables, *bolded lowercase letters* for vectors and sets, *bolded uppercase letters* for set of random variables and matrices, *calligraphic letters* for probability distributions and sets of sets, and *small caps* for algorithm names. Intuitively, we use non-bolded versions for singletons, bolded versions for collections of items, and calligraphic for more complicated objects. The context should be clear enough to distinguish between various representations.

## 1.1. Our main results

We give the first known results in distribution learning[1] with imperfect advice. Our techniques are piecewise elementary and easy to follow. Furthermore, we provide polynomial time algorithms for producing the estimates $\widehat{\boldsymbol{\mu}}$ and $\widehat{\boldsymbol{\Sigma}}$ based on LASSO and SDP formulations.

Given a mean $\widetilde{\boldsymbol{\mu}} \in \mathbb{R}^d$ and covariance matrix $\widetilde{\boldsymbol{\Sigma}} \in \mathbb{R}^{d \times d}$ as advice, we present two algorithms TESTANDOPTIMIZE-MEAN and TESTANDOPTIMIZECOVARIANCE that provably improve on the sample complexities of $\widetilde{\Theta}(d/\varepsilon^2)$ and $\widetilde{\Theta}(d^2/\varepsilon^2)$ for identity and general covariances respectively when given high quality advice.

**Theorem 1.1.** *For any given $\varepsilon, \delta \in (0, 1)$, $\eta \in [0, \frac{1}{4}]$, and $\widetilde{\boldsymbol{\mu}} \in \mathbb{R}^d$, the TESTANDOPTIMIZEMEAN algorithm uses $n \in \widetilde{\mathcal{O}}\left(\frac{d}{\varepsilon^2} \cdot (d^{-\eta} + \min\{1, f(\boldsymbol{\mu}, \widetilde{\boldsymbol{\mu}}, d, \eta, \varepsilon)\})\right)$ where*

$$f(\boldsymbol{\mu}, \widetilde{\boldsymbol{\mu}}, d, \eta, \varepsilon) = \frac{\|\boldsymbol{\mu} - \widetilde{\boldsymbol{\mu}}\|_1^2}{d^{1-4\eta}\varepsilon^2}$$

*i.i.d. samples from $N(\boldsymbol{\mu}, \mathbf{I}_d)$ for some unknown mean $\boldsymbol{\mu}$ and identity covariance $\mathbf{I}_d$, and can produce $\widehat{\boldsymbol{\mu}}$ in $\mathrm{poly}(n, d)$ time such that $\mathrm{d_{TV}}(N(\boldsymbol{\mu}, \mathbf{I}_d), N(\widehat{\boldsymbol{\mu}}, \mathbf{I}_d)) \leq \varepsilon$ with success probability at least $1 - \delta$.*

[1]There is a recent concurrent work on discrete distribution *testing* with imperfect advice (Aliakbarpour et al., 2024).

**Theorem 1.2.** *For any given $\varepsilon, \delta \in (0, 1)$, $\eta \in [0, 1]$ and $\widetilde{\boldsymbol{\Sigma}} \in \mathbb{R}^{d \times d}$, TESTANDOPTIMIZECOVARIANCE uses $n \in \widetilde{\mathcal{O}}\left(\frac{d^2}{\varepsilon^2} \cdot \left(d^{-\eta} + \min\left\{1, f(\boldsymbol{\Sigma}, \widetilde{\boldsymbol{\Sigma}}, d, \eta, \varepsilon)\right\}\right)\right)$ where*

$$f(\boldsymbol{\Sigma}, \widetilde{\boldsymbol{\Sigma}}, d, \eta, \varepsilon) = \frac{\|\mathrm{vec}(\widetilde{\boldsymbol{\Sigma}}^{-1/2}\boldsymbol{\Sigma}\widetilde{\boldsymbol{\Sigma}}^{-1/2} - \mathbf{I}_d)\|_1^2}{d^{2-\eta}\varepsilon^2}$$

*i.i.d. samples from $N(\boldsymbol{\mu}, \boldsymbol{\Sigma})$ for some unknown mean $\boldsymbol{\mu}$ and unknown covariance $\boldsymbol{\Sigma}$, and can produce $\widehat{\boldsymbol{\mu}}$ and $\widehat{\boldsymbol{\Sigma}}$ in $\mathrm{poly}(n, d, \log(1/\varepsilon))$ time such that $\mathrm{d_{TV}}(N(\boldsymbol{\mu}, \boldsymbol{\Sigma}), N(\widehat{\boldsymbol{\mu}}, \widehat{\boldsymbol{\Sigma}})) \leq \varepsilon$ with success probability at least $1 - \delta$.*

In particular, TESTANDOPTIMIZEMEAN uses only $\widetilde{\mathcal{O}}(\frac{d^{1-\eta}}{\varepsilon^2})$ samples when $\|\boldsymbol{\mu} - \widetilde{\boldsymbol{\mu}}\|_1 < \varepsilon d^{(1-5\eta)/2} = \varepsilon\sqrt{d} \cdot d^{-5\eta/2}$, for any $\eta \in [0, \frac{1}{4}]$. Similarly, TESTANDOP-TIMIZECOVARIANCE uses only $\widetilde{\mathcal{O}}(\frac{d^{2-\eta}}{\varepsilon^2})$ samples when $\|\mathrm{vec}(\widetilde{\boldsymbol{\Sigma}}^{-1/2}\boldsymbol{\Sigma}\widetilde{\boldsymbol{\Sigma}}^{-1/2} - \mathbf{I}_d)\|_1 < \varepsilon d^{1-\eta} = \varepsilon d \cdot d^{-\eta}$, for any $\eta \in [0, 1]$. Both algorithms have polynomial runtime.

The choice of representing the quality of the advice in terms of the $\ell_1$-norm is well-motivated. It is known, e.g. see Theorem 2.5 of (Foucart & Rauhut, 2013), that if a vector $\boldsymbol{x}$ satisfies $\|\boldsymbol{x}\|_1 \leq \tau$, then for any positive integer $s$, $\sigma_s(\boldsymbol{x}) \leq \tau/(2\sqrt{s})$, where $\sigma_s(\boldsymbol{x})$ is the $\ell_2$-error of the best $s$-sparse approximation to $\boldsymbol{x}$. Thus, if $\|\widetilde{\boldsymbol{\mu}} - \boldsymbol{\mu}\|_1 \leq 2\varepsilon d^{(1-\eta)/2}$, then $\sigma_{d^{1-\eta}}(\widetilde{\boldsymbol{\mu}} - \boldsymbol{\mu}) \leq \varepsilon$. The latter may be very reasonable, as one may have good predictions for most of the coordinates of the mean with the error in the advice concentrated on a sublinear ($d^{1-\eta}$) number of coordinates. Algorithmically, we employ sublinear property testing algorithms to evaluate the quality of the given advice before deciding how to produce a final estimate, similar in spirit to the TE-STANDMATCH approach in (Choo et al., 2024). The idea of incorporating property testing as a way to verify whether certain distributional assumptions are satisfied that enable efficient subsequent learning has also been explored in recent works on testable learning (Rubinfeld & Vasilyan, 2023; Klivans et al., 2024; Vasilyan, 2024).

We supplement the above with information-theoretic lower bounds. Here, we say that an algorithm $(\varepsilon, 1 - \delta)$-PAC learns a distribution $\mathcal{P}$ if it can produce another distribution $\widehat{\mathcal{P}}$ such that $\mathrm{d_{TV}}(\mathcal{P}, \widehat{\mathcal{P}}) \leq \varepsilon$ with success probability at least $1 - \delta$. Our lower bounds tell us that $\widetilde{\Omega}(d/\varepsilon^2)$ and $\widetilde{\Omega}(d^2/\varepsilon^2)$ samples are unavoidable for PAC-learning $N(\boldsymbol{\mu}, \mathbf{I}_d)$ and $N(\boldsymbol{\mu}, \boldsymbol{\Sigma})$ respectively when given low quality advice.

**Theorem 1.3.** *Suppose we are given $\widetilde{\boldsymbol{\mu}} \in \mathbb{R}^d$ as advice with only the guarantee that $\|\boldsymbol{\mu} - \widetilde{\boldsymbol{\mu}}\|_1 \leq \Delta$. Then, any algorithm that $(\varepsilon, \frac{2}{3})$-PAC learns $N(\boldsymbol{\mu}, \mathbf{I}_d)$ requires $\Omega\left(\frac{\min\{d, \Delta^2/\varepsilon^2\}}{\varepsilon^2 \log(1/\varepsilon)}\right)$ samples in the worst case.*

**Theorem 1.4.** *Suppose we are given a symmetric and positive-definite $\widetilde{\boldsymbol{\Sigma}} \in \mathbb{R}^{d \times d}$ as advice with only the guarantee that $\|\mathrm{vec}\left(\widetilde{\boldsymbol{\Sigma}}^{-\frac{1}{2}}\boldsymbol{\Sigma}\widetilde{\boldsymbol{\Sigma}}^{-\frac{1}{2}} - \mathbf{I}_d\right)\|_1 \leq \Delta$. Then,*

*any algorithm that $(\varepsilon, \frac{2}{3})$-PAC learns $N(\mathbf{0}, \boldsymbol{\Sigma})$ requires $\Omega\left(\frac{\min\{d^2, \Delta^2/\varepsilon^2\}}{\varepsilon^2 \log(1/\varepsilon)}\right)$ samples in the worst case.*

Both of our lower bounds are tight in the following sense. Our algorithm TESTANDOPTIMIZEMEAN gives a polynomially-smaller sample complexity compared to $\widetilde{\mathcal{O}}(d/\varepsilon^2)$ when the advice quality (measured in terms of the $\ell_1$-norm) is polynomially smaller compared to $\varepsilon\sqrt{d}$. Theorem 1.3 shows that this is the best we can do; there is a hard instance where the advice quality is $\leq \varepsilon\sqrt{d}$ and we need $\widetilde{\Omega}(d/\varepsilon^2)$ samples. A similar situation happens between TESTANDOPTIMIZECOVARIANCE and Theorem 1.4, when the guarantee on the advice quality is $\leq \varepsilon d$.

Note that the lower bounds in Theorems 1.3 and 1.4 apply even when the parameter $\Delta$ is known to the algorithm, while our algorithms are stronger since they do not need to know $\Delta$ beforehand. In case $\Delta$ is known, the sample complexity of the distribution learning component of our algorithms match the above lower bounds up to log factors.

## 1.2. Technical overview

To obtain our upper bounds, we first show that the existing test statistics for non-tolerant testing can actually be used for tolerant testing with the same asymptotic sample complexity bounds and then use these new tolerant testers to test the advice quality. The tolerance is with respect to the $\ell_2$-norm for mean testing and with respect to the Frobenius norm for covariance testing. These results are folklore, but since they may be of independent interest, we present their proofs in Appendix B.1 for completeness.

**Lemma 1.5** (Tolerant mean tester). *Given $\varepsilon_2 > \varepsilon_1 > 0$, $\delta \in (0,1)$, and $d \geq \left(\frac{16\varepsilon_2^2}{\varepsilon_2^2 - \varepsilon_1^2}\right)^2$, there is a tolerant tester that uses $\mathcal{O}\left(\frac{\sqrt{d}}{\varepsilon_2^2 - \varepsilon_1^2} \log\left(\frac{1}{\delta}\right)\right)$ i.i.d. samples from $N(\boldsymbol{\mu}, \mathbf{I}_d)$ and satisfies both conditions below:*
*1. If $\|\boldsymbol{\mu}\|_2 \leq \varepsilon_1$, then the tester outputs Accept,*
*2. If $\|\boldsymbol{\mu}\|_2 \geq \varepsilon_2$, then the tester outputs Reject,*
*each with success probability at least $1 - \delta$.*

**Lemma 1.6** (Tolerant covariance tester). *Given $\varepsilon_2 > \varepsilon_1 > 0$, $\delta \in (0,1)$, and $d \geq \varepsilon_2^2$, there is a tolerant tester that uses $\mathcal{O}\left(d \cdot \max\left\{\frac{1}{\varepsilon_1^2}, \left(\frac{\varepsilon_2^2}{\varepsilon_2^2 - \varepsilon_1^2}\right)^2, \left(\frac{\varepsilon_2}{\varepsilon_2^2 - \varepsilon_1^2}\right)^2\right\} \log\left(\frac{1}{\delta}\right)\right)$ i.i.d. samples from $N(\mathbf{0}, \boldsymbol{\Sigma})$ and satisfies both conditions below:*
*1. If $\|\boldsymbol{\Sigma} - \mathbf{I}_d\|_F \leq \varepsilon_1$, then the tester outputs Accept,*
*2. If $\|\boldsymbol{\Sigma} - \mathbf{I}_d\|_F \geq \varepsilon_2$, then the tester outputs Reject,*
*each with success probability at least $1 - \delta$.*

We will first explain how to obtain our result for TESTANDOPTIMIZEMEAN before explaining how a similar approach works for TESTANDOPTIMIZECOVARIANCE.

### 1.2.1. APPROACH FOR TESTANDOPTIMIZEMEAN

Without loss of generality, we may assume henceforth that $\widetilde{\boldsymbol{\mu}} = \mathbf{0}$ since one can always pre-process samples by subtracting $\widetilde{\boldsymbol{\mu}}$ and then add $\widetilde{\boldsymbol{\mu}}$ back to the estimated $\widehat{\boldsymbol{\mu}}$. Our overall approach is quite natural: (i) use the tolerant testing algorithm in Lemma 1.5 to get an upper bound on the "advice quality", and (ii) enforce the constraint on the "advice quality" when learning $\widehat{\boldsymbol{\mu}}$.

The most immediate notion of advice quality one may posit is $\|\boldsymbol{\mu} - \mathbf{0}\|_2 = \|\boldsymbol{\mu}\|_2$. Let us see what issues arise. Using an exponential search process, we can invoke Lemma 1.5 directly to find some $r > 0$, such that $r/2 \leq \|\boldsymbol{\mu} - \widehat{\boldsymbol{\mu}}\|_2 = \|\boldsymbol{\mu}\|_2 \leq r$. To argue about the sample complexity for learning $\widehat{\boldsymbol{\mu}}$, and ignoring computational efficiency, one can invoke the Scheffé tournament approach for density estimation. Let $\mathcal{N}$ be an $\varepsilon$-cover in $\ell_2$ of the the $\ell_2$-ball of radius $r$ around $\mathbf{0}$. Clearly, $\boldsymbol{\mu}$ is $\varepsilon$-close in $\ell_2$ to one of the points in $\mathcal{N}$. It is known (e.g. see Chapter 4 of (Devroye & Lugosi, 2001)) that the sample complexity of the Scheffé tournament algorithm scales as $\log|\mathcal{N}|$. However, we have that $\log|\mathcal{N}| = \Omega(d)$; e.g. see Proposition 4.2.13 of (Vershynin, 2018). Indeed, one can get a formal lower bound showing that the sample complexity cannot be made sublinear in $d$ for non-trivial values of $r$. To get around this barrier, we will instead take the notion of advice quality to be $\|\boldsymbol{\mu}\|_1$ instead of $\|\boldsymbol{\mu}\|_2$. It is known that $d^{\frac{cr^2}{\varepsilon^2}}$ $\ell_2$ balls of radius $\varepsilon$ suffice to cover an $\ell_1$-ball of radius $r$, for some absolute constant $c > 0$; e.g. see Chapter 4, Example 2.8 of (Vershynin, 2012). Using this modified approach, the Scheffé tournament only requires $\mathcal{O}(\frac{r^2}{\varepsilon^4} \log d)$ samples which could be $o(d/\varepsilon^2)$ for a wide range of values of $r$.

There are still two issues to address: (i) how to obtain an $\ell_1$ estimate $r$ of $\boldsymbol{\mu}$, i.e., $r/2 \leq \|\boldsymbol{\mu}\|_1 \leq r$, and (ii) how to get a computationally efficient learning algorithm.

To address (i), we can apply the standard inequality $\|\boldsymbol{\mu}\|_2 \leq \|\boldsymbol{\mu}\|_1 \leq \sqrt{d}\|\boldsymbol{\mu}\|_2$ bound to transform our $\ell_2$ estimate from Lemma 1.5 into an $\ell_1$ one. However, since the number of samples has a quadratic relation with $r$, we need a better approximation than $\sqrt{d}$ to achieve sample complexity that is sublinear in $d$. To achieve this, we partition the $\boldsymbol{\mu}$ vector into blocks of size at most $k \leq d$ and approximate the $\ell_1$ norm of each smaller dimension vector separately and then add them up to obtain an $\ell_1$ estimate of the overall $\boldsymbol{\mu}$. Doing so improves the resulting multiplicative error to $\approx \sqrt{d/k}$ instead of $\sqrt{d}$. In effect, we devise a tolerant tester for a mixed $\ell_{1,2}$ norm instead of the $\ell_1$ or $\ell_2$ norms directly.

To address (ii), observe that the Scheffé tournament approach requires time at least linear in the size of the $\varepsilon$-cover. In order to do better, we observe that we can formulate our task as an optimization problem with an $\ell_1$-constraint. Specifically, given samples $\mathbf{y}_1, \ldots, \mathbf{y}_n$, we solve the follow-

ing program: $\widehat{\boldsymbol{\mu}} = \operatorname{argmin}_{\|\boldsymbol{\beta}\|_1 \leq r} \frac{1}{n} \sum_{i=1}^{n} \|\mathbf{y}_i - \boldsymbol{\beta}\|_2^2$. The error $\|\boldsymbol{\mu} - \widehat{\boldsymbol{\mu}}\|_2$ can be analyzed by similar techniques as those used for analyzing $\ell_1$-regularization in the context of LASSO or compressive sensing; e.g. see (Tibshirani, 1996; 1997; Hastie et al., 2015).

### 1.2.2. APPROACH FOR TESTANDOPTIMIZECOVARIANCE

As before, we may assume without loss of generality that $\widetilde{\boldsymbol{\Sigma}} = \mathbf{I}_d$ by pre-processing the samples appropriately. Furthermore, we can invest $\Omega(d/\varepsilon^2)$ samples up-front to ensure that the empirical mean $\widehat{\boldsymbol{\mu}}$ will be an $\varepsilon$-good estimate of $\boldsymbol{\mu}$. Then, it will suffice to obtain an estimate $\widehat{\boldsymbol{\Sigma}}$ of $\boldsymbol{\Sigma}$ such that $\|\boldsymbol{\Sigma}^{-1}\widehat{\boldsymbol{\Sigma}} - \mathbf{I}_d\|_F \leq \mathcal{O}(\varepsilon)$ suffices. Furthermore, we may assume that we get i.i.d. samples from $N(\mathbf{0}, \boldsymbol{\Sigma})$ and also that $\boldsymbol{\Sigma}$ is full rank. These are without loss of generality for the following two reasons. Firstly, instead of a single sample from $N(\boldsymbol{\mu}, \boldsymbol{\Sigma})$, we will draw two samples $\boldsymbol{x}_1, \boldsymbol{x}_2 \sim N(\boldsymbol{\mu}, \boldsymbol{\Sigma})$ and consider $\boldsymbol{x}' = \frac{\boldsymbol{x}_1 - \boldsymbol{x}_2}{\sqrt{2}}$, which is distributed according to $N(\mathbf{0}, \boldsymbol{\Sigma})$. Secondly, it is known that the empirical covariance constructed from $d$ i.i.d. samples of $N(\mathbf{0}, \boldsymbol{\Sigma})$ will have the same rank as $\boldsymbol{\Sigma}$ itself, with probability at least $1 - \delta$; see see Lemma A.13. So, we can simply project and solve the problem on the full rank subspace of the empirical covariance matrix.

At a high level, the approach for TESTANDOPTIMIZECOVARIANCE is the same as TESTANDOPTIMIZEMEAN after three adjustments to adapt from vectors to matrices.

The first adjustment is that we perform a suitable preconditioning process using an additional $\mathcal{O}(d)$ samples so that we can subsequently argue that $\|\boldsymbol{\Sigma}^{-1}\|_2 \leq 1$. This will then allow us to argue that $\|\boldsymbol{\Sigma}^{-1}\widehat{\boldsymbol{\Sigma}} - \mathbf{I}_d\|_F \leq \|\boldsymbol{\Sigma}^{-1}\|_2 \|\widehat{\boldsymbol{\Sigma}} - \boldsymbol{\Sigma}\|_F \in \mathcal{O}(\varepsilon)$. Our preconditioning technique is inspired by (Kamath et al., 2019); while they use $\mathcal{O}(d)$ samples to construct a preconditioner to control the maximum eigenvalue, we use a similar approach to control the minimum eigenvalue.

In more detail, our technique is as follows: we will compute a preconditioning matrix $\mathbf{A}$ using $d$ i.i.d. samples such that $\mathbf{A}\boldsymbol{\Sigma}\mathbf{A}$ has eigenvalues at least 1, i.e. $\lambda_{\min}(\mathbf{A}\boldsymbol{\Sigma}\mathbf{A}) \geq 1$. That is, $\|(\mathbf{A}\boldsymbol{\Sigma}\mathbf{A})^{-1}\|_2 = \frac{1}{\lambda_{\min}(\mathbf{A}\boldsymbol{\Sigma}\mathbf{A})} \leq 1$. Then, we solve the problem treating $\mathbf{A}\boldsymbol{\Sigma}\mathbf{A}$ as our new $\boldsymbol{\Sigma}$. This adjustment succeeds with probability at least $1 - \delta$ for any given $\delta \in (0, 1)$ and is possible because, with probability 1, the empirical covariance $\widehat{\boldsymbol{\Sigma}}$ formed by using $d$ i.i.d. samples would have the same eigenspace as $\boldsymbol{\Sigma}$, and so we would have a bound on the ratios between the minimum eigenvalues between $\widehat{\boldsymbol{\Sigma}}$ and $\boldsymbol{\Sigma}$; see Lemma A.13.

**Lemma 1.7.** *For any $\delta \in (0, 1)$, there is an explicit preconditioning process that uses $d$ i.i.d. samples from $N(\mathbf{0}, \boldsymbol{\Sigma})$ and succeeds with probability at least $1 - \delta$ in constructing a matrix $\mathbf{A} \in \mathbb{R}^{d \times d}$ such that $\lambda_{\min}(\mathbf{A}\boldsymbol{\Sigma}\mathbf{A}) \geq 1$. Furthermore, for any full rank PSD matrix $\widetilde{\boldsymbol{\Sigma}} \in$*

$\mathbb{R}^{d \times d}$*, we have* $\|(\mathbf{A}\widetilde{\boldsymbol{\Sigma}}\mathbf{A})^{-1/2}\mathbf{A}\boldsymbol{\Sigma}\mathbf{A}(\mathbf{A}\widetilde{\boldsymbol{\Sigma}}\mathbf{A})^{-1/2} - \mathbf{I}_d\| = \|\widetilde{\boldsymbol{\Sigma}}^{-1/2}\boldsymbol{\Sigma}\widetilde{\boldsymbol{\Sigma}}^{-1/2} - \mathbf{I}_d\|$.

The matrix $\mathbf{A}$ in Lemma 1.7 is essentially constructed by combining the eigenspace corresponding to "large eigenvalues" with a suitably upscaled eigenspace corresponding to "small eigenvalues" in the empirical covariance matrix obtained by $d$ i.i.d. samples.

The second adjustment pertains to the partitioning idea used for multiplicatively approximating $\|\operatorname{vec}(\boldsymbol{\Sigma} - \mathbf{I}_d)\|_1$. Observe that the covariance matrix of a marginal of a multivariate Gaussian is precisely the principal submatrix of the original covariance $\boldsymbol{\Sigma}$ on the corresponding projected coordinates. For example, if one focuses on coordinates $\{i, j\} \subseteq [d]$ of each sample, then the corresponding covariance matrix is $\begin{bmatrix} \boldsymbol{\Sigma}_{i,i} & \boldsymbol{\Sigma}_{i,j} \\ \boldsymbol{\Sigma}_{j,i} & \boldsymbol{\Sigma}_{j,j} \end{bmatrix}$, for $i < j$. To this end, we generalize the partitioning scheme described for TESTANDOPTIMIZEMEAN to higher ordered objects.

**Definition 1.8** (Partitioning scheme). Fix $q \geq 1$, $d \geq 1$, and a $q$-ordered $d$-dimensional tensor $\mathcal{T} \in \mathbb{R}^{d^{\otimes q}}$. Let $\mathbf{B} \subseteq [d]$ be a subset of indices and define $\mathcal{T}_{\mathbf{B}}$ as the principal subtensor of $\mathcal{T}$ indexed by $\mathbf{B}$. A collection of subsets $\mathbf{B}_1, \ldots, \mathbf{B}_w \subseteq [d]$ is called an $(q, d, k, a, b)$-partitioning of the tensor $\mathcal{T}$ if the following three properties hold:
1. $|\mathbf{B}_1| \leq k, \ldots, |\mathbf{B}_w| \leq k$
2. For every cell of $\mathcal{T}$ appears in *at least* $a$ of the $w$ principal subtensors $\mathcal{T}_{\mathbf{B}_1}, \ldots, \mathcal{T}_{\mathbf{B}_w}$.
3. For every cell of $\mathcal{T}$ appears in *at most* $b$ of the $w$ principal subtensors $\mathcal{T}_{\mathbf{B}_1}, \ldots, \mathcal{T}_{\mathbf{B}_w}$.

For example, when $q = 2$, $\mathbf{T} \in \mathbb{R}^{d \times d}$ is just a $d \times d$ matrix. Observe one can always obtain a partitioning with $k \leq d^q$ by letting the index sets $\mathbf{B}_1, \ldots, \mathbf{B}_w$ encode every possible index, but this results in a large $w = \binom{d}{q}$ which can be undesirable for downstream analysis. The partitioning used in TESTANDOPTIMIZEMEAN is a special case of Definition 1.8 with $q = a = b = 1$, $k = \lceil d/w \rceil$. For TESTANDOPTIMIZECOVARIANCE, we are interested in the case where $q = 2$ and $a = 1$. Ideally, we want to minimize $k$ and $b$ as well. Figure 1 illustrates an example of a $(q = 2, d = 5, k = 3, a = 1, b = 3)$-partitioning.

While an existence result suffices, we show that a probabilistic construction will in fact succeed with high probability.

**Lemma 1.9.** *Fix dimension $d \geq 2$ and group size $k \leq d$. Consider the $q = 2$ setting where $\mathbf{T} \in \mathbb{R}^{d \times d}$ is a matrix. Define $w = \frac{10d(d-1)\log d}{k(k-1)}$. Pick sets $\mathbf{B}_1, \ldots, \mathbf{B}_w$ each of size $k$ uniformly at random (with replacement) from all the possible $\binom{d}{k}$ sets. With high probability in $d$, this is a $(q = 2, d, k, a = 1, b = \frac{30(d-1)\log d}{(k-1)})$-partitioning scheme.*

The key idea behind utilizing partitioning schemes is that the marginal over a subset of indices $\mathbf{B} \subseteq [d]$ of a $d$-dimensional

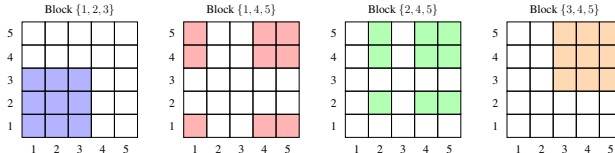

Figure 1: Consider partitioning a $d \times d$ matrix (i.e., $d = 5$ and $q = 2$) with $w = 4$ blocks $\{(1,2,3), (1,4,5), (2,4,5), (3,4,5)\}$, each of size $k = 3$. We see that every cell in the original $5 \times 5$ matrix appears in at least $a = 1$ and at most $b = 3$ times across all the induced submatrices.

Gaussian with covariance matrix $\boldsymbol{\Sigma}$ has covariance matrix that is the principal submatrix $\boldsymbol{\Sigma}_{\mathbf{B}}$ of $\boldsymbol{\Sigma}$. So, if we can obtain a multiplicative $\alpha$-approximation of a collection of principal submatrices $\boldsymbol{\Sigma}_{\mathbf{B}_1}, \ldots \boldsymbol{\Sigma}_{\mathbf{B}_w}$ such that all cells of $\boldsymbol{\Sigma}$ are present, then we can obtain a multiplicative $\alpha$-approximation of $\boldsymbol{\Sigma}$ just like in Section 2. Meanwhile, the $b$ parameter allows us to upper bound the overestimation factor due to repeated occurrences of any cell of $\boldsymbol{\Sigma}$.

Finally, the third and last adjustment is to the optimization program for learning $\widehat{\boldsymbol{\Sigma}}$. Given samples $\mathbf{y}_1, \ldots, \mathbf{y}_n$ from $N(\boldsymbol{\mu}, \boldsymbol{\Sigma})$, we define:

$$\widehat{\boldsymbol{\Sigma}} = \underset{\substack{\mathbf{A} \in \mathbb{R}^{d \times d} \text{ is p.s.d.} \\ \|\text{vec}(\mathbf{A}-\mathbf{I}_d)\|_1 \le r \\ \|\mathbf{A}^{-1}\|_2 \le 1}}{\operatorname{argmin}} \sum_{i=1}^{n} \|\mathbf{A} - \mathbf{y}_i \mathbf{y}_i^\top\|_F^2$$

Observe that $\boldsymbol{\Sigma}$ is a feasible solution to the above program. The optimization problem can be solved efficiently since it can be written as an SDP with convex constraints; see Appendix D.3. We finally bound $\|\boldsymbol{\Sigma} - \widehat{\boldsymbol{\Sigma}}\|_F$ using an analysis that mirrors that for TESTANDOPTIMIZEMEAN but is in terms of matrix algebra. We provide the pseudocode and analysis of the TESTANDOPTIMIZECOVARIANCE algorithm in Appendix D.

### 1.2.3. LOWER BOUND

To prove Theorem 1.3 and Theorem 1.4, we make use of a lemma in (Ashtiani et al., 2020) that informally says the following: If we can construct a cover $f_1, \ldots, f_M$ of distributions such that the pairwise KL divergence is at most $\kappa$ and the pairwise TV distance is $> 2\varepsilon$, then, given sample access to an unknown $f_i$, the sample complexity of learning a distribution which is $\varepsilon$-close to $f_i$ in total variation with probability $\ge \frac{2}{3}$ over the samples (which is referred to as $(\varepsilon, \frac{2}{3})$-PAC learning in total variation) is $\ge \widetilde{\Omega}\left(\frac{\log M}{\kappa}\right)$. This lemma gives an information-theoretic lower bound and is a consequence of the generalized Fano's inequality.

To apply this lemma in the context of learning with advice, we need to fix an advice $\boldsymbol{a}$ (mean or covariance, in the case of our problem) and find a large cover of distribu-

tions $f_1, \ldots, f_M$ that satisfy the conditions of the lemma (pairwise KL $\le \kappa$ and pairwise TV $> 2\varepsilon$), while also satisfying a guarantee on the advice quality with respect to *all* $f_1, \ldots, f_M$ (say, the quality of $\boldsymbol{a}$ is $Q$). Then, applying the lemma will show a sample complexity lower bound for learning a distribution given advice with quality $Q$, since an adversary can choose an $f_i$ in the cover set and give $\boldsymbol{a}$ (fixed) as the advice in each case while still satisfying the advice quality requirement. Note that the advice $\boldsymbol{a}$ is immaterial here as the underlying ground truth is one of $f_1, \ldots, f_M$. The lemma asserts that we still need $\widetilde{\Omega}\left(\frac{\log M}{\kappa}\right)$ samples to learn a distribution close to the given $f_i$ (where the pairwise TV separation of $> 2\varepsilon$ is crucial in ensuring that the learning algorithm would need to identify the correct $f_i$ to succeed, since no distribution $f$ will be $\varepsilon$-close in TV to $f_i$ and $f_j$ for $i \ne j$ due to the triangle inequality).

In the context of learning a Gaussian with unknown mean, the advice quality that we consider is $\|\widetilde{\boldsymbol{\mu}} - \boldsymbol{\mu}\|_1$, where $\widetilde{\boldsymbol{\mu}}$ is the advice and $\boldsymbol{\mu}$ is the ground truth. To show Theorem 1.3, we construct a cover of $M$ distributions $N(\boldsymbol{\mu}_i, \mathbf{I}_d)$ such that $\|\widetilde{\boldsymbol{\mu}} - \boldsymbol{\mu}_i\|_1$ is precisely the same for all $\boldsymbol{\mu}_i$'s. Then, we ensure that the pairwise TV and KL requirements are satisfied by controlling the $\ell_2$ distance $\|\boldsymbol{\mu}_i - \boldsymbol{\mu}_j\|_2$ for each pair $i \ne j$. This enables us to use a construction where we set the first $k$ coordinates of each $\boldsymbol{\mu}_i$ based on the codewords of an error correcting code with distance $\ge \Omega(k)$, and we can show the existence of such a code with $2^{\Omega(k)}$ codewords using the Gilbert-Varshamov bound.

In the context of learning Gaussians with unknown covariance, we consider the advice quality $\|\widehat{\boldsymbol{\Sigma}}^{-\frac{1}{2}} \boldsymbol{\Sigma} \widehat{\boldsymbol{\Sigma}}^{-\frac{1}{2}} - \mathbf{I}_d\|_1$ where $\boldsymbol{\Sigma}$ is the ground truth and $\widetilde{\boldsymbol{\Sigma}}$ is the advice. To prove a lower bound on the sample complexity of learning given good advice, we follow a similar strategy where again, we want to construct a cover of $M$ distributions $N(\mathbf{0}, \boldsymbol{\Sigma}_i)$ which all satisfy a bound on the advice quality and also satisfy the pairwise TV and KL requirements. (Ashtiani et al., 2020) also pursue the same goal but without the advice quality constraint. We adapt their construction by defining a family of block-diagonal orthogonal matrices such that the size of the submatrices can be used to control the entrywise $\ell_1$-norm distance to the identity. Quantifying the KL divergences and TV distances between the constructed gaussians then gives the desired lower bound.

**Remainder of the paper.** Due to space constraints, our main paper focuses on presenting results for the identity covariance setting and defer details for the general covariance setting to the appendix; see also Appendix A for a review on Gaussian distributions. TESTANDOPTIMIZEMEAN is presented in Section 2 and the hardness result Lemma 3.2 is given in Section 3. Some experimental results illustrating the savings in sample complexity are shown in Section 4 before we conclude with some open directions in Section 5.

## 2. Identity covariance setting

We begin by defining a parameterized sample count $m(d, \varepsilon, \delta)$. Then, we will describe APPROXL1 and show how to use it according to the strategy in Section 1.2.1.

**Definition 2.1.** For any $d \geq 1$, $\varepsilon > 0$, and $\delta \in (0, 1)$, we define $m(d, \varepsilon, \delta) = n_{d,\varepsilon} \cdot r_\delta = \lceil \frac{16\sqrt{d}}{\varepsilon^2} \rceil \cdot (1 + \lceil \log \left( \frac{12}{\delta} \right) \rceil)$.

Given samples from a $d$-dimensional isotropic Gaussian $N(\boldsymbol{\mu}, \mathbf{I}_d)$ with unknown mean $\boldsymbol{\mu}$ and identity covariance, the APPROXL1 algorithm partitions the $d$ coordinates into $w = \lceil d/k \rceil$ buckets each of length at most $k \in [d]$ and separately perform an exponential search to find the 2-approximation of the $\ell_2$ norm of each bucket by repeatedly invoking the tolerant tester from Lemma 1.5. In the terminology of Definition 1.8, this is a partitioning scheme with $q = 1$, $a = 1$, and $b = 1$. Crucially, projecting the samples in $\mathbb{R}^d$ of $N(\boldsymbol{\mu}, \mathbf{I}_d)$ into the subcoordinates of $\mathbf{B} \subseteq [d]$ yields samples in $\mathbb{R}^{|\mathbf{B}|}$ from $N(\boldsymbol{\mu_B}, \mathbf{I}_{|\mathbf{B}|})$ so we can obtain valid estimates using each of these marginals. After obtaining the $\ell_2$ estimate of each bucket, we use Fact A.1 to obtain bounds on the $\ell_1$ and then combine them by summing up these estimates: if we have an $\varepsilon$-multiplicative approximation of each bucket's $\ell_1$, then their sum will be an $\mathcal{O}(\varepsilon)$-multiplicative approximation of the entire $\boldsymbol{\mu}$ vector whenever the partition overlap parameters $a$ and $b$ of Definition 1.8 are constants.

In Appendix C.1, we give the pseudocode of the APPROXL1 algorithm and prove that it has the following guarantees.

**Lemma 2.2.** *Let $k$, $\alpha$, and $\zeta$ be the input parameters to the APPROXL1 algorithm (Algorithm 4). Given $m(k, \alpha, \delta')$ i.i.d. samples from $N(\boldsymbol{\mu}, \mathbf{I}_d)$, APPROXL1 succeeds with probability at least $1 - \delta$ and has the following properties:*
*1. If APPROXL1 outputs Fail, then $\|\boldsymbol{\mu}\|_2 > \zeta/2$.*
*2. If APPROXL1 outputs $\lambda \in \mathbb{R}$, then $\|\boldsymbol{\mu}\|_1 \leq \lambda \leq 2\sqrt{k} \cdot (\lceil d/k \rceil \cdot \alpha + 2\|\boldsymbol{\mu}\|_1)$.*

Now, suppose APPROXL1 tells us that $\|\boldsymbol{\mu}\|_1 \leq r$. We can then perform a constrained LASSO to search for a candidate $\widehat{\boldsymbol{\mu}} \in \mathbb{R}^d$ using $\mathcal{O}(\frac{r^2}{\varepsilon^4} \log \frac{d}{\delta})$ samples from $N(\boldsymbol{\mu}, \mathbf{I}_d)$.

**Lemma 2.3.** *Fix $d \geq 1$, $r \geq 0$, and $\varepsilon, \delta > 0$. Given $\mathcal{O}(\frac{r^2}{\varepsilon^4} \log \frac{d}{\delta})$ samples from $N(\boldsymbol{\mu}, \mathbf{I}_d)$ for some unknown $\boldsymbol{\mu} \in \mathbb{R}^d$ with $\|\boldsymbol{\mu}\|_1 \leq r$, one can produce an estimate $\widehat{\boldsymbol{\mu}} \in \mathbb{R}^d$ in $\mathrm{poly}(n, d)$ time such that $\mathrm{d}_{\mathrm{TV}}(N(\boldsymbol{\mu}, \mathbf{I}_d), N(\widehat{\boldsymbol{\mu}}, \mathbf{I}_d)) \leq \varepsilon$ with success probability at least $1 - \delta$.*

*Proof.* Suppose we get $n$ samples $\mathbf{y}_1, \ldots, \mathbf{y}_n \sim N(\boldsymbol{\mu}, \mathbf{I}_d)$. For $i \in [n]$, we can re-express each $\mathbf{y}_i$ as $\mathbf{y}_i = \boldsymbol{\mu} + \mathbf{g}_i$ for some $\mathbf{g}_i \sim N(\mathbf{0}, \mathbf{I}_d)$. Let us define $\widehat{\boldsymbol{\mu}} \in \mathbb{R}^d$ as follows:

$$\widehat{\boldsymbol{\mu}} = \underset{\|\boldsymbol{\beta}\|_1 \leq r}{\mathrm{argmin}} \frac{1}{n} \sum_{i=1}^{n} \|\mathbf{y}_i - \boldsymbol{\beta}\|_2^2 \qquad (1)$$

By optimality of $\widehat{\boldsymbol{\mu}}$ in Equation (1), we have

$$\frac{1}{n} \sum_{i=1}^{n} \|\mathbf{y}_i - \widehat{\boldsymbol{\mu}}\|_2^2 \leq \frac{1}{n} \sum_{i=1}^{n} \|\mathbf{y}_i - \boldsymbol{\mu}\|_2^2 \qquad (2)$$

By expanding and rearranging Equation (2), one can show (see Appendix C.2)

$$\|\widehat{\boldsymbol{\mu}} - \boldsymbol{\mu}\|_2^2 \leq \frac{2}{n} \langle \sum_{i=1}^{n} \mathbf{g}_i, \widehat{\boldsymbol{\mu}} - \boldsymbol{\mu} \rangle \qquad (3)$$

Meanwhile, it is known (see Lemma A.15) that $\Pr(\| \sum_{i=1}^{n} \mathbf{g}_i \|_\infty \geq \sqrt{2n \log \left( \frac{2d}{\delta} \right)}) \leq \delta$. Therefore, using Hölder's inequality and triangle inequality with the above, we see that, with probability at least $1 - \delta$,

$$\|\widehat{\boldsymbol{\mu}} - \boldsymbol{\mu}\|_2^2 \leq \frac{2}{n} \langle \sum_{i=1}^{n} \mathbf{g}_i, \widehat{\boldsymbol{\mu}} - \boldsymbol{\mu} \rangle \leq \frac{2}{n} \cdot \left\| \sum_{i=1}^{n} \mathbf{g}_i \right\|_\infty \cdot \|\widehat{\boldsymbol{\mu}} - \boldsymbol{\mu}\|_1$$

$$\leq \frac{2}{n} \cdot \left\| \sum_{i=1}^{n} \mathbf{g}_i \right\|_\infty \cdot (\|\widehat{\boldsymbol{\mu}}\|_1 + \|\boldsymbol{\mu}\|_1) \leq 4r \cdot \sqrt{\frac{2 \log \left( \frac{2d}{\delta} \right)}{n}}$$

When $n = \frac{2r^2 \log \frac{2d}{\delta}}{\varepsilon^4} \in \mathcal{O}\left( \frac{r^2}{\varepsilon^4} \log \frac{d}{\delta} \right)$, we have $\|\widehat{\boldsymbol{\mu}} - \boldsymbol{\mu}\|_2^2 \leq 4r \cdot \sqrt{\frac{2 \log \left( \frac{2d}{\delta} \right)}{n}} = 4\varepsilon^2$. So, by Pinsker's inequality (see Theorem A.10) and KL divergence of Gaussians (see Lemma A.8), we see that $\mathrm{d}_{\mathrm{TV}}(N(\boldsymbol{\mu}, \mathbf{I}_d), N(\widehat{\boldsymbol{\mu}}, \mathbf{I}_d)) \leq \sqrt{\frac{1}{2}\mathrm{d}_{\mathrm{KL}}(N(\boldsymbol{\mu}, \mathbf{I}_d), N(\widehat{\boldsymbol{\mu}}, \mathbf{I}_d))} \leq \sqrt{\frac{1}{4}\|\boldsymbol{\mu} - \widehat{\boldsymbol{\mu}}\|_2^2} \leq \varepsilon$. Finally, it is known that LASSO runs in $\mathrm{poly}(n, d)$ time. $\square$

Using Lemma 2.3, we now ready to prove Theorem 1.1.

---

**Algorithm 1** The TESTANDOPTIMIZEMEAN algorithm.

1: **Input**: Error rate $\varepsilon > 0$, failure rate $\delta \in (0, 1)$, parameter $\eta \in [0, \frac{1}{4}]$, and sample access to $N(\boldsymbol{\mu}, \mathbf{I}_d)$
2: **Output**: $\widehat{\boldsymbol{\mu}} \in \mathbb{R}^d$
3: Define $k = \lceil d^{4\eta} \rceil$, $\alpha = \varepsilon \cdot d^{-(1-3\eta)/2}$, $\zeta = 4\varepsilon \cdot \sqrt{d}$, and $\delta' = \frac{\delta}{\lceil d/k \rceil \cdot \lceil \log_2 \zeta/\alpha \rceil}$     ▷ Note: $\zeta > 2\alpha$
4: Draw $m(k, \alpha, \delta')$ i.i.d. samples from $N(\boldsymbol{\mu}, \mathbf{I}_d)$ and store it into a set $\mathcal{S}$     ▷ See Definition 2.1
5: Let Outcome be the output of the APPROXL1 algorithm given $k$, $\alpha$, $\zeta$, and $\mathbf{S}$ as inputs
6: **if** Outcome is $\lambda \in \mathbb{R}$ and $\lambda < \varepsilon\sqrt{d}$ **then**
7:    Draw $n \in \widetilde{\mathcal{O}}(\lambda^2/\varepsilon^4)$ i.i.d. samples $\mathbf{y}_1, \ldots, \mathbf{y}_n \in \mathbb{R}^d$
8:    **return** $\widehat{\boldsymbol{\mu}} = \mathrm{argmin}_{\|\boldsymbol{\beta}\|_1 \leq \lambda} \frac{1}{n} \sum_{i=1}^{n} \|\mathbf{y}_i - \boldsymbol{\beta}\|_2^2$
9: **else**
10:    Draw $n \in \widetilde{\mathcal{O}}(d/\varepsilon^2)$ i.i.d. samples $\mathbf{y}_1, \ldots, \mathbf{y}_n \in \mathbb{R}^d$
11:    **return** $\widehat{\boldsymbol{\mu}} = \frac{1}{n} \sum_{i=1}^{n} \mathbf{y}_i$     ▷ Empirical mean
12: **end if**

*Proof of Theorem 1.1.* Without loss of generality, we may assume that $\widetilde{\boldsymbol{\mu}} = \mathbf{0}$. This is because we can pre-process all samples by subtracting $\widetilde{\boldsymbol{\mu}}$ to yield i.i.d. samples from $N(\boldsymbol{\mu}', \mathbf{I}_d)$ where $\boldsymbol{\mu}' = \boldsymbol{\mu} - \widetilde{\boldsymbol{\mu}}$. Suppose we solved this problem to produce $\widehat{\boldsymbol{\mu}}'$ where $\mathrm{d}_{\mathrm{TV}}(N(\boldsymbol{\mu}', \mathbf{I}_d), N(\widehat{\boldsymbol{\mu}}', \mathbf{I}_d)) \leq 10\varepsilon$, we can then output $\widehat{\boldsymbol{\mu}} = \widehat{\boldsymbol{\mu}}' + \widetilde{\boldsymbol{\mu}}$ and see from data processing inequality that $\mathrm{d}_{\mathrm{TV}}(N(\boldsymbol{\mu}, \mathbf{I}_d), N(\widehat{\boldsymbol{\mu}}, \mathbf{I}_d)) = \mathrm{d}_{\mathrm{TV}}(N(\boldsymbol{\mu}', \mathbf{I}_d), N(\widehat{\boldsymbol{\mu}}', \mathbf{I}_d)) \leq 10\varepsilon$; see the coupling characterization of TV in (Devroye et al., 2018).

**Correctness of $\widehat{\boldsymbol{\mu}}$ output.** TESTANDOPTIMIZEMEAN (Algorithm 1) has two possible outputs for $\widehat{\boldsymbol{\mu}}$:
*Case 1*: $\widehat{\boldsymbol{\mu}} = \mathrm{argmin}_{\|\boldsymbol{\beta}\|_1 \leq \lambda} \frac{1}{n} \sum_{i=1}^n \|\mathbf{y}_i - \boldsymbol{\beta}\|_2^2$, which can only happen when `Outcome` is $\lambda \in \mathbb{R}$ and $\lambda < \varepsilon\sqrt{d}$
*Case 2*: $\widehat{\boldsymbol{\mu}} = \frac{1}{n} \sum_{i=1}^n \mathbf{y}_i$

Conditioned on APPROXL1 succeeding, with probability at least $1 - \delta$, we will show that $\mathrm{d}_{\mathrm{TV}}(N(\boldsymbol{\mu}, \mathbf{I}_d), N(\widehat{\boldsymbol{\mu}}, \mathbf{I}_d)) \leq \varepsilon$ and failure probability at most $\delta$ in each of these cases, which implies the theorem statement.

*Case 1:* Using $r = \lambda$ as the upper bound, Lemma 2.3 tells us that $\mathrm{d}_{\mathrm{TV}}(N(\boldsymbol{\mu}, \mathbf{I}_d), N(\widehat{\boldsymbol{\mu}}, \mathbf{I}_d)) \leq \varepsilon$ with failure probability at most $\delta$ when $\widetilde{\mathcal{O}}(\lambda^2/\varepsilon^4)$ i.i.d. samples are used.

*Case 2:* With $\widetilde{\mathcal{O}}(d/\varepsilon^2)$ samples, it is known that the empirical mean $\widehat{\boldsymbol{\mu}}$ achieves $\mathrm{d}_{\mathrm{TV}}(N(\boldsymbol{\mu}, \mathbf{I}_d), N(\widehat{\boldsymbol{\mu}}, \mathbf{I}_d)) \leq \varepsilon$ with failure probability at most $\delta$; see Lemma A.12.

**Sample complexity used.** By Definition 2.1, APPROXL1 uses $|\mathbf{S}| = m(k, \alpha, \delta') \in \widetilde{\mathcal{O}}(\sqrt{k}/\alpha^2)$ samples to produce `Outcome`. Then, APPROXL1 further uses $\widetilde{\mathcal{O}}(\lambda^2/\varepsilon^4)$ samples or $\widetilde{\mathcal{O}}(d/\varepsilon^2)$ samples depending on whether $\lambda < \varepsilon\sqrt{d}$. So, TESTANDOPTIMIZEMEAN has a total sample complexity of $\widetilde{\mathcal{O}}\left(\frac{\sqrt{k}}{\alpha^2} + \min\left\{\frac{\lambda^2}{\varepsilon^4}, \frac{d}{\varepsilon^2}\right\}\right)$. Meanwhile, Lemma 2.2 states that $\|\boldsymbol{\mu}\|_1 \leq \lambda \leq 2\sqrt{k} \cdot (\lceil d/k \rceil \cdot \alpha + 2\|\boldsymbol{\mu}\|_1)$ whenever `Outcome` is $\lambda \in \mathbb{R}$. Since $(a + b)^2 \leq 2a^2 + 2b^2$ for any two real numbers $a, b \in \mathbb{R}$, we see that $\frac{\lambda^2}{\varepsilon^4} \in \mathcal{O}\left(\frac{k}{\varepsilon^4} \cdot \left(\frac{d^2\alpha^2}{k^2} + \|\boldsymbol{\mu}\|_1^2\right)\right) \subseteq \mathcal{O}\left(\frac{d}{\varepsilon^2} \cdot \left(\frac{d\alpha^2}{\varepsilon^2 k} + \frac{k \cdot \|\boldsymbol{\mu}\|_1^2}{d\varepsilon^2}\right)\right)$. Putting together the above observations, we see that the total sample complexity is

$$\widetilde{\mathcal{O}}\left(\frac{\sqrt{k}}{\alpha^2} + \frac{d}{\varepsilon^2} \cdot \min\left\{1, \frac{d\alpha^2}{\varepsilon^2 k} + \frac{k \cdot \|\boldsymbol{\mu}\|_1^2}{d\varepsilon^2}\right\}\right)$$

Recalling that $\boldsymbol{\mu}$ in the analysis above actually refers to the pre-processed $\boldsymbol{\mu} - \widetilde{\boldsymbol{\mu}}$, and that TESTANDOPTIMIZEMEAN sets $k = \lceil d^{4\eta} \rceil$ and $\alpha = \varepsilon d^{-(1-3\eta)/2}$, with $0 \leq \eta \leq \frac{1}{4}$, the above expression simplifies to

$$\widetilde{\mathcal{O}}\left(\frac{d}{\varepsilon^2} \cdot \left(d^{-\eta} + \min\{1, f(\boldsymbol{\mu}, \widetilde{\boldsymbol{\mu}}, d, \eta, \varepsilon)\}\right)\right)$$

where $f(\boldsymbol{\mu}, \widetilde{\boldsymbol{\mu}}, d, \eta, \varepsilon) = \frac{\|\boldsymbol{\mu} - \widetilde{\boldsymbol{\mu}}\|_1^2}{d^{1-4\eta}\varepsilon^2}$. $\qquad\square$

**Remark on setting upper bound $\zeta$.** As $\zeta$ only affects the sample complexity logarithmically, one may be tempted to use a larger value than $\zeta = 4\varepsilon\sqrt{d}$. However, observe that running APPROXL1 with a larger upper bound than $\zeta = 4\varepsilon\sqrt{d}$ would not be helpful since $\|\boldsymbol{\mu}\|_2 > \zeta/4$ whenever APPROXL1 currently returns Fail and we have $\|\boldsymbol{\mu}\|_1 \leq \lambda$ whenever APPROXL1 returns $\lambda \in \mathbb{R}$. So, $\varepsilon\sqrt{d} = \zeta/4 < \|\boldsymbol{\mu}\|_2 \leq \|\boldsymbol{\mu}\|_1 \leq \lambda$ and TESTANDOPTIMIZEMEAN would have resorted to using the empirical mean anyway.

**Remark about early termination without the optimization step.** If there is no Fail amongst $\{o_1, \ldots, o_w\}$ and $4\sum_{j=1}^w o_j^2 \leq \varepsilon^2$ after Line 10 of APPROXL1, then we could have just output $\widehat{\boldsymbol{\mu}} = \mathbf{0}_d$ without running the optimization step. This ie because since $4\sum_{j=1}^w o_j^2 \leq \varepsilon^2$ would imply $\|\boldsymbol{\mu}\|_2 \leq \varepsilon$ via $\|\boldsymbol{\mu}\|_2^2 = \sum_{j=1}^w \|\boldsymbol{\mu}_{\mathbf{B}_j}\|_2^2 \leq \sum_{j=1}^w (2o_j)^2 = 4\sum_{j=1}^w o_j^2 \leq \varepsilon^2$ and thus $\mathrm{d}_{\mathrm{TV}}(N(\boldsymbol{\mu}, \mathbf{I}_d), N(\widehat{\boldsymbol{\mu}}, \mathbf{I}_d)) \leq \sqrt{\frac{1}{2} \cdot \mathrm{d}_{\mathrm{KL}}(N(\boldsymbol{\mu}, \mathbf{I}_d), N(\widehat{\boldsymbol{\mu}}, \mathbf{I}_d))} = \sqrt{\frac{1}{4} \cdot \|\boldsymbol{\mu} - \mathbf{0}\|_2^2} \leq \sqrt{\frac{\varepsilon^2}{4}} \leq \varepsilon$ via Pinsker's inequality (Theorem A.10).

## 3. Hardness for the identity covariance setting

Theorem 1.3 is implied by Lemma 3.2, which depends on the following corollary of Fano's inequality.

**Lemma 3.1** (Lemma 6.1 of (Ashtiani et al., 2020))**.** *Let $\kappa : \mathbb{R} \to \mathbb{R}$ be a function and let $\mathcal{F}$ be a class of distributions such that, for all $\varepsilon > 0$, there exist distributions $f_1, \ldots, f_M \in \mathcal{F}$ such that $\mathrm{d}_{\mathrm{KL}}(f_i, f_j) \leq \kappa(\varepsilon)$ and $\mathrm{d}_{\mathrm{TV}}(f_i, f_j) > 2\varepsilon$ for all $i \neq j \in [M]$. Then any method that learns $\mathcal{F}$ to within total variation distance $\varepsilon$ with probability $\geq 2/3$ has sample complexity $\Omega\left(\frac{\log M}{\kappa(\varepsilon)\log(1/\varepsilon)}\right)$.*

**Lemma 3.2.** *Suppose we are given sample access to $N(\boldsymbol{\mu}, \mathbf{I}_d)$ for some unknown $\boldsymbol{\mu} \in \mathbb{R}^d$, and an advice $\widetilde{\boldsymbol{\mu}} \in \mathbb{R}^d$. Then, any algorithm that $(\varepsilon, \frac{2}{3})$-PAC learns $N(\boldsymbol{\mu}, \mathbf{I}_d)$ requires $\widetilde{\Omega}\left(\min\left\{\frac{\|\boldsymbol{\mu} - \widetilde{\boldsymbol{\mu}}\|_1^2}{\varepsilon^4}, \frac{d}{\varepsilon^2}\right\}\right)$ samples.*

*Proof.* Without loss of generality, we can consider $\widetilde{\boldsymbol{\mu}} = 0$ since we can sample from $N(\boldsymbol{\mu} - \widetilde{\boldsymbol{\mu}}, \mathbf{I}_d)$ by sampling $N(\boldsymbol{\mu}, \mathbf{I}_d)$ and subtracting $\widetilde{\boldsymbol{\mu}}$ from each sample. Let $\widehat{\boldsymbol{\mu}}$ denote the output of the learning algorithm.

Recall that $\mathrm{d}_{\mathrm{KL}}(N(\boldsymbol{\mu}, \mathbf{I}_d), N(\boldsymbol{\mu}', \mathbf{I}_d)) = \frac{1}{2}\|\boldsymbol{\mu} - \boldsymbol{\mu}'\|_2^2$. For $\|\boldsymbol{\mu} - \boldsymbol{\mu}'\|_2 \leq 1$, it is known that $\mathrm{d}_{\mathrm{TV}}(N(\boldsymbol{\mu}, \mathbf{I}_d), N(\boldsymbol{\mu}', \mathbf{I}_d)) \in \left[\frac{\|\boldsymbol{\mu} - \boldsymbol{\mu}'\|_2}{200}, \frac{\|\boldsymbol{\mu} - \boldsymbol{\mu}'\|_2}{2}\right]$; see (Devroye et al., 2018). Now, for any $\varepsilon' > 0$, we show the existence of $M = 2^{\Omega\left(\min\left\{d, \frac{\lambda^2}{\varepsilon^2}\right\}\right)}$ distributions $\{f_i\}_{i=1}^M$, $f_i \triangleq N(\boldsymbol{\mu}_i, \mathbf{I}_d)$, with $\|\boldsymbol{\mu}_i - \widetilde{\boldsymbol{\mu}}\|_1 = \lambda$ and $\|\boldsymbol{\mu}_i - \boldsymbol{\mu}_j\|_2 \in [\varepsilon', 2\varepsilon'] \forall i \neq j \in [M]$.. As long as $\varepsilon' \leq \frac{1}{2}$, the above implies that $\mathrm{d}_{\mathrm{TV}}(f_i, f_j) \geq \frac{\varepsilon'}{200}$, and $\min\{\mathrm{d}_{\mathrm{KL}}(f_i\|f_j), \mathrm{d}_{\mathrm{KL}}(f_j\|f_i)\} \leq 2(\varepsilon')^2$.

Taking $\varepsilon = \frac{\varepsilon'}{400}$, such a cover $f_1, \ldots, f_M$ will satisfy

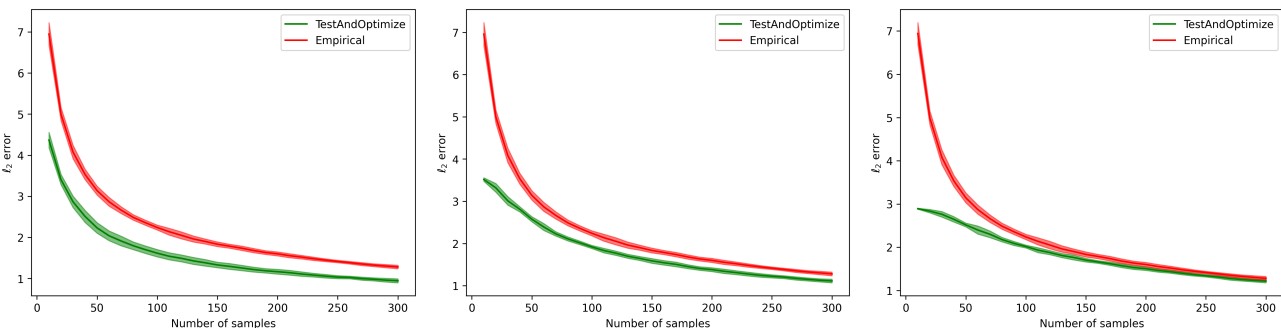

Figure 2: Here, $d = 500$, $s = \{100, 200, 300\}$, and $q = \|\boldsymbol{\mu} - \widetilde{\boldsymbol{\mu}}\|_1 = 50$. Error bars show standard deviation over 10 runs.

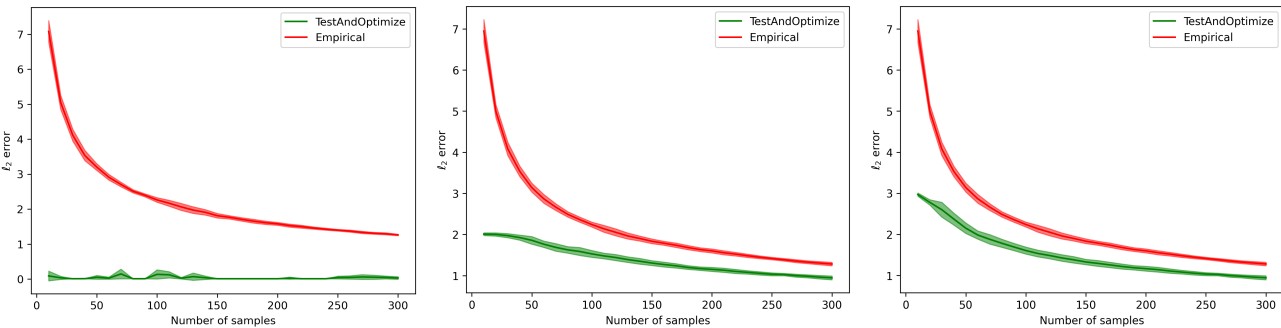

Figure 3: Here, $d = 500$, $s = 100$, and $q = \|\boldsymbol{\mu} - \widetilde{\boldsymbol{\mu}}\|_1 \in \{0.1, 20, 30\}$. Error bars show standard deviation over 10 runs.

the conditions of Lemma 3.1 with $\kappa(\varepsilon) = 2 \cdot 400^2 \cdot \varepsilon^2$. This gives a sample complexity lower bound of $\Omega\left(\min\left\{d, \frac{\|\boldsymbol{\mu}_i - \widetilde{\boldsymbol{\mu}}\|_1^2}{\varepsilon^2}\right\} \cdot \frac{1}{\varepsilon^2 \log(1/\varepsilon)}\right)$ for $(\varepsilon, \frac{2}{3})$-PAC learning in TV distance given advice.

Construct $\boldsymbol{\mu}_1, \ldots, \boldsymbol{\mu}_M$ as follows: Choose $k = \min\left\{d, \left\lceil \frac{\lambda^2}{\varepsilon^2} \right\rceil\right\}$. By the Gilbert-Varshamov bound, for any $k > 4$, there exists a code $C \subseteq \{0, 1\}^k$ with pairwise Hamming distance $\in [k/4, k]$ such that $|C| \geq \frac{2^{k-1}}{\sum_{i=0}^{k/4-1}\binom{k}{i}} \geq 2^{\Omega(k)}$ (via Stirling's approximation).

With $M = 2^{\Omega(k)}$, choose such a code $C$ and get $\{\boldsymbol{v}_1, \ldots, \boldsymbol{v}_M\} \subseteq \{\pm 1\}^k$ by applying $(x_1, \ldots, x_k) \mapsto ((-1)^{x_1}, \ldots, (-1)^{x_k})$ to each $\boldsymbol{x} \in C$. The first $k$ coordinates of each $\boldsymbol{\mu}_i \in \mathbb{R}^d$ are set to $\frac{\lambda}{k} \cdot \boldsymbol{v}_i$ and the remaining $d - k$ coordinates are set to 0. Then, by construction, $\|\boldsymbol{\mu}_i - \widetilde{\boldsymbol{\mu}}\|_1 = \|\boldsymbol{\mu}_i\|_1 = k\left(\frac{\lambda}{k}\right) = \lambda$ for each $\boldsymbol{\mu}_i$, and $\|\boldsymbol{\mu}_i - \boldsymbol{\mu}_j\|_2 = \left(2\frac{\lambda}{k}\right)\sqrt{\|\boldsymbol{v}_i - \boldsymbol{v}_j\|_0}$. Thus, we will have $\|\boldsymbol{\mu}_i - \boldsymbol{\mu}_j\|_2 \in \left[\frac{\lambda}{\sqrt{k}}, \frac{2\lambda}{\sqrt{k}}\right]$ for each $i \neq j \in [M]$. $\square$

## 4. Experiments

Here, we explore the sample complexity gains in the identity covariance setting when one is given high quality advice, specifically the benefits of performing the optimization in

line 8 of Algorithm 1 versus returning the empirical mean as in line 11. As such, we do *not* invoke ApproxL1 but instead explore how to $\|\boldsymbol{\mu} - \widehat{\boldsymbol{\mu}}_{\mathrm{ALG}}\|_2$ behaves as a function of $\|\boldsymbol{\mu} - \widehat{\boldsymbol{\mu}}\|_1$ and number of samples, where ALG is either our TestAndOptimize approach or simply computing the empirical mean. For reproducibility, our code and scripts are provided in the supplementary materials.

We perform two experiments on multivariate Gaussians of dimension $d = 500$ while varying two parameters: sparsity $s \in [d]$ and advice quality $q \in \mathbb{R}_{\geq 0}$. In both experiments, the difference vector $\boldsymbol{\mu} - \widetilde{\boldsymbol{\mu}} \in \mathbb{R}^d$ is generated with random $\pm q/s$ values in the first $s$ coordinates and zeros in the remaining $d - s$ coordinates. In the first experiment (see Figure 2), we fix $q = 50$ and vary $s \in \{100, 200, 300\}$. In the second experiment (see Figure 3), we fix $s = 100$ and vary $q \in \{0.1, 20, 30\}$. In both experiments, we see that TestAndOptimize beats the empirical mean estimate in terms of incurred $\ell_2$ error (which translate directly to $\mathrm{d}_{\mathrm{TV}}$), with the diminishing benefits as $q$ or $s$ increases.

For computational efficiency, we solve the LASSO optimization in its Lagrangian form $\widehat{\boldsymbol{\mu}} = \operatorname{argmin}_{\boldsymbol{\beta} \in \mathbb{R}^d} \frac{1}{n} \sum_{i=1}^{n} \|\mathbf{y}_i - \boldsymbol{\beta}\|_2^2 + \lambda\|\boldsymbol{\beta}\|_1$, using the `LassoLarsCV` method in `scikit-learn`, instead of the equivalent penalized form. The value of the hyperparameter $\lambda$ is chosen using 5-fold cross-validation.

# 5. Conclusion

We propose a learning-augmented algorithm for learning multivariate Gaussians that incorporates property testing as a subroutine, where the advice quality is stated in terms of $\ell_1$ error, and provide matching information-theoretic lower bounds. While running our experiments, we observe an interesting phenomenon: the rate of improvement does not worsen as $\ell_1$ increases if we fixed the $\ell_0$ sparsity; see Appendix E. As such, it would be interesting to show theoretical guarantees with advice error in the $\ell_0$-norm.

# Acknowledgements

- AB: This research was supported by the National Research Foundation (NRF), Prime Minister's Office, Singapore under its Campus for Research Excellence and Technological Enterprise (CREATE) programme, by the NRF-AI Fellowship R-252-100-B13-281, Amazon Faculty Research Award, and Google South & Southeast Asia Research Award.

- DC: This research/project is supported by the National Research Foundation, Singapore under its AI Singapore Programme (AISG Award No: AISG-PhD/2021-08-013).

- TG: This research was partially supported by MoE AcRF Tier 1 A-8000980-00-00 while at NUS and by an NTU startup grant while at NTU.

# Impact Statement

This paper presents work whose goal is to advance the field of Machine Learning. It addresses the abstract problem of PAC learning high-dimensional Gaussians, and hence does not have any direct societal impact. There are potential indirect societal consequences of our work (through algorithms that use Gaussian learning), none which we feel must be specifically highlighted here.

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

# A. Preliminaries

For any integer $d \geq 1$, we write $[d]$ to mean the set of integers $\{1, \ldots, d\}$. We will write $\mathbf{v} \sim N(\boldsymbol{\mu}, \boldsymbol{\Sigma})$ to mean drawing a multivariate Gaussian sample and $\mathcal{M} = \{\mathbf{v}_1, \ldots, \mathbf{v}_{|\mathcal{M}|}\}$ to mean a collection of $|\mathcal{M}|$ independently drawn such vectors.

In the rest of this section, we will state some basic facts and lemmas that would be useful for our work. Most of them are folklore results and we supplement proofs for them when we could not nail down a direct reference.

## A.1. Matrix facts

**Fact A.1** (e.g. see Exercise 5.4.P3 of (Horn & Johnson, 2012)). *Let $\mathbf{x} \in \mathbb{R}^d$ be an arbitrary $d$-dimensional real vector. Then, the $\ell_1$ and $\ell_2$ norms of $\mathbf{x}$ are defined as $\|\mathbf{x}\|_1 = \sum_{i=1}^d |\mathbf{x}_i|$ and $\|\mathbf{x}\|_2 = \sqrt{\sum_{i=1}^d \mathbf{x}_i^2}$ respectively. They satisfy the inequality: $\|\mathbf{x}\|_2 \leq \|\mathbf{x}\|_1 \leq \sqrt{d} \cdot \|\mathbf{x}\|_2$.*

For a real matrix $\mathbf{M} \in \mathbb{R}^{d \times d}$, we define its vectorized form $\mathrm{vec}(M) \in \mathbb{R}^{d^2}$ by $\mathrm{vec}(\mathbf{M}) = (\mathbf{M}_{1,1}, \ldots, \mathbf{M}_{d,d})$ and we see that $\|\mathbf{M}\|_F^2 = \|\mathrm{vec}(\mathbf{M})\|_2^2$. We recover a matrix given its vectorized form via $\mathbf{M} = \mathrm{mat}(\mathrm{vec}(\mathbf{M}))$. For any matrix $\mathbf{A}$, we use $\sigma_{\min}(\mathbf{A})$ to denote its smallest eigenvalue. Note that for any full rank matrix $\mathbf{A} \in \mathbb{R}^{d \times d}$, we have $\frac{1}{\|\mathbf{A}\|_2} \leq \|\mathbf{A}^{-1}\|_2$, $\|\mathbf{A}\|_2 \leq \|\mathbf{A}\|_F \leq \sqrt{d} \cdot \|\mathbf{A}\|_2$ (e.g. see Exercise 5.6.P23 of (Horn & Johnson, 2012)), and $\|\mathbf{A}\|_F = \|\mathrm{vec}(\mathbf{A})\|_2 \leq \|\mathrm{vec}(\mathbf{A})\|_1 \leq \sqrt{d} \cdot \|\mathrm{vec}(\mathbf{A})\|_2$. For any two matrices $\mathbf{A}$ and $\mathbf{B}$ of the same dimension, we also know that $\|\mathbf{AB}\|_F \leq \min\{\|\mathbf{A}\|_2\|\mathbf{B}\|_F, \|\mathbf{A}\|_F\|\mathbf{B}\|_2\}$.

**Lemma A.2** (Chapter 5.6 of (Horn & Johnson, 2012)). *Let $\mathbf{A}$ and $\mathbf{B}$ be two square real matrices where $\mathbf{A}$ is an invertible matrix. Then, $\|\mathbf{AB}\| = \|\mathbf{BA}\|$.*

*Proof.* Exercise 5.6.P58(b) of (Horn & Johnson, 2012) tells us that $\|\mathbf{AB}\| = \|\mathbf{BA}\|$ when $\mathbf{A}$ normal and $\mathbf{B}$ is Hermitian. Since normal matrices are invertible and every real matrix is Hermitian, the claim follows. $\square$

**Lemma A.3.** *Let $\mathbf{A}$ and $\mathbf{B}$ be two square $d \times d$ matrices where $\mathbf{A}$ is an invertible matrix with a square root. Then, $\|\mathbf{A}^{-1/2}\mathbf{B}\mathbf{A}^{-1/2} - I\| = \|\mathbf{A}^{-1}\mathbf{B} - \mathbf{I}_d\|$*

*Proof.* $\|\mathbf{A}^{-1/2}\mathbf{B}\mathbf{A}^{-1/2} - \mathbf{I}_d\| = \|(\mathbf{A}^{-1/2}\mathbf{B} - \mathbf{A}^{1/2})\mathbf{A}^{-1/2}\| = \|\mathbf{A}^{-1/2}(\mathbf{A}^{-1/2}\mathbf{B} - \mathbf{A}^{1/2})\| = \|\mathbf{A}^{-1}\mathbf{B} - \mathbf{I}_d\|$. $\square$

**Definition A.4** (Projected vector). Let $\mathbf{x} = (\mathbf{x}_1, \ldots, \mathbf{x}_d) \in \mathbb{R}^d$ be a $d$-dimensional vector and $\mathbf{B} = \{i_1, \ldots, i_w\} \subseteq [d]$ be a subset of $1 \leq w \leq d$ indices, where $i_1 < \ldots < i_w$. Then, we define $\mathbf{x_B} = (\mathbf{x}_{i_1}, \ldots, \mathbf{x}_{i_w}) \in \mathbb{R}^w$ as the projection of the vector $\mathbf{x}$ to the coordinates indicated by $\mathbf{B}$.

**Lemma A.5** (Trace inequality). *For any three matrices $\mathbf{A}, \mathbf{B}, \mathbf{C} \in \mathbb{R}^{d \times d}$, we have $\mathrm{Tr}(\mathbf{ABC}) \leq \|\mathrm{vec}(\mathbf{BA})\|_1 \cdot \|\mathbf{C}\|_2$.*

*Proof.* Let $\lambda_1(\mathbf{M}), \ldots, \lambda_d(\mathbf{M})$ denote the eigenvalues of a matrix $\mathbf{M} \in \mathbb{R}^{d \times d}$.

$$
\begin{aligned}
\mathrm{Tr}(\mathbf{ABC}) &\leq \sum_i \lambda_i(\mathbf{AB}) \cdot \lambda_i(\mathbf{C}) && \text{(by von Neumann trace inequality)} \\
&= \sum_i \lambda_i(\mathbf{BA}) \cdot \lambda_i(\mathbf{C}) && \text{(e.g. see Theorem 1.3.22 of (Horn \& Johnson, 2012))} \\
&\leq \sum_i |\lambda_i(\mathbf{BA}) \cdot \lambda_i(\mathbf{C})| \\
&\leq \left\| \begin{pmatrix} \lambda_1(\mathbf{BA}) \\ \vdots \\ \lambda_d(\mathbf{BA}) \end{pmatrix} \right\|_1 \cdot \left\| \begin{pmatrix} \lambda_1(\mathbf{C}) \\ \vdots \\ \lambda_d(\mathbf{C}) \end{pmatrix} \right\|_\infty && \text{(Hölder's inequality)} \\
&= \sum_i |\lambda_i(\mathbf{BA})| \cdot \max_i \lambda_i(\mathbf{C}) && \text{(Definitions of vector } \ell_1 \text{ and } \ell_\infty \text{ norms)} \\
&\leq \sum_i |\lambda_i(\mathbf{BA})| \cdot \|\mathbf{C}\|_2 && \text{(Definition of matrix spectral norm)}
\end{aligned}
$$

It remains to argue that $\sum_i |\lambda_i(\mathbf{BA})| \le \|\text{vec}(\mathbf{BA})\|_1$. To this end, consider the singular value decomposition (SVD) of $\mathbf{BA} = \mathbf{U\Sigma V}^\top$ with unitary matrices $\mathbf{U}, \mathbf{V}$ and diagonal matrix $\mathbf{\Sigma} = \text{diag}(\sigma_1, \dots, \sigma_d)$. Let us denote the eigenvalues of $\mathbf{BA}$ by $\sigma_1, \dots, \sigma_d$ and the columns of $\mathbf{BA}$ by $\mathbf{z}_1, \dots, \mathbf{z}_d \in \mathbb{R}^d$. Then,

$$
\begin{aligned}
\sum_i |\lambda_i(\mathbf{BA})| &\le \sum_i \sigma_i && \text{(e.g. see Equation (7.3.17) in (Horn \& Johnson, 2012))} \\
&= \text{Tr}(\mathbf{\Sigma}) && \text{(By definition of } \mathbf{\Sigma}) \\
&= \text{Tr}(\mathbf{V}^\top \mathbf{V}\mathbf{U}^\top \mathbf{U}\mathbf{\Sigma}) && \text{(Since } \mathbf{U} \text{ and } \mathbf{V} \text{ are unitary matrices)} \\
&= \text{Tr}(\mathbf{V}\mathbf{U}^\top \mathbf{U}\mathbf{\Sigma}\mathbf{V}^\top) && \text{(By cyclic property of trace)} \\
&= \text{Tr}(\mathbf{V}\mathbf{U}^\top \mathbf{BA}) && \text{(By SVD of } \mathbf{BA}) \\
&= \sum_{i=1}^d (\mathbf{V}\mathbf{U}^\top \mathbf{z}_i)_i && \text{(By definition of trace)} \\
&\le \sum_{i=1}^d \|\mathbf{V}\mathbf{U}^\top \mathbf{z}_i\|_2 && \text{(Since } (\mathbf{V}\mathbf{U}^\top \mathbf{z}_i)_i^2 \text{ is just one term in summation of } \|\mathbf{V}\mathbf{U}^\top \mathbf{z}_i\|_2^2) \\
&= \sum_{i=1}^d \|\mathbf{z}_i\|_2 && \text{(Since } \mathbf{U} \text{ and } \mathbf{V} \text{ are unitary matrices)} \\
&\le \sum_{i=1}^d \|\mathbf{z}_i\|_1 && \text{(Since } \ell_2 \le \ell_1) \\
&= \sum_{i=1}^d \sum_{j=1}^d |(\mathbf{BA})_{i,j}| && \text{(By definition of vector } \ell_1 \text{ norm)} \\
&= \|\text{vec}(\mathbf{BA})\|_1 && \text{(By definition of } \|\text{vec}(\mathbf{BA})\|_1)
\end{aligned}
$$

Putting together, we get $\text{Tr}(\mathbf{ABC}) \le \sum_i |\lambda_i(\mathbf{BA})| \cdot \|\mathbf{C}\|_2 \le \|\text{vec}(\mathbf{BA})\|_1 \cdot \|\mathbf{C}\|_2$ as desired. $\qquad\square$

**Lemma A.6.** *For any two matrices $\mathbf{A}, \mathbf{B} \in \mathbb{R}^{d\times d}$, we have $\|\text{vec}(\mathbf{A}+\mathbf{B})\|_1 \le \|\text{vec}(\mathbf{A})\|_1 + \|\text{vec}(\mathbf{B})\|_1$ and $\|\text{vec}(\mathbf{AB})\|_1 \le \|\text{vec}(\mathbf{A})\|_1 \cdot \|\text{vec}(\mathbf{B})\|_1$.*

*Proof.* To see $\|\text{vec}(\mathbf{A} + \mathbf{B})\|_1 \le \|\text{vec}(\mathbf{A})\|_1 + \|\text{vec}(\mathbf{B})\|_1$, observe that

$$
\|\text{vec}(\mathbf{A} + \mathbf{B})\|_1 = \sum_{i=1}^d \sum_{j=1}^d |\mathbf{A}_{ij} + \mathbf{B}_{ij}| \le \sum_{i=1}^d \sum_{j=1}^d |\mathbf{A}_{ij}| + \sum_{i=1}^d \sum_{j=1}^d |\mathbf{B}_{ij}| = \|\text{vec}(\mathbf{A})\|_1 + \|\text{vec}(\mathbf{B})\|_1
$$

To see $\|\text{vec}(\mathbf{AB})\|_1 \le \|\text{vec}(\mathbf{A})\|_1 \cdot \|\text{vec}(\mathbf{B})\|_1$, observe that

$$
\|\text{vec}(\mathbf{AB})\|_1 = \sum_{i=1}^d \sum_{j=1}^d \sum_{k=1}^d |\mathbf{A}_{ij}\mathbf{B}_{jk}| \le \left( \sum_{i=1}^d \sum_{j=1}^d |\mathbf{A}_{ij}| \right) \cdot \left( \sum_{j=1}^d \sum_{k=1}^d |\mathbf{B}_{jk}| \right) = \|\text{vec}(\mathbf{A})\|_1 \cdot \|\text{vec}(\mathbf{B})\|_1
$$

$\qquad\square$

## A.2. Distance measures between distributions

**Definition A.7** (Kullback–Leibler (KL) divergence)**.**
For two continuous distributions $\mathcal{P}$ and $\mathcal{Q}$ over $\mathbf{X}$,

$$
d_{\text{KL}}(\mathcal{P}, \mathcal{Q}) = \int_{\mathbf{x} \in \mathbf{X}} \mathcal{P}(\mathbf{x}) \log \left( \frac{\mathcal{P}(\mathbf{x})}{\mathcal{Q}(\mathbf{x})} \right) d\mathbf{x}
$$

Note that KL divergence is not symmetric in general.

**Lemma A.8** (Known fact about KL divergence). *Given two $d$-dimensional multivariate Gaussian distributions $\mathcal{P} \sim N(\boldsymbol{\mu}_{\mathcal{P}}, \boldsymbol{\Sigma}_{\mathcal{P}})$ and $\mathcal{Q} \sim N(\boldsymbol{\mu}_{\mathcal{Q}}, \boldsymbol{\Sigma}_{\mathcal{Q}})$ where $\boldsymbol{\Sigma}_{\mathcal{P}}$ and $\boldsymbol{\Sigma}_{\mathcal{Q}}$ are invertible, we have*

$$d_{\mathrm{KL}}(\mathcal{P}, \mathcal{Q}) = \frac{1}{2} \cdot \left( \mathrm{Tr}(\boldsymbol{\Sigma}_{\mathcal{Q}}^{-1}\boldsymbol{\Sigma}_{\mathcal{P}}) - d + (\boldsymbol{\mu}_{\mathcal{Q}} - \boldsymbol{\mu}_{\mathcal{P}})^{\top}\boldsymbol{\Sigma}_{\mathcal{Q}}^{-1}(\boldsymbol{\mu}_{\mathcal{Q}} - \boldsymbol{\mu}_{\mathcal{P}}) + \ln\left(\frac{\det \boldsymbol{\Sigma}_{\mathcal{Q}}}{\det \boldsymbol{\Sigma}_{\mathcal{P}}}\right) \right)$$

$$\leq \frac{1}{2} \cdot \left( (\boldsymbol{\mu}_{\mathcal{Q}} - \boldsymbol{\mu}_{\mathcal{P}})^{\top}\boldsymbol{\Sigma}_{\mathcal{Q}}^{-1}(\boldsymbol{\mu}_{\mathcal{Q}} - \boldsymbol{\mu}_{\mathcal{P}}) + \|\mathbf{X}\|_F^2 \right)$$

*where $\mathbf{X} = \boldsymbol{\Sigma}_{\mathcal{Q}}^{-1/2}\boldsymbol{\Sigma}_{\mathcal{P}}\boldsymbol{\Sigma}_{\mathcal{Q}}^{-1/2} - \mathbf{I}_d$ with eigenvalues $\lambda_1, \ldots, \lambda_d$. In particular, $d_{\mathrm{KL}}(\mathcal{P}, \mathcal{Q}) = \frac{1}{2}\|\boldsymbol{\mu}_{\mathcal{Q}} - \boldsymbol{\mu}_{\mathcal{P}}\|_2^2$ when $\boldsymbol{\Sigma}_{\mathcal{P}} = \boldsymbol{\Sigma}_{\mathcal{Q}} = \mathbf{I}_d$ and $d_{\mathrm{KL}}(\mathcal{P}, \mathcal{Q}) \leq \frac{1}{2}\|\mathbf{X}\|_F^2$ when $\boldsymbol{\mu}_{\mathcal{P}} = \boldsymbol{\mu}_{\mathcal{Q}}$.*

*Proof.* Let $\mathcal{P} \sim N(\boldsymbol{\mu}_{\mathcal{P}}, \boldsymbol{\Sigma}_{\mathcal{P}})$ and $\mathcal{Q} \sim N(\boldsymbol{\mu}_{\mathcal{Q}}, \boldsymbol{\Sigma}_{\mathcal{Q}})$ be two $d$-dimensional multivariate Gaussian distributions where $\boldsymbol{\Sigma}_{\mathcal{P}}$ and $\boldsymbol{\Sigma}_{\mathcal{Q}}$ are full rank invertible covariance matrices.

By definition, the KL divergence between $\mathcal{P}$ and $\mathcal{Q}$ is

$$d_{\mathrm{KL}}(\mathcal{P}, \mathcal{Q}) = \frac{1}{2} \cdot \left( \mathrm{Tr}(\boldsymbol{\Sigma}_{\mathcal{Q}}^{-1}\boldsymbol{\Sigma}_{\mathcal{P}}) - d + (\boldsymbol{\mu}_{\mathcal{Q}} - \boldsymbol{\mu}_{\mathcal{P}})^{\top}\boldsymbol{\Sigma}_{\mathcal{Q}}^{-1}(\boldsymbol{\mu}_{\mathcal{Q}} - \boldsymbol{\mu}_{\mathcal{P}}) + \ln\left(\frac{\det \boldsymbol{\Sigma}_{\mathcal{Q}}}{\det \boldsymbol{\Sigma}_{\mathcal{P}}}\right) \right) \tag{4}$$

Let us define the matrix $\mathbf{X} = \boldsymbol{\Sigma}_{\mathcal{Q}}^{-1/2}\boldsymbol{\Sigma}_{\mathcal{P}}\boldsymbol{\Sigma}_{\mathcal{Q}}^{-1/2} - \mathbf{I}_d$ with eigenvalues $\lambda_1, \ldots, \lambda_d$. Note that $\mathbf{X}$ is invertible because $\boldsymbol{\Sigma}_{\mathcal{P}}$ and $\boldsymbol{\Sigma}_{\mathcal{Q}}$ are invertible, so $\lambda_1, \ldots, \lambda_d > 0$. Then, Equation (4) can be upper bounded as

$$d_{\mathrm{KL}}(\mathcal{P}, \mathcal{Q}) = \frac{1}{2} \cdot \left( \mathrm{Tr}(\boldsymbol{\Sigma}_{\mathcal{Q}}^{-1}\boldsymbol{\Sigma}_{\mathcal{P}}) - d + (\boldsymbol{\mu}_{\mathcal{Q}} - \boldsymbol{\mu}_{\mathcal{P}})^{\top}\boldsymbol{\Sigma}_{\mathcal{Q}}^{-1}(\boldsymbol{\mu}_{\mathcal{Q}} - \boldsymbol{\mu}_{\mathcal{P}}) + \ln\left(\frac{\det \boldsymbol{\Sigma}_{\mathcal{Q}}}{\det \boldsymbol{\Sigma}_{\mathcal{P}}}\right) \right)$$

$$\leq \frac{1}{2}\left( (\boldsymbol{\mu}_{\mathcal{Q}} - \boldsymbol{\mu}_{\mathcal{P}})^{\top}\boldsymbol{\Sigma}_{\mathcal{Q}}^{-1}(\boldsymbol{\mu}_{\mathcal{Q}} - \boldsymbol{\mu}_{\mathcal{P}}) + \|\mathbf{X}\|_F^2 \right) \tag{5}$$

This is because $\mathrm{Tr}(\boldsymbol{\Sigma}_{\mathcal{Q}}^{-1}\boldsymbol{\Sigma}_{\mathcal{P}}) = \mathrm{Tr}(\boldsymbol{\Sigma}_{\mathcal{Q}}^{-1/2}\boldsymbol{\Sigma}_{\mathcal{P}}\boldsymbol{\Sigma}_{\mathcal{Q}}^{-1/2}) = \mathrm{Tr}(\mathbf{X} + \mathbf{I}_d) = \mathrm{Tr}(\mathbf{X}) + d$ and

$$-\ln\left(\frac{\det \boldsymbol{\Sigma}_{\mathcal{Q}}}{\det \boldsymbol{\Sigma}_{\mathcal{P}}}\right) = \ln\det\left(\boldsymbol{\Sigma}_{\mathcal{Q}}^{-1}\boldsymbol{\Sigma}_{\mathcal{P}}\right) = \ln\det(\mathbf{X} + \mathbf{I}_d) = \ln\prod_{i=1}^{d}(1 + \lambda_i)$$

$$= \sum_{i=1}^{d}\ln(1 + \lambda_i) \geq \sum_{i=1}^{d}(\lambda_i - \lambda_i^2) = \mathrm{Tr}(\mathbf{X}) - \sum_{i=1}^{d}\lambda_i^2 = \mathrm{Tr}(\mathbf{X}) - \|\mathbf{X}\|_F^2$$

where the inequality holds due to $\lambda_1, \ldots, \lambda_d > 0$.

When $\boldsymbol{\Sigma}_{\mathcal{P}} = \boldsymbol{\Sigma}_{\mathcal{Q}} = \mathbf{I}_d$, Equation (4) reduces to $d_{\mathrm{KL}}(\mathcal{P}, \mathcal{Q}) = \frac{1}{2}\|\boldsymbol{\mu}_{\mathcal{Q}} - \boldsymbol{\mu}_{\mathcal{P}}\|_2^2$. Meanwhile, when $\boldsymbol{\mu}_{\mathcal{P}} = \boldsymbol{\mu}_{\mathcal{Q}}$, Equation (5) reduces to $d_{\mathrm{KL}}(\mathcal{P}, \mathcal{Q}) \leq \frac{1}{2}\left(\|\mathbf{X}\|_F^2\right)$. $\square$

**Definition A.9** (Total variation (TV) distance). For two continuous distributions $\mathcal{P}$ and $\mathcal{Q}$ over domain $\mathbf{X}$, with density functions $f$ and $g$ respectively, $d_{\mathrm{TV}}(\mathcal{P}, \mathcal{Q}) = \frac{1}{2}\int_{\mathbf{x} \in \mathbf{X}}|f(\mathbf{x}) - g(\mathbf{x})|\, dx$.

**Theorem A.10** (Pinsker's inequality). *If $\mathcal{P}$ and $\mathcal{Q}$ are two probability distributions on the same measurable space, then $d_{\mathrm{TV}}(\mathcal{P}, \mathcal{Q}) \leq \sqrt{d_{\mathrm{KL}}(\mathcal{P}, \mathcal{Q})/2}$.*

### A.3. Properties of Gaussians

The following are standard results about empirical statistics of Gaussian samples.

**Lemma A.11** (Lemma C.4 in (Ashtiani et al., 2020); Corollary 5.50 in (Vershynin, 2010)). *Let $\mathbf{g}_1, \ldots, \mathbf{g}_n \sim N(\mathbf{0}, \mathbf{I}_d)$ and let $0 < \varepsilon < 1 < t$. If $n \geq c_0 \cdot \frac{t^2 d}{\varepsilon^2}$, for some absolute constant $c_0$, then*

$$\mathrm{Pr}\left( \left\| \frac{1}{n}\sum_{i=1}^{n}\mathbf{g}_i\mathbf{g}_i^{\top} - \mathbf{I}_d \right\|_2 > \varepsilon \right) \leq 2\exp(-t^2 d)$$

**Lemma A.12** (Folklore; e.g. see Appendix C of (Ashtiani et al., 2020)). *Fix $\varepsilon, \delta \in (0, 1)$. Given $2n$ i.i.d. samples $\mathbf{x}_1, \ldots, \mathbf{x}_{2n} \sim N(\boldsymbol{\mu}, \boldsymbol{\Sigma})$ for some unknown mean $\boldsymbol{\mu}$ and unknown covariance $\boldsymbol{\Sigma}$, define empirical mean and covariance as*

$$\widehat{\boldsymbol{\mu}} = \frac{1}{2n} \sum_{i=1}^{2n} \boldsymbol{x}_i \quad and \quad \widehat{\boldsymbol{\Sigma}} = \frac{1}{2n} \sum_{i=1}^{n} (\boldsymbol{x}_{2i} - \boldsymbol{x}_{2i-1})(\boldsymbol{x}_{2i} - \boldsymbol{x}_{2i-1})^\top$$

*Then,*

- *When $n \in \mathcal{O}\left(\frac{d^2 + d\log(1/\delta)}{\varepsilon^2}\right)$, we have $\Pr\left(\mathrm{d_{TV}}(N(\boldsymbol{\mu}, \boldsymbol{\Sigma}), N(\widehat{\boldsymbol{\mu}}, \widehat{\boldsymbol{\Sigma}})) \leq \varepsilon\right) \geq 1 - \delta$*

- *When $n \in \mathcal{O}\left(\frac{d + \sqrt{d\log(1/\delta)}}{\varepsilon^2}\right)$, we have $\Pr\left((\widehat{\boldsymbol{\mu}} - \boldsymbol{\mu})^\top \boldsymbol{\Sigma}^{-1}(\widehat{\boldsymbol{\mu}} - \boldsymbol{\mu}) \leq \varepsilon^2\right) \geq 1 - \delta$*

**Lemma A.13** (Properties of empirical covariance). *Let $\widehat{\boldsymbol{\Sigma}} \in \mathbb{R}^{d \times d}$ be the empirical covariance constructed from $n$ i.i.d. samples from $N(\mathbf{0}, \boldsymbol{\Sigma})$ for some unknown covariance $\boldsymbol{\Sigma}$. Then,*

- *When $n = d$, with probability 1, we have that $\widehat{\boldsymbol{\Sigma}}$ and $\boldsymbol{\Sigma}$ share the same eigenspace.*

- *Let $\lambda_1 \leq \ldots \leq \lambda_d$ and $\widehat{\lambda}_1 \leq \ldots \leq \widehat{\lambda}_d$ be the eigenvalues of $\boldsymbol{\Sigma}$ and $\widehat{\boldsymbol{\Sigma}}$ respectively. With probability at least $1 - \delta$, we have $\frac{\widehat{\lambda}_1}{\lambda_1} \leq 1 + \mathcal{O}\left(\sqrt{\frac{d + \log 1/\delta}{n}}\right)$.*

*Proof.* For item 1, let $1 \leq r \leq d$ be the rank of $\boldsymbol{\Sigma}$. We consider the case of the $d$-dimensional Gaussian with zero mean and covariance $\boldsymbol{\Gamma}_r = \begin{bmatrix} \mathbf{I}_r & \mathbf{0} \\ \mathbf{0} & \mathbf{0} \end{bmatrix}$, where $\mathbf{I}_r$ denotes the $r$-dimensional identity matrix and the zero-padding is added when $r < d$. Note that there is an invertible transformation between samples from $N(\mathbf{0}, \boldsymbol{\Gamma}_r)$ and $N(\mathbf{0}, \boldsymbol{\Sigma})$ with samples from $N(\mathbf{0}, \boldsymbol{\Gamma}_r)$ having the $r + 1, \ldots, d$ coordinates be fixed to 0. Now, let us denote the $i$-th standard basis vector by $\boldsymbol{e}_i$ and apply an induction argument on $r$ from 1 to $d$. The base case ($r = 1$) is obviously true since a single sample $\boldsymbol{x}_1$ will span $\{\boldsymbol{e}_1\}$ unless $\boldsymbol{x}_1 = \mathbf{0}$, which will happen with probability 0. When $r > 1$, by strong induction, $r$ samples $\boldsymbol{x}_1, \ldots, \boldsymbol{x}_r$ will not span $\{\boldsymbol{e}_1, \ldots, \boldsymbol{e}_r\}$ only if the $r$-th sample $\boldsymbol{x}_r$ lies in the subspace spanned by $\boldsymbol{x}_1, \ldots, \boldsymbol{x}_{r-1}$. This is a measure 0 event under the $N(\mathbf{0}, \boldsymbol{\Gamma}_r)$ measure.

For item 2, see Fact 3.4 of (Kamath et al., 2019). $\square$

**Lemma A.14.** *Fix $n \geq 1$ and $d \geq 1$. Suppose $\boldsymbol{\mu} \in \mathbb{R}^d$ is a hidden mean vector and we draw $n$ samples $\mathbf{x}_1, \ldots, \mathbf{x}_n \sim N(\boldsymbol{\mu}, \mathbf{I}_d)$. Define $\mathbf{z}_n = \frac{1}{\sqrt{n}} \sum_{i=1}^{n} \mathbf{x}_i$ and $y_n = \|\mathbf{z}_n\|_2^2$. Then,*

1. *$y_n$ follows the non-central chi-squared distribution $\chi_d'^2(\lambda)$ for $\lambda = n\|\boldsymbol{\mu}\|_2^2$. This also implies that $\mathbb{E}[y_n] = d + \lambda$ and $\mathsf{Var}(y_n) = 2d + 4\lambda$.*

2. *For any $t > 0$,*

$$\Pr(y_n > d + \lambda + t) \leq \exp\left(-\frac{d}{2}\left(\frac{t}{d + 2\lambda} - \log\left(1 + \frac{t}{d + 2\lambda}\right)\right)\right)$$

$$\leq \exp\left(-\frac{dt^2}{4(d + 2\lambda)(d + 2\lambda + t)}\right)$$

3. *For any $t \in (0, d + \lambda)$,*

$$\Pr(y_n < d + \lambda - t) \leq \exp\left(\frac{d}{2}\left(\frac{t}{d + 2\lambda} + \log\left(1 - \frac{t}{d + 2\lambda}\right)\right)\right)$$

$$\leq \exp\left(-\frac{dt^2}{4(d + 2\lambda)^2}\right)$$

*Proof.* The first item follows from the definition of the non-central chi-squared distribution, noting that the random vector $\mathbf{z}_n$ is distributed as $N(\sqrt{n} \cdot \boldsymbol{\mu}, \mathbf{I}_d)$. The second and third items follow from Theorems 3 and 4 of (Ghosh, 2021) respectively. $\qquad\square$

**Lemma A.15.** *Suppose* $\mathbf{g}_1, \ldots, \mathbf{g}_n \sim N(0, \mathbf{I}_d)$. *Then,*

$$\Pr\left( \left\| \sum_{i=1}^{n} \mathbf{g}_i \right\|_\infty \geq \sqrt{2n \log\left(\frac{2d}{\delta}\right)} \right) \leq \delta$$

*Proof.* Since $\mathbf{g}_1, \ldots, \mathbf{g}_n \sim N(0, \mathbf{I}_d)$, we see that $\mathbf{y} = \mathbf{g}_1 + \ldots + \mathbf{g}_n \sim N(0, n\mathbf{I}_d)$. Furthermore, each coordinate $i \in [d]$ of $\mathbf{y}_i = (y_1, \ldots, y_d)$ is distributed according to $N(0, n)$. By standard Gaussian tail bounds, we know that $\Pr(|y_i| \geq t) \leq 2 \exp\left(-\frac{t^2}{2n}\right)$ for any $i \in [d]$ and $t > 0$. So,

$$\begin{aligned}
\Pr\left( \left\| \sum_{i=1}^{n} \mathbf{g}_i \right\|_\infty \geq \sqrt{2n \log\left(\frac{2d}{\delta}\right)} \right) &= \Pr\left( \|\mathbf{y}\|_\infty \geq \sqrt{2n \log\left(\frac{2d}{\delta}\right)} \right) \\
&= \Pr\left( \max_{i \in [d]} \|y_i\| \geq \sqrt{2n \log\left(\frac{2d}{\delta}\right)} \right) \\
&\leq \sum_{i=1}^{d} \Pr\left( \|y_i\| \geq \sqrt{2n \log\left(\frac{2d}{\delta}\right)} \right) \quad \text{(Union bound over all } d \text{ coordinates)} \\
&\leq 2d \exp\left( -\frac{2n \log\left(\frac{2d}{\delta}\right)}{2n} \right) \quad \text{(Setting } t = 2n \log\left(\frac{2d}{\delta}\right)) \\
&= \delta
\end{aligned}$$

$\qquad\square$

# B. Additional results

## B.1. Tolerant testing

In this section, we present an algorithm for testing whether an unknown distribution is close to a standard normal distribution. More specifically, we first describe a tolerant tester for the property that the mean of an isotropic Gaussian distribution equals zero. Subsequently, we present a tolerant tester for the property that the covariance matrix equals the identity matrix.

### B.1.1. TOLERANT TESTING FOR MEAN

The definition of a tolerant tester for the mean of an isotropic Gaussian distribution is given below.

**Definition B.1** (Tolerant testing of isotropic Gaussian mean). Fix $m \geq 1$, $d \geq 1$, $\varepsilon_2 > \varepsilon_1 > 0$, and $\delta > 0$. Suppose $\boldsymbol{\mu} \in \mathbb{R}^d$ is a hidden mean vector and we draw $m$ samples $\mathbf{x}_1, \ldots, \mathbf{x}_m \sim N(\boldsymbol{\mu}, \mathbf{I}_d)$. An algorithm ALG is said to be a $(\varepsilon_1, \varepsilon_2, \delta)$-tolerant isotropic Gaussian mean tester if it satisfies the following two conditions:

1. If $\|\boldsymbol{\mu}\|_2 \leq \varepsilon_1$, then ALG should *Accept* with probability at least $1 - \delta$

2. If $\|\boldsymbol{\mu}\|_2 \geq \varepsilon_2$, then ALG should *Reject* with probability at least $1 - \delta$.

ALG is allowed to decide arbitrarily when $\varepsilon_1 < \|\boldsymbol{\mu}\|_2 < \varepsilon_2$.

It is known that the test statistic $y_n = \left\| \frac{1}{\sqrt{n}} \sum_{i=1}^{n} \mathbf{x}_i \right\|_2^2$ can be used for *non-tolerant* isotropic Gaussian mean testing with an appropriate threshold; see Appendix C of (Diakonikolas et al., 2017). With the following lemma we show that $y_n$ can also be used for *tolerant* isotropic Gaussian mean testing.

**Lemma B.2.** *Fix $m \geq 1$, $d \geq 1$, $\varepsilon_2 > \varepsilon_1 > 0$, and $\delta > 0$. Suppose $\boldsymbol{\mu} \in \mathbb{R}^d$ is a hidden mean vector and we draw $m$ i.i.d. samples $\mathbf{x}_1, \ldots, \mathbf{x}_m \sim N(\boldsymbol{\mu}, \mathbf{I}_d)$. When $d \geq \left( \frac{16\varepsilon_2^2}{\varepsilon_2^2 - \varepsilon_1^2} \right)^2$ and $m \in \mathcal{O}\left( \frac{\sqrt{d}}{\varepsilon_2^2 - \varepsilon_1^2} \log\left(\frac{1}{\delta}\right) \right)$, TOLERANTIGMT (Algorithm 2) is a $(\varepsilon_1, \varepsilon_2, \delta)$-tolerant isotropic Gaussian mean tester.*

---

**Algorithm 2** The TOLERANTIGMT algorithm.

1: **Input**: $\varepsilon_2 > \varepsilon_1 > 0$, $\delta \in (0,1)$, $m$ i.i.d. samples of $N(\boldsymbol{\mu}, \mathbf{I}_d)$, where $\boldsymbol{\mu} \in \mathbb{R}^d$
2: **Output**: Fail (too little samples), Accept ($\|\boldsymbol{\mu}\|_2 \leq \varepsilon_1$), or Reject ($\|\boldsymbol{\mu}\|_2 \geq \varepsilon_2$).
3: Define sample batch size $n = \lceil \frac{16\sqrt{d}}{\varepsilon_2^2 - \varepsilon_1^2} \rceil$
4: Define number of rounds $r = \lceil \log(\frac{12}{\delta}) \rceil$ if $\lceil \log(\frac{12}{\delta}) \rceil$ is odd, otherwise define $r = 1 + \lceil \log(\frac{12}{\delta}) \rceil$
5: Define testing threshold $\tau = d + \frac{n(\varepsilon_1^2 + \varepsilon_2^2)}{2}$
6: **if** $m < nr$ **then**
7:     **return** Fail
8: **else**
9:     **for** $i \in \{1, \ldots, r\}$ **do**
10:         Use an unused batch of $n$ i.i.d. samples $\mathbf{x}_1^{(i)}, \ldots, \mathbf{x}_n^{(i)} \sim N(\boldsymbol{\mu}, \mathbf{I}_d)$
11:         Compute test statistic $y_n^{(i)} = \left\| \frac{1}{\sqrt{n}} \sum_{i=1}^n \mathbf{x}_i^{(i)} \right\|_2^2$ for the $i^{th}$ test
12:         Define $i^{th}$ outcome $\mathrm{R}^{(i)}$ as Accept if $y_n^{(i)} \leq \tau$, and Reject otherwise
13:     **end for**
14:     **return** majority$(\mathrm{R}^{(1)}, \ldots, \mathrm{R}^{(r)})$
15: **end if**

---

*Proof.* The total number of samples $m$ required is $nr \in \mathcal{O}\left( \frac{\sqrt{d}}{\varepsilon_2^2 - \varepsilon_1^2} \log\left(\frac{1}{\delta}\right) \right)$ since TOLERANTIGMT uses $n = \frac{16\sqrt{d}}{\varepsilon_2^2 - \varepsilon_1^2}$ i.i.d. samples in each of the $r \in \mathcal{O}(\log(\frac{1}{\delta}))$ rounds.

For correctness, we will prove that each round $i \in \{1, \ldots, r\}$ succeeds with probability at least $2/3$. Then, by Chernoff bound, the majority outcome out of $r \geq \log(\frac{12}{\delta})$ independent tests will be correct with probability at least $1 - \delta$.

Now, fix an arbitrary round $i \in \{1, \ldots, r\}$. TOLERANTIGMT uses $n = \frac{16\sqrt{d}}{\varepsilon_2^2 - \varepsilon_1^2} \geq 1$ i.i.d. samples to form a statistic $y_n^{(i)}$ and tests against the threshold $\tau = d + \frac{n(\varepsilon_1^2 + \varepsilon_2^2)}{2}$. From Lemma A.14 (first item), we know that $y_n^{(i)} \sim \chi_d^2(\lambda)$ is a non-central chi-square random variable with $\lambda = n\|\boldsymbol{\mu}\|_2^2$. Let us define $t = \frac{n(\varepsilon_2^2 - \varepsilon_1^2)}{2} > 0$. Observe that we can rewrite the testing threshold $\tau$ in two different ways: $\tau = d + \frac{n(\varepsilon_1^2 + \varepsilon_2^2)}{2} = d + n\varepsilon_1^2 + t = d + n\varepsilon_2^2 - t$.

**Case 1**: $\|\boldsymbol{\mu}\|_2 \leq \varepsilon_1$

In this case, we have $\lambda = n\|\boldsymbol{\mu}\|_2^2 \leq n\varepsilon_1^2$ and $\tau = d + n\varepsilon_1^2 + t$. So,

$$
\begin{aligned}
\Pr(y_n^{(i)} > \tau) &= \Pr(y_n^{(i)} > d + n\varepsilon_1^2 + t) && \text{(since } \tau = d + n\varepsilon_1^2 + t) \\
&\leq \Pr(y_n^{(i)} > d + \lambda + t) && \text{(since } \lambda \leq n\varepsilon_1^2) \\
&\leq \exp\left( -\frac{dt^2}{4(d + 2\lambda)(d + 2\lambda + t)} \right) && \text{(apply Lemma A.14 (second item) with } t > 0) \\
&\leq \exp\left( -\frac{dt^2}{4(d + 2n\varepsilon_1^2)(d + 2n\varepsilon_1^2 + t)} \right) && \text{(since } \lambda \leq n\varepsilon_1^2) \\
&\leq \exp\left( -\frac{dn^2(\varepsilon_2^2 - \varepsilon_1^2)^2}{16(d + 2n\varepsilon_1^2)(d + 2n\varepsilon_2^2)} \right) && \text{(since } t = \frac{n(\varepsilon_2^2 - \varepsilon_1^2)}{2} \leq 2n(\varepsilon_2^2 - \varepsilon_1^2)) \\
&= \exp\left( -\frac{16^2 d^2}{16(d + 2n\varepsilon_1^2)(d + 2n\varepsilon_2^2)} \right) && \text{(since } n = \frac{16\sqrt{d}}{\varepsilon_2^2 - \varepsilon_1^2}) \\
&= \exp\left( -\frac{16}{\left(1 + \frac{2n\varepsilon_1^2}{d}\right)\left(1 + \frac{2n\varepsilon_2^2}{d}\right)} \right) && \text{(dividing both numerator and denominator by } 16d^2) \\
&= \exp\left( -\frac{16}{\left(1 + \frac{32\varepsilon_1^2}{\sqrt{d}(\varepsilon_2^2 - \varepsilon_1^2)}\right)\left(1 + \frac{32\varepsilon_2^2}{\sqrt{d}(\varepsilon_2^2 - \varepsilon_1^2)}\right)} \right) && \text{(since } n = \frac{16\sqrt{d}}{\varepsilon_2^2 - \varepsilon_1^2})
\end{aligned}
$$

$$= \exp\left(-\frac{16}{(1+2)(1+2)}\right) \qquad \text{(since } d \geq \left(\frac{16\varepsilon_2^2}{\varepsilon_2^2-\varepsilon_1^2}\right)^2 \geq \left(\frac{16\varepsilon_1^2}{\varepsilon_2^2-\varepsilon_1^2}\right)^2\text{)}$$

$$= \exp\left(-\frac{16}{9}\right) < \frac{1}{3}$$

Thus, when $\|\boldsymbol{\mu}\|_2 \leq \varepsilon_1$, we have $\Pr(y_n^{(i)} \leq \tau) \geq 2/3$ and the $i^{th}$ test outcome will be correctly an Accept with probability at least $2/3$.

**Case 2**: $\|\boldsymbol{\mu}\|_2 \geq \varepsilon_2$

In this case, we have $\lambda = n\|\boldsymbol{\mu}\|_2^2 \geq n\varepsilon_2^2 > n\varepsilon_1^2$ and $\tau = d + n\varepsilon_2^2 - t$. We first observe the following inequalities:

- Since $n \geq 1$, $d \geq 1$, $\lambda \geq n\varepsilon_2^2$, and $\varepsilon_2 > \varepsilon_1 > 0$, we see that

$$\left(2 - \frac{n\varepsilon_1^2}{\lambda} - \frac{n\varepsilon_2^2}{\lambda}\right)^2 \geq \left(1 - \frac{\varepsilon_1^2}{\varepsilon_2^2}\right)^2 \quad \text{and} \quad \left(\frac{d}{\lambda} + 2\right)^2 \leq \left(\frac{d}{n\varepsilon_2^2} + 2\right)^2 \tag{6}$$

- Since $n = \frac{16\sqrt{d}}{\varepsilon_2^2-\varepsilon_1^2} \geq 1$ and $d \geq \left(\frac{16\varepsilon_2^2}{\varepsilon_2^2-\varepsilon_1^2}\right)^2 \geq 1$, we see that

$$\left(1 + \frac{2n\varepsilon_2^2}{d}\right)^2 \leq 3^2 \tag{7}$$

So,

$$
\begin{aligned}
\Pr(y_n^{(i)} < \tau) &= \Pr(y_n^{(i)} < d + n\varepsilon_2^2 - t) && \text{(since } \tau = d + n\varepsilon_2^2 - t\text{)} \\
&= \Pr(y_n^{(i)} < d + \lambda - (\lambda + t - n\varepsilon_2^2)) && \text{(Rewriting)} \\
&\leq \exp\left(-\frac{d(\lambda + t - n\varepsilon_2^2)^2}{4(d+2\lambda)^2}\right) && \text{(apply Lemma A.14 (third item) with } 0 < \lambda + t - n\varepsilon_2^2 < d + \lambda) \\
&= \exp\left(-\frac{d\left(\lambda - \frac{n}{2}\varepsilon_1^2 - \frac{n}{2}\varepsilon_2^2\right)^2}{4(d+2\lambda)^2}\right) && \text{(since } t = \frac{n(\varepsilon_2^2-\varepsilon_1^2)}{2}\text{)} \\
&= \exp\left(-\frac{d\left(2 - \frac{n\varepsilon_1^2}{\lambda} - \frac{n\varepsilon_2^2}{\lambda}\right)^2}{16\left(\frac{d}{\lambda}+2\right)^2}\right) && \text{(Pulling out the factor of } \frac{\lambda}{2} \text{ from numerator)} \\
&\leq \exp\left(-\frac{d\left(1 - \frac{\varepsilon_1^2}{\varepsilon_2^2}\right)^2}{16\left(\frac{d}{n\varepsilon_2^2}+2\right)^2}\right) && \text{(by Equation (6))} \\
&\leq \exp\left(-\frac{n^2\left(\varepsilon_2^2-\varepsilon_1^2\right)^2}{16d\left(1+\frac{n\varepsilon_2^2}{d}\right)^2}\right) && \text{(Pulling out factors of } n, d, \text{ and } \varepsilon_2^2\text{)} \\
&= \exp\left(-\frac{16}{\left(1+\frac{n\varepsilon_2^2}{d}\right)^2}\right) && \text{(since } n = \frac{16\sqrt{d}}{\varepsilon_2^2-\varepsilon_1^2}\text{)} \\
&= \exp\left(-\frac{16}{3^2}\right) = \exp\left(-\frac{16}{9}\right) < \frac{1}{3} && \text{(by Equation (7))}
\end{aligned}
$$

Thus, when $\|\boldsymbol{\mu}\|_2 \geq \varepsilon_2$, we have $\Pr(y_n^{(i)} \geq \tau) \geq 2/3$ and the $i^{th}$ test outcome will be correctly a Reject with probability at least $2/3$. $\qquad\square$

We are now ready to state the main theorem below.

**Lemma B.3** (Tolerant mean tester). *Given $\varepsilon_2 > \varepsilon_1 > 0$, $\delta \in (0,1)$, and $d \geq \left(\frac{16\varepsilon_2^2}{\varepsilon_2^2 - \varepsilon_1^2}\right)^2$, there is a tolerant tester that uses*
$\mathcal{O}\left(\frac{\sqrt{d}}{\varepsilon_2^2 - \varepsilon_1^2} \log\left(\frac{1}{\delta}\right)\right)$ *i.i.d. samples from $N(\boldsymbol{\mu}, \mathbf{I}_d)$ and satisfies both conditions below:*
*1. If $\|\boldsymbol{\mu}\|_2 \leq \varepsilon_1$, then the tester outputs* Accept,
*2. If $\|\boldsymbol{\mu}\|_2 \geq \varepsilon_2$, then the tester outputs* Reject,
*each with success probability at least $1 - \delta$.*

*Proof.* Use the guarantee of Lemma B.2 on TOLERANTIGMT (Algorithm 2) with parameters $\varepsilon_1 = \varepsilon$ and $\varepsilon_2 = 2\varepsilon$. $\square$

### B.1.2. TOLERANT TESTING FOR COVARIANCE MATRIX

We now give the definition of a tolerant tester for the unknown covariance matrix being equal to identity.

**Definition B.4** (Tolerant testing of zero-mean Gaussian covariance matrix). Fix $m \geq 1$, $d \geq 1$, $\varepsilon_2 > \varepsilon_1 > 0$, and $\delta > 0$. Suppose $\boldsymbol{\Sigma} \in \mathbb{R}^{d \times d}$ is a hidden full rank covariance matrix and we draw $m$ samples $\mathbf{x}_1, \ldots, \mathbf{x}_m \sim N(\mathbf{0}, \boldsymbol{\Sigma})$. An algorithm ALG is said to be a $(\varepsilon_1, \varepsilon_2, \delta)$-tolerant zero-mean Gaussian covariance tester if it satisfies the following two conditions:

1. If $\|\boldsymbol{\Sigma} - \mathbf{I}_d\|_F \leq \varepsilon_1$, then ALG should *Accept* with probability at least $1 - \delta$

2. If $\|\boldsymbol{\Sigma} - \mathbf{I}_d\|_F \geq \varepsilon_2$, then ALG should *Reject* with probability at least $1 - \delta$.

ALG is allowed to decide arbitrarily when $\varepsilon_1 < \|\boldsymbol{\Sigma} - \mathbf{I}_d\|_2 < \varepsilon_2$.

**Definition B.5** (Test statistic $\mathtt{T}_n$). Let $x_1, \ldots, x_n$ be $n$ i.i.d. samples from $\sim N(\mathbf{0}, \boldsymbol{\Sigma})$ for an unknown $\boldsymbol{\Sigma} \in \mathbb{R}^{d \times d}$. For $i \neq j$, we define $h(x_i, x_j) = (x_i^\top x_j)^2 - (x_i^\top x_i + x_j^\top x_j) + d$. Then, we define $\mathtt{T}_n$ as

$$\mathtt{T}_n = \frac{2}{n(n-1)} \sum_{1 \leq i < j \leq n} h(x_i, x_j)$$

It is known that the test statistic $\mathtt{T}_n$ (Definition B.5) can be used for *non-tolerant* zero-mean Gaussian covariance testing with an appropriate threshold; see (Cai & Ma, 2013). With the following lemma, we show that $\mathtt{T}_n$ can also be used for *tolerant* zero-mean Gaussian covariance testing.

---

**Algorithm 3** TOLERANTZMGCT.

---

1: **Input**: $\varepsilon_2 > \varepsilon_1 > 0$, $\delta \in (0,1)$, $m$ i.i.d. samples of $N(\mathbf{0}, \boldsymbol{\Sigma})$, where $\boldsymbol{\Sigma} \in \mathbb{R}^{d \times d}$
2: **Output**: Fail (too little samples), Accept ($\|\boldsymbol{\Sigma} - \mathbf{I}_d\|_F^2 \leq \varepsilon_1^2$), or Reject ($\|\boldsymbol{\Sigma} - \mathbf{I}_d\|_F^2 \geq \varepsilon_2^2$)
3: Define sample batch size $n = \left\lceil 3200 \cdot d \cdot \max\left\{\frac{1}{\varepsilon_1^2}, \left(\frac{\varepsilon_1^2}{\varepsilon_2^2 - \varepsilon_1^2}\right)^2, 2\left(\frac{\varepsilon_2}{\varepsilon_2^2 - \varepsilon_1^2}\right)^2\right\}\right\rceil$
4: Define number of rounds $r = \left\lceil\log(\frac{12}{\delta})\right\rceil$ if $\left\lceil\log(\frac{12}{\delta})\right\rceil$ is odd, otherwise define $r = 1 + \left\lceil\log(\frac{12}{\delta})\right\rceil$
5: Define testing threshold $\tau = \frac{\varepsilon_2^2 + \varepsilon_1^2}{2}$
6: **if** $m < nr$ **then**
7:     **return** Fail
8: **else**
9:     **for** $i \in \{1, \ldots, r\}$ **do**
10:         Use an unused batch of $n$ i.i.d. samples $\mathbf{x}_1^{(i)}, \ldots, \mathbf{x}_n^{(i)} \sim N(\mathbf{0}, \boldsymbol{\Sigma})$
11:         Compute test statistic $T_n^{(i)}$ according to Definition B.5 for the $i^{th}$ test
12:         Define $i^{th}$ outcome $R^{(i)}$ as Accept if $T_n^{(i)} \leq \tau$, and Reject otherwise
13:     **end for**
14:     **return** majority$(R^{(1)}, \ldots, R^{(r)})$
15: **end if**

---

**Lemma B.6.** *Fix $m \geq 1$, $d \geq 1$, $\varepsilon_2 > \varepsilon_1 > 0$, and $\delta > 0$. Suppose $\mathbf{\Sigma} \in \mathbb{R}^{d \times d}$ is a hidden full rank covariance matrix and we draw $m$ i.i.d. samples $\mathbf{x}_1, \ldots, \mathbf{x}_m \sim N(\mathbf{0}, \mathbf{\Sigma})$. When $d \geq \varepsilon_2^2$ and*

$$m \geq \mathcal{O}\left( d \cdot \max\left\{ \frac{1}{\varepsilon_1^2}, \left(\frac{\varepsilon_1^2}{\varepsilon_2^2 - \varepsilon_1^2}\right)^2, \left(\frac{\varepsilon_2}{\varepsilon_2^2 - \varepsilon_1^2}\right)^2 \right\} \cdot \log\left(\frac{1}{\delta}\right) \right) ,$$

TOLERANTZMGCT *(Algorithm 3) is a $(\varepsilon_1, \varepsilon_2, \delta)$-tolerant zero-mean Gaussian covariance tester.*

To prove Lemma B.6, we first state the expectation and variance of $T_n$ known from (Cai & Ma, 2013), and give an upper bound on the variance that will be useful for subsequent analysis.

**Lemma B.7** ((Cai & Ma, 2013))**.** *For the test statistic $T_n$ defined in Definition B.5, we have $\mathbb{E}(T_n) = \|\mathbf{\Sigma} - \mathbf{I}_d\|_F^2$ and $\sigma^2(T_n) = \frac{4}{n(n-1)}\left[\text{Tr}^2(\mathbf{\Sigma}^2) + \text{Tr}(\mathbf{\Sigma}^4)\right] + \frac{8}{n}\text{Tr}(\mathbf{\Sigma}^2(\mathbf{\Sigma} - \mathbf{I}_d)^2)$.*

**Lemma B.8.** *Fix $d, n \geq 1$, $\mathbf{\Sigma} \in \mathbb{R}^{d \times d}$, and $b \geq 0$. If $\|\mathbf{\Sigma} - \mathbf{I}_d\|_F^2 = \frac{b^2 d}{n}$, then $\|\mathbf{\Sigma}\|_F^2 \leq d \cdot \left(1 + \frac{b}{\sqrt{n}}\right)^2$.*

*Proof.* Since the matrices can be treated as vectors in $\mathbb{R}^{d^2}$ and then the Frobenius norm corresponds to the $\ell_2$ norm, we see that

$$\|\mathbf{\Sigma}\|_F \leq \|\mathbf{\Sigma} - \mathbf{I}_d\|_F + \|\mathbf{I}_d\|_F \qquad \text{(Triangle inequality)}$$

$$= b \cdot \sqrt{\frac{d}{n}} + \sqrt{d} \qquad \text{(Since } \|\mathbf{\Sigma} - \mathbf{I}_d\|_F^2 = \frac{b^2 d}{n} \text{ and } \|\mathbf{I}_d\|_F^2 = d)$$

$$= \sqrt{d}\left(1 + \frac{b}{\sqrt{n}}\right)$$

Therefore, $\|\mathbf{\Sigma}\|_F^2 \leq d \cdot \left(1 + \frac{b}{\sqrt{n}}\right)^2$ as desired. $\qquad \square$

**Lemma B.9.** *Fix $d \geq 1$, $n \geq 2$, $\mathbf{\Sigma} \in \mathbb{R}^{d \times d}$, and $b \geq 0$. If $\|\mathbf{\Sigma} - \mathbf{I}_d\|_F^2 = \frac{b^2 d}{n}$, then for the test statistic $T_n$ defined in Definition B.5, we have*

$$\sigma^2(T_n) \leq \frac{64 d^2}{n^2} \cdot \left(1 + \frac{b^2}{n}\right) \cdot \left(1 + \frac{b^2}{n} + b^2\right)$$

*Proof.* We begin by observing two simple upper bounds for $\text{Tr}(\mathbf{\Sigma}^4)$ and $\text{Tr}(\mathbf{\Sigma}^2(\mathbf{\Sigma} - \mathbf{I}_d)^2)$.

$$\text{Tr}(\mathbf{\Sigma}^4) = \|\mathbf{\Sigma}^2\|_F^2 \leq \|\mathbf{\Sigma}\|_F^2 \cdot \|\mathbf{\Sigma}\|_F^2 = \|\mathbf{\Sigma}\|_F^4 = \text{Tr}^2(\mathbf{\Sigma}^2) \tag{8}$$

Since $\mathbf{\Sigma}(\mathbf{\Sigma} - \mathbf{I}_d) = \mathbf{\Sigma}^2 - \mathbf{\Sigma} = (\mathbf{\Sigma} - \mathbf{I}_d)\mathbf{\Sigma}$, i.e. $\mathbf{\Sigma}$ and $\mathbf{\Sigma} - \mathbf{I}_d$ commute, we have

$$\text{Tr}(\mathbf{\Sigma}^2(\mathbf{\Sigma} - \mathbf{I}_d)^2) = \text{Tr}((\mathbf{\Sigma}(\mathbf{\Sigma} - \mathbf{I}_d))^2) = \|\mathbf{\Sigma}(\mathbf{\Sigma} - \mathbf{I}_d)\|_F^2 \leq \|\mathbf{\Sigma}\|_F^2 \cdot \|\mathbf{\Sigma} - \mathbf{I}_d\|_F^2 = \text{Tr}(\mathbf{\Sigma}^2) \cdot \text{Tr}((\mathbf{\Sigma} - \mathbf{I}_d)^2) \tag{9}$$

$$\mathbf{\Sigma}^2(T_n)$$

$$= \frac{4}{n(n-1)}\left[\text{Tr}^2(\mathbf{\Sigma}^2) + \text{Tr}(\mathbf{\Sigma}^4)\right] + \frac{8}{n}\text{Tr}(\mathbf{\Sigma}^2(\mathbf{\Sigma} - \mathbf{I}_d)^2) \qquad \text{(By Lemma B.7)}$$

$$\leq \frac{8}{n(n-1)}\left[\text{Tr}^2(\mathbf{\Sigma}^2) + (n-1) \cdot \text{Tr}(\mathbf{\Sigma}^2(\mathbf{\Sigma} - \mathbf{I}_d)^2)\right] \qquad \text{(By Equation (8))}$$

$$\leq \frac{8}{n(n-1)}\left[\text{Tr}^2(\mathbf{\Sigma}^2) + (n-1) \cdot \text{Tr}(\mathbf{\Sigma}^2) \cdot \text{Tr}((\mathbf{\Sigma} - \mathbf{I}_d)^2)\right] \qquad \text{(By Equation (9))}$$

$$= \frac{8}{n(n-1)} \cdot \text{Tr}(\mathbf{\Sigma}^2) \cdot \left[\text{Tr}(\mathbf{\Sigma}^2) + (n-1) \cdot \text{Tr}((\mathbf{\Sigma} - \mathbf{I}_d)^2)\right]$$

$$\leq \frac{8}{n(n-1)} \cdot \mathrm{Tr}(\boldsymbol{\Sigma}^2) \cdot \left[\mathrm{Tr}(\boldsymbol{\Sigma}^2) + n \cdot \mathrm{Tr}((\boldsymbol{\Sigma} - \mathbf{I}_d)^2)\right] \qquad \text{(Since } \mathrm{Tr}((\boldsymbol{\Sigma} - \mathbf{I}_d)^2) \geq 0)$$

$$\leq \frac{8}{n(n-1)} \cdot d \cdot \left(1 + \frac{b}{\sqrt{n}}\right)^2 \cdot \left(d \cdot \left(1 + \frac{b}{\sqrt{n}}\right)^2 + n \cdot \mathrm{Tr}((\boldsymbol{\Sigma} - \mathbf{I}_d)^2)\right)$$
$$\text{(Since } \mathrm{Tr}(\boldsymbol{\Sigma}^2) = \|\boldsymbol{\Sigma}\|_F^2 \text{ and by Lemma B.8)}$$

$$= \frac{8}{n(n-1)} \cdot d \cdot \left(1 + \frac{b}{\sqrt{n}}\right)^2 \cdot \left(d \cdot \left(1 + \frac{b}{\sqrt{n}}\right)^2 + b^2 \cdot d\right) \qquad \text{(Since } \mathrm{Tr}((\boldsymbol{\Sigma} - \mathbf{I}_d)^2) = \|\boldsymbol{\Sigma} - \mathbf{I}_d\|_F^2 = \frac{b^2 d}{n})$$

$$= \frac{8d^2}{n(n-1)} \cdot \left(1 + \frac{b}{\sqrt{n}}\right)^2 \cdot \left(\left(1 + \frac{b}{\sqrt{n}}\right)^2 + b^2\right)$$

$$\leq \frac{16d^2}{n^2} \cdot \left(1 + \frac{b}{\sqrt{n}}\right)^2 \cdot \left(\left(1 + \frac{b}{\sqrt{n}}\right)^2 + b^2\right) \qquad \text{(Since } n \geq 2)$$

$$\leq \frac{64d^2}{n^2} \cdot \left(1 + \frac{b^2}{n}\right) \cdot \left(1 + \frac{b^2}{n} + b^2\right) \qquad \text{(Since } (a+b)^2 \leq 2a^2 + 2b^2)$$

$$\square$$

*Proof of Lemma B.6.* Let us define $\Delta_{\varepsilon_1,\varepsilon_2} = \max\left\{\frac{1}{\varepsilon_1^2}, \left(\frac{\varepsilon_1^2}{\varepsilon_2^2 - \varepsilon_1^2}\right)^2, 2\left(\frac{\varepsilon_2}{\varepsilon_2^2 - \varepsilon_1^2}\right)^2\right\} > 0$ and suppose $\|\boldsymbol{\Sigma} - \mathbf{I}_d\|_F^2 = \frac{b^2 d}{n}$ for some $b \geq 0$.

The total number of samples $m$ required is $nr \in \mathcal{O}\left(d \cdot \Delta_{\varepsilon_1,\varepsilon_2} \cdot \log\left(\frac{1}{\delta}\right)\right)$ since TOLERANTZMGCT uses $n = 3200 \cdot d \cdot \Delta_{\varepsilon_1,\varepsilon_2}$ i.i.d. samples in each of the $r \in \mathcal{O}(\log(\frac{1}{\delta}))$ rounds.

For correctness, we will prove that each round $i \in \{1, \ldots, r\}$ succeeds with probability at least $2/3$. Then, by Chernoff bound, the majority outcome out of $r \geq \log(\frac{12}{\delta})$ independent tests will be correct with probability at least $1 - \delta$.

Now, fix an arbitrary round $i \in \{1, \ldots, r\}$. TOLERANTZMGCT uses $n = 3200 \cdot d \cdot \Delta_{\varepsilon_1,\varepsilon_2}$ i.i.d. samples to form a statistic $T_n^{(i)}$ (Definition B.5) and tests against the threshold $\tau = \frac{\varepsilon_2^2 + \varepsilon_1^2}{4}$.

**Case 1**: $\|\boldsymbol{\Sigma} - \mathbf{I}_d\|_F^2 \leq \varepsilon_1^2$

We see that

$$b^2 = \frac{n}{d} \cdot \|\boldsymbol{\Sigma} - \mathbf{I}_d\|_F^2 \qquad \text{(Since } \|\boldsymbol{\Sigma} - \mathbf{I}_d\|_F^2 = \frac{b^2 d}{n})$$
$$= 3200 \cdot \Delta_{\varepsilon_1,\varepsilon_2} \cdot \|\boldsymbol{\Sigma} - \mathbf{I}_d\|_F^2 \qquad \text{(Since } n = 3200 \cdot d \cdot \Delta_{\varepsilon_1,\varepsilon_2})$$
$$\leq 3200 \cdot \Delta_{\varepsilon_1,\varepsilon_2} \cdot \varepsilon_1^2 \qquad \text{(Since } \|\boldsymbol{\Sigma} - \mathbf{I}_d\|_F^2 \leq \varepsilon_1^2)$$

and

$$1 + \frac{b^2}{n} = 1 + \frac{\|\boldsymbol{\Sigma} - \mathbf{I}_d\|_F^2}{d} \qquad \text{(Since } \|\boldsymbol{\Sigma} - \mathbf{I}_d\|_F^2 = \frac{b^2 d}{n})$$
$$\leq 1 + \frac{\varepsilon_1^2}{d} \qquad \text{(Since } \|\boldsymbol{\Sigma} - \mathbf{I}_d\|_F^2 \leq \varepsilon_1^2)$$
$$\leq 2 \qquad \text{(Since } d \geq \varepsilon_2^2 > \varepsilon_1^2)$$

So,

$$\sigma^2(T_n) \leq \frac{64d^2}{n^2} \cdot \left(1 + \frac{b^2}{n}\right) \cdot \left(1 + \frac{b^2}{n} + b^2\right) \qquad \text{(By Lemma B.9)}$$

$$\leq \frac{64d^2}{n^2} \cdot 2 \cdot \left(2 + 3200 \cdot \Delta_{\varepsilon_1,\varepsilon_2} \cdot \varepsilon_1^2\right) \qquad \text{(From above)}$$

$$= \frac{64 \cdot 2}{3200^2} \cdot \frac{1}{\Delta_{\varepsilon_1,\varepsilon_2}^2} \cdot \left(2 + 3200 \cdot \Delta_{\varepsilon_1,\varepsilon_2} \cdot \varepsilon_1^2\right) \qquad \text{(Since } n = 3200 \cdot d \cdot \Delta_{\varepsilon_1,\varepsilon_2})$$

$$\leq \frac{64 \cdot 2}{3200^2} \cdot \frac{1}{\Delta_{\varepsilon_1,\varepsilon_2}^2} \cdot 3202 \cdot \Delta_{\varepsilon_1,\varepsilon_2} \cdot \varepsilon_1^2 \qquad \text{(Since } \Delta_{\varepsilon_1,\varepsilon_2}\varepsilon_1^2 \geq 1)$$

$$\leq \frac{64 \cdot 2 \cdot 3202}{3200^2} \cdot (\varepsilon_2^2 - \varepsilon_1^2)^2 \qquad \text{(Since } \left(\frac{\varepsilon_1^2}{\varepsilon_2^2 - \varepsilon_1^2}\right)^2 \leq \Delta_{\varepsilon_1,\varepsilon_2})$$

Chebyshev's inequality then tells us that

$$\Pr\left(\mathtt{T}_n > \tau\right) = \Pr\left(\mathtt{T}_n > \varepsilon_1^2 + \frac{\varepsilon_2^2 - \varepsilon_1^2}{2}\right) \qquad \text{(Since } \tau = \frac{\varepsilon_2^2 + \varepsilon_1^2}{2} = \varepsilon_1^2 + \frac{\varepsilon_2^2 - \varepsilon_1^2}{2})$$

$$\leq \Pr\left(\mathtt{T}_n > \|\boldsymbol{\Sigma} - \mathbf{I}_d\|_F^2 + \frac{\varepsilon_2^2 - \varepsilon_1^2}{2}\right) \qquad \text{(Since } \|\boldsymbol{\Sigma} - \mathbf{I}_d\|_F^2 \leq \varepsilon_1^2)$$

$$= \Pr\left(\mathtt{T}_n > \mathbb{E}[\mathtt{T}_n] + \frac{\varepsilon_2^2 - \varepsilon_1^2}{2}\right) \qquad \text{(By Lemma B.7)}$$

$$\leq \Pr\left(|\mathtt{T}_n - \mathbb{E}[\mathtt{T}_n]| > \frac{\varepsilon_2^2 - \varepsilon_1^2}{2}\right) \qquad \text{(Adding absolute sign)}$$

$$\leq \sigma^2(\mathtt{T}_n) \cdot \left(\frac{2}{\varepsilon_2^2 - \varepsilon_1^2}\right)^2 \qquad \text{(Chebyshev's inequality)}$$

$$\leq \frac{64 \cdot 2 \cdot 3202}{3200^2} \cdot (\varepsilon_2^2 - \varepsilon_1^2)^2 \cdot \frac{4}{(\varepsilon_2^2 - \varepsilon_1^2)^2} \qquad \text{(From above)}$$

$$< \frac{1}{3}$$

Thus, when $\|\boldsymbol{\Sigma} - \mathbf{I}_d\|_F^2 \leq \varepsilon_1^2$, we have $\Pr\left(\mathtt{T}_n < \tau\right) \geq 2/3$ and the $i^{th}$ test outcome will be correctly an Accept with probability at least $2/3$.

**Case 2**: $\|\boldsymbol{\Sigma} - \mathbf{I}_d\|_F^2 \geq \varepsilon_2^2$

We can lower bound $b^2$ as follows:

$$b^2 = \frac{n}{d} \cdot \|\boldsymbol{\Sigma} - \mathbf{I}_d\|_F^2 \qquad \text{(Since } \|\boldsymbol{\Sigma} - \mathbf{I}_d\|_F^2 = \frac{b^2 d}{n})$$

$$= 3200 \cdot \Delta_{\varepsilon_1,\varepsilon_2} \cdot \|\boldsymbol{\Sigma} - \mathbf{I}_d\|_F^2 \qquad \text{(Since } n = 3200 \cdot d \cdot \Delta_{\varepsilon_1,\varepsilon_2})$$

$$\geq 3200 \cdot \Delta_{\varepsilon_1,\varepsilon_2} \cdot \varepsilon_2^2 \qquad \text{(Since } \|\boldsymbol{\Sigma} - \mathbf{I}_d\|_F^2 \geq \varepsilon_2^2)$$

Meanwhile, we can lower bound $n$ as follows:

$$n = 3200 \cdot d \cdot \Delta_{\varepsilon_1,\varepsilon_2} \qquad \text{(Since } n = 3200 \cdot d \cdot \Delta_{\varepsilon_1,\varepsilon_2})$$

$$\geq 3200 \cdot \varepsilon_2^2 \cdot \Delta_{\varepsilon_1,\varepsilon_2} \qquad \text{(Since } d \geq \varepsilon_2^2)$$

$$\geq \frac{3200 \cdot \varepsilon_2^2 \cdot \Delta_{\varepsilon_1,\varepsilon_2}}{\Delta_{\varepsilon_1,\varepsilon_2} \cdot \left(\frac{\varepsilon_2^2 - \varepsilon_1^2}{\varepsilon_2}\right)^2 - 1} \qquad \text{(Since } \Delta_{\varepsilon_1,\varepsilon_2} \geq 2\left(\frac{\varepsilon_2}{\varepsilon_2^2 - \varepsilon_1^2}\right)^2)$$

Using these lower bounds on $b^2$ and $n$ (which we color for convenience), we can conclude that $1 + \frac{b^2}{n} \leq \frac{b^2}{3200} \cdot \left(\frac{\varepsilon_2^2 - \varepsilon_1^2}{\varepsilon_2^2}\right)^2$ via the following two equivalences:

$$1 + \frac{b^2}{n} \leq \frac{b^2}{3200} \cdot \left(\frac{\varepsilon_2^2 - \varepsilon_1^2}{\varepsilon_2^2}\right)^2 \iff b^2 \geq \frac{n}{\frac{n}{3200} \cdot \left(\frac{\varepsilon_2^2 - \varepsilon_1^2}{\varepsilon_2^2}\right)^2 - 1}$$

and

$$3200 \cdot \Delta_{\varepsilon_1,\varepsilon_2} \cdot \varepsilon_2^2 \geq \frac{n}{\frac{n}{3200} \cdot \left(\frac{\varepsilon_2^2 - \varepsilon_1^2}{\varepsilon_2^2}\right)^2 - 1} \iff n \geq \frac{3200 \cdot \Delta_{\varepsilon_1,\varepsilon_2} \cdot \varepsilon_2^2}{\Delta_{\varepsilon_1,\varepsilon_2} \cdot \varepsilon_2^2 \cdot \left(\frac{\varepsilon_2^2 - \varepsilon_1^2}{\varepsilon_2^2}\right)^2 - 1} = \frac{3200 \cdot \varepsilon_2^2 \cdot \Delta_{\varepsilon_1,\varepsilon_2}}{\Delta_{\varepsilon_1,\varepsilon_2} \cdot \left(\frac{\varepsilon_2^2 - \varepsilon_1^2}{\varepsilon_2}\right)^2 - 1}$$

So,

$$\sigma^2(\mathtt{T}_n) \leq \frac{64d^2}{n^2} \cdot \left(1 + \frac{b^2}{n}\right) \cdot \left(1 + \frac{b^2}{n} + b^2\right) \qquad \text{(By Lemma B.9)}$$

$$\leq 64 \cdot 2 \cdot \frac{d^2}{n^2} \cdot \left(\frac{b^2}{3200} \cdot \left(\frac{\varepsilon_2^2 - \varepsilon_1^2}{\varepsilon_2^2}\right)^2\right) \cdot \left(\frac{b^2}{3200} \cdot \left(\frac{\varepsilon_2^2 - \varepsilon_1^2}{\varepsilon_2^2}\right)^2 + b^2\right) \quad \text{(Since } 1 + \frac{b^2}{n} \leq \frac{b^2}{3200} \cdot \left(\frac{\varepsilon_2^2 - \varepsilon_1^2}{\varepsilon_2^2}\right)^2\text{)}$$

$$= \frac{64 \cdot 2 \cdot 2}{3200} \cdot \left(\frac{\varepsilon_2^2 - \varepsilon_1^2}{\varepsilon_2^2}\right)^2 \cdot \frac{d^2}{n^2} \cdot b^4 \qquad \text{(Since } \frac{1}{3200}\left(\frac{\varepsilon_2^2 - \varepsilon_1^2}{\varepsilon_2^2}\right)^2 \leq 1\text{)}$$

$$= \frac{64 \cdot 2 \cdot 2}{3200} \cdot \left(\frac{\varepsilon_2^2 - \varepsilon_1^2}{\varepsilon_2^2}\right)^2 \cdot \|\mathbf{\Sigma} - \mathbf{I}_d\|_F^4 \qquad \text{(Since } \|\mathbf{\Sigma} - \mathbf{I}_d\|_F^2 = \frac{b^2 d}{n}\text{)}$$

Chebyshev's inequality then tells us that

$$\Pr\left(\mathtt{T}_n < \tau\right) = \Pr\left(\mathtt{T}_n < \varepsilon_2^2 \cdot \left(1 - \frac{\varepsilon_2^2 - \varepsilon_1^2}{2\varepsilon_2^2}\right)\right) \qquad \text{(Since } \tau = \frac{\varepsilon_2^2 + \varepsilon_1^2}{2} = \varepsilon_2^2 - \frac{\varepsilon_2^2 - \varepsilon_1^2}{2} = \varepsilon_2^2 \cdot \left(1 - \frac{\varepsilon_2^2 - \varepsilon_1^2}{2\varepsilon_2^2}\right)\text{)}$$

$$\leq \Pr\left(\mathtt{T}_n < \|\mathbf{\Sigma} - \mathbf{I}_d\|_F^2 \cdot \left(1 - \frac{\varepsilon_2^2 - \varepsilon_1^2}{2\varepsilon_2^2}\right)\right) \qquad \text{(Since } \|\mathbf{\Sigma} - \mathbf{I}_d\|_F^2 \geq \varepsilon_2^2\text{)}$$

$$= \Pr\left(\|\mathbf{\Sigma} - \mathbf{I}_d\|_F^2 - \mathtt{T}_n > \|\mathbf{\Sigma} - \mathbf{I}_d\|_F^2 \cdot \frac{\varepsilon_2^2 - \varepsilon_1^2}{2\varepsilon_2^2}\right) \qquad \text{(Rearranging)}$$

$$= \Pr\left(\mathbb{E}[\mathtt{T}_n] - \mathtt{T}_n > \|\mathbf{\Sigma} - \mathbf{I}_d\|_F^2 \cdot \frac{\varepsilon_2^2 - \varepsilon_1^2}{2\varepsilon_2^2}\right) \qquad \text{(By Lemma B.7)}$$

$$\leq \Pr\left(|\mathbb{E}[\mathtt{T}_n] - \mathtt{T}_n| > \|\mathbf{\Sigma} - \mathbf{I}_d\|_F^2 \cdot \frac{\varepsilon_2^2 - \varepsilon_1^2}{2\varepsilon_2^2}\right) \qquad \text{(Adding absolute sign)}$$

$$\leq \sigma^2(\mathtt{T}_n) \cdot \left(\frac{1}{\|\mathbf{\Sigma} - \mathbf{I}_d\|_F^2} \cdot \frac{2\varepsilon_2^2}{\varepsilon_2^2 - \varepsilon_1^2}\right)^2 \qquad \text{(Chebyshev's inequality)}$$

$$\leq \frac{64 \cdot 2 \cdot 2}{3200} \cdot \left(\frac{\varepsilon_2^2 - \varepsilon_1^2}{\varepsilon_2^2}\right)^2 \cdot \|\mathbf{\Sigma} - \mathbf{I}_d\|_F^4 \cdot \left(\frac{1}{\|\mathbf{\Sigma} - \mathbf{I}_d\|_F^2} \cdot \frac{2\varepsilon_2^2}{\varepsilon_2^2 - \varepsilon_1^2}\right)^2 \qquad \text{(From above)}$$

$$= \frac{64 \cdot 2 \cdot 2 \cdot 4}{3200}$$

$$< \frac{1}{3}$$

Thus, when $\|\mathbf{\Sigma} - \mathbf{I}_d\|_F^2 \geq \varepsilon_2^2$, we have $\Pr\left(\mathtt{T}_n > \tau\right) \geq 2/3$ and the $i^{th}$ test outcome will be correctly an Reject with probability at least $2/3$. $\qquad \square$

**Lemma B.10** (Tolerant covariance tester). *Given $\varepsilon_2 > \varepsilon_1 > 0$, $\delta \in (0,1)$, and $d \geq \varepsilon_2^2$, there is a tolerant tester that uses*
$\mathcal{O}\left(d \cdot \max\left\{\frac{1}{\varepsilon_1^2}, \left(\frac{\varepsilon_2^2}{\varepsilon_2^2 - \varepsilon_1^2}\right)^2, \left(\frac{\varepsilon_2}{\varepsilon_2^2 - \varepsilon_1^2}\right)^2\right\} \log\left(\frac{1}{\delta}\right)\right)$ *i.i.d. samples from $N(\mathbf{0}, \mathbf{\Sigma})$ and satisfies both conditions below:*
*1. If $\|\mathbf{\Sigma} - \mathbf{I}_d\|_F \leq \varepsilon_1$, then the tester outputs* Accept*,*
*2. If $\|\mathbf{\Sigma} - \mathbf{I}_d\|_F \geq \varepsilon_2$, then the tester outputs* Reject*,*
*each with success probability at least $1 - \delta$.*

*Proof.* Use the guarantee of Lemma B.6 on TOLERANTZMGCT (Algorithm 3) with parameters $\varepsilon_1^2 = \varepsilon^2$ and $\varepsilon_2^2 = 2\varepsilon^2$. $\qquad \square$

## C. Identity covariance setting

### C.1. Guarantees of APPROXL1

Here, we show that the guarantees of the APPROXL1 algorithm (Algorithm 4).

**Lemma C.1.** *Let $k$, $\alpha$, and $\zeta$ be the input parameters to the APPROXL1 algorithm (Algorithm 4). Given $m(k, \alpha, \delta')$ i.i.d. samples from $N(\boldsymbol{\mu}, \mathbf{I}_d)$, APPROXL1 succeeds with probability at least $1 - \delta$ and has the following properties:*

---

**Algorithm 4** The APPROXL1 algorithm.

1: **Input**: Block size $k \in [d]$, lower bound $\alpha > 0$, upper bound $\zeta > 2\alpha$, failure rate $\delta \in (0,1)$, and i.i.d. samples $\mathcal{S}$ from $N(\boldsymbol{\mu}, \mathbf{I}_d)$

2: **Output**: Fail or $\lambda \in \mathbb{R}$

3: Define $w = \lceil d/k \rceil$ and $\delta' = \frac{\delta}{w \cdot \lceil \log_2 \zeta/\alpha \rceil}$

4: Partition the index set $[d]$ into $w$ blocks:

$$\mathbf{B}_1 = \{1, \ldots, k\}, \mathbf{B}_2 = \{k+1, \ldots, 2k\}, \ldots, \mathbf{B}_w = \{k(w-1)+1, \ldots, d\}$$

5: **for** $j \in \{1, \ldots, w\}$ **do**

6:     Define $\mathcal{S}_j = \{\mathbf{x}_{\mathbf{B}_j} \in \mathbb{R}^{|\mathbf{B}_j|} : \mathbf{x} \in \mathcal{S}\}$ as the samples projected to $\mathbf{B}_j$     ▷ See Definition A.4

7:     Initialize $o_j =$ Fail

8:     **for** $i = 1, 2, \ldots, \lceil \log_2 \zeta/\alpha \rceil$ **do**

9:         Define $l_i = 2^{i-1} \cdot \alpha$

10:         Let Outcome be the output of the tolerant tester of Lemma 1.5 using sample set $\mathcal{S}_j$ with parameters
            $\varepsilon_1 = l_i$, $\varepsilon_2 = 2l_i$, and $\delta = \delta'$

11:         **if** Outcome is Accept **then**

12:             Set $o_j = l_i$ and **break**     ▷ Escape inner loop for block $j$

13:         **end if**

14:     **end for**

15: **end for**

16: **if** there exists a Fail amongst $\{o_1, \ldots, o_w\}$ **then**

17:     **return** Fail

18: **else**

19:     **return** $\lambda = 2 \sum_{j=1}^{w} \sqrt{|\mathbf{B}_j|} \cdot o_j$     ▷ $\lambda$ is an estimate for $\|\boldsymbol{\mu}\|_1$

20: **end if**

---

*1. If* APPROXL1 *outputs* Fail, *then* $\|\boldsymbol{\mu}\|_2 > \zeta/2$.

*2. If* APPROXL1 *outputs* $\lambda \in \mathbb{R}$, *then* $\|\boldsymbol{\mu}\|_1 \leq \lambda \leq 2\sqrt{k} \cdot (\lceil d/k \rceil \cdot \alpha + 2\|\boldsymbol{\mu}\|_1)$.

*Proof.* We begin by stating some properties of $o_1, \ldots, o_w$. Fix an arbitrary index $j \in \{1, \ldots, w\}$ and suppose $o_j$ is *not* a Fail, i.e. the tolerant tester of Lemma 1.5 outputs Accept for some $i^* \in \{1, 2, \ldots, \lceil \log_2 \zeta/\alpha \rceil\}$. Note that APPROXL1 sets $o_j = \ell_{i^*}$ and the tester outputs Reject for all smaller indices $i \in \{1, \ldots, i^* - 1\}$. Since the tester outputs Accept for $i^*$, we have that $\|\boldsymbol{\mu}_{\mathbf{B}_j}\|_2 \leq 2\ell_{i^*} = 2o_j$. Meanwhile, if $i^* > 1$, then $\|\boldsymbol{\mu}_{\mathbf{B}_j}\|_2 > \ell_{i^*-1} = \ell_{i^*}/2 = o_j/2$ since the tester outputs Reject for $i^* - 1$. Thus, we see that

- When $o_j$ is not Fail, we have $\|\boldsymbol{\mu}_{\mathbf{B}_j}\|_2 \leq 2o_j$.

- When $\|\boldsymbol{\mu}_{\mathbf{B}_j}\|_2 \leq 2\alpha$, we have $i^* = 1$ and $o_j = \ell_1 = \alpha$.

- When $\|\boldsymbol{\mu}_{\mathbf{B}_j}\|_2 > 2\alpha = 2\ell_1$, we have $i^* > 1$ and so $o_j < 2\|\boldsymbol{\mu}_{\mathbf{B}_j}\|_2$.

**Success probability.** Fix an arbitrary index $i \in \{1, 2, \ldots, \lceil \log_2 \zeta/\alpha \rceil\}$ with $\ell_i = 2^{i-1}\alpha$, where $\ell_i \leq \ell_1 = \alpha$ for any $i$. We invoke the tolerant tester with $\varepsilon_2 = 2\ell_i = 2\varepsilon_1$, so the $i^{th}$ invocation uses at most $n_{k,\varepsilon} \cdot r_\delta$ i.i.d. samples to succeed with probability at least $1 - \delta$; see Definition 2.1 and Algorithm 2. So, with $m(k, \alpha, \delta')$ samples, *any* call to the tolerant tester succeeds with probability at least $1 - \delta'$, where $\delta' = \frac{\delta}{w \cdot \lceil \log_2 \zeta/\alpha \rceil}$. By construction, there will be at most $w \cdot \lceil \log_2 \zeta/\alpha \rceil$ calls to the tolerant tester. Therefore, by union bound, *all* calls to the tolerant tester jointly succeed with probability at least $1 - \delta$.

**Property 1.** When APPROXL1 outputs Fail, there exists a Fail amongst $\{o_1, \ldots, o_w\}$. For any fixed index $j \in \{1, \ldots, w\}$, this can only happen when all calls to the tolerant tester outputs Reject. This means that $\|\boldsymbol{x}_{\mathbf{B}_j}\|_2 > \varepsilon_1 = \ell_i = 2^{i-1} \cdot \alpha$ for all $i \in \{1, 2, \ldots, \lceil \log_2 \zeta/\alpha \rceil\}$. In particular, this means that $\|\boldsymbol{x}_{\mathbf{B}_j}\|_2 > \zeta/2$.

**Property 2.** When APPROXL1 outputs $\lambda = 2 \sum_{j=1}^{w} \sqrt{|\mathbf{B}_j|} \cdot o_j \in \mathbb{R}$, we can lower bound $\lambda$ as follows:

$$
\begin{aligned}
\lambda &= 2 \sum_{j=1}^{w} \sqrt{|\mathbf{B}_j|} \cdot o_j \\
&\geq 2 \sum_{j=1}^{w} \sqrt{|\mathbf{B}_j|} \cdot \frac{\|\boldsymbol{\mu}_{\mathbf{B}_j}\|_2}{2} && \text{(since } \|\boldsymbol{\mu}_{\mathbf{B}_j}\|_2 \leq 2o_j) \\
&\geq \sum_{j=1}^{w} \|\boldsymbol{\mu}_{\mathbf{B}_j}\|_1 && \text{(since } \|\boldsymbol{\mu}_{\mathbf{B}_j}\|_1 \leq \sqrt{|\mathbf{B}_j|} \cdot \|\boldsymbol{\mu}_{\mathbf{B}_j}\|_2) \\
&= \|\boldsymbol{\mu}\|_1 && \text{(since } \sum_{j=1}^{w} \|\boldsymbol{\mu}_{\mathbf{B}_j}\|_1 = \|\boldsymbol{\mu}_{\mathbf{B}_j}\|_1)
\end{aligned}
$$

That is, $\lambda \geq \|\boldsymbol{\mu}\|_1$. Meanwhile, we can also upper bound $\lambda$ as follows:

$$
\begin{aligned}
\lambda &= 2 \sum_{j=1}^{w} \sqrt{|\mathbf{B}_j|} \cdot o_j \\
&\leq 2\sqrt{k} \sum_{j=1}^{w} o_j && \text{(since } |\mathbf{B}_j| \leq k) \\
&= 2\sqrt{k} \cdot \left( \sum_{\substack{j=1 \\ \|\boldsymbol{\mu}_{\mathbf{B}_j}\|_2 \leq 2\alpha}}^{w} o_j + \sum_{\substack{j=1 \\ \|\boldsymbol{\mu}_{\mathbf{B}_j}\|_2 > 2\alpha}}^{w} o_j \right) && \text{(partitioning the blocks based on } \|\boldsymbol{\mu}_{\mathbf{B}_j}\|_2 \text{ versus } 2\alpha) \\
&= 2\sqrt{k} \cdot \left( \sum_{\substack{j=1 \\ \|\boldsymbol{\mu}_{\mathbf{B}_j}\|_2 \leq 2\alpha}}^{w} \alpha + \sum_{\substack{j=1 \\ \|\boldsymbol{\mu}_{\mathbf{B}_j}\|_2 > 2\alpha}}^{w} o_j \right) && \text{(since } \|\boldsymbol{\mu}_{\mathbf{B}_j}\|_2 \leq 2\alpha \text{ implies } o_j = \alpha) \\
&\leq 2\sqrt{k} \cdot \left( \sum_{\substack{j=1 \\ \|\boldsymbol{\mu}_{\mathbf{B}_j}\|_2 \leq 2\alpha}}^{w} \alpha + \sum_{\substack{j=1 \\ \|\boldsymbol{\mu}_{\mathbf{B}_j}\|_2 > 2\alpha}}^{w} 2\|\boldsymbol{\mu}_{\mathbf{B}_j}\|_2 \right) && \text{(since } \|\boldsymbol{\mu}_{\mathbf{B}_j}\|_2 > 2\alpha \text{ implies } o_j \leq 2\|\boldsymbol{\mu}_{\mathbf{B}_j}\|_2) \\
&\leq 2\sqrt{k} \cdot \left( \sum_{\substack{j=1 \\ \|\boldsymbol{\mu}_{\mathbf{B}_j}\|_2 \leq 2\alpha}}^{w} \alpha + 2 \sum_{\substack{j=1 \\ \|\boldsymbol{\mu}_{\mathbf{B}_j}\|_2 > 2\alpha}}^{w} \|\boldsymbol{\mu}_{\mathbf{B}_j}\|_1 \right) && \text{(since } \|\boldsymbol{\mu}_{\mathbf{B}_j}\|_2 \leq \|\boldsymbol{\mu}_{\mathbf{B}_j}\|_1) \\
&\leq 2\sqrt{k} \cdot \left( \lceil d/k \rceil \cdot \alpha + 2 \sum_{\substack{j=1 \\ \|\boldsymbol{\mu}_{\mathbf{B}_j}\|_2 > 2\alpha}}^{w} \|\boldsymbol{\mu}_{\mathbf{B}_j}\|_1 \right) && \text{(since } |\{j \in [w] : \boldsymbol{\mu}_{\mathbf{B}_j}\|_2 \leq 2\alpha\}| \leq w) \\
&\leq 2\sqrt{k} \cdot \left( \lceil d/k \rceil \cdot \alpha + 2\|\boldsymbol{\mu}\|_1 \right) && \text{(since } \sum_{\substack{j=1 \\ \|\boldsymbol{\mu}_{\mathbf{B}_j}\|_2 > 2\alpha}}^{w} \|\boldsymbol{\mu}_{\mathbf{B}_j}\|_1 \leq \sum_{j=1}^{w} \|\boldsymbol{\mu}_{\mathbf{B}_j}\|_1 = \|\boldsymbol{\mu}_{\mathbf{B}_j}\|_1)
\end{aligned}
$$

That is, $\lambda \leq 2\sqrt{k} \cdot (\lceil d/k \rceil \cdot \alpha + 2\|\boldsymbol{\mu}\|_1)$. The property follows by putting together both bounds. $\square$

### C.2. Deferred derivation

Here, we show how to derive Equation (3) from Equation (2).

For any two vectors $\mathbf{a}, \mathbf{b} \in \mathbb{R}^d$, observe that $\|\mathbf{a} - \mathbf{b}\|_2^2 = \langle \mathbf{a} - \mathbf{b}, \mathbf{a} - \mathbf{b} \rangle = (\mathbf{a} - \mathbf{b})^{\top}(\mathbf{a} - \mathbf{b}) = \mathbf{a}^{\top}\mathbf{a} - 2\mathbf{a}^{\top}\mathbf{b} + \mathbf{b}^{\top}\mathbf{b}$, since

$\mathbf{a}^\top \mathbf{b} = \mathbf{b}^\top \mathbf{a}$ is just a number. So,

$$\frac{1}{n} \sum_{i=1}^{n} \|\mathbf{y}_i - \widehat{\boldsymbol{\mu}}\|_2^2 = \frac{1}{n} \sum_{i=1}^{n} \left( \mathbf{y}_i^\top \mathbf{y}_i - 2\mathbf{y}_i^\top \widehat{\boldsymbol{\mu}} + \widehat{\boldsymbol{\mu}}^\top \widehat{\boldsymbol{\mu}} \right)$$

$$\frac{1}{n} \sum_{i=1}^{n} \|\mathbf{y}_i - X\boldsymbol{\mu}\|_2^2 = \frac{1}{n} \sum_{i=1}^{n} \left( \mathbf{y}_i^\top \mathbf{y}_i - 2\mathbf{y}_i^\top \boldsymbol{\mu} + \boldsymbol{\mu}^\top \boldsymbol{\mu} \right)$$

Therefore,

$$
\begin{aligned}
\|\widehat{\boldsymbol{\mu}} - \boldsymbol{\mu}\|_2^2 &= \frac{1}{n} \sum_{i=1}^{n} \|\widehat{\boldsymbol{\mu}} - \boldsymbol{\mu}\|_2^2 \\
&= \frac{1}{n} \sum_{i=1}^{n} \left( \widehat{\boldsymbol{\mu}}^\top \widehat{\boldsymbol{\mu}} - 2\boldsymbol{\mu}^\top \widehat{\boldsymbol{\mu}} + \boldsymbol{\mu}^\top \boldsymbol{\mu} \right) \\
&\leq \frac{1}{n} \sum_{i=1}^{n} \left( 2\mathbf{y}_i^\top \widehat{\boldsymbol{\mu}} - 2\mathbf{y}_i^\top \boldsymbol{\mu} + \boldsymbol{\mu}^\top \boldsymbol{\mu} - 2\boldsymbol{\mu}^\top \widehat{\boldsymbol{\mu}} + \boldsymbol{\mu}^\top \boldsymbol{\mu} \right) \\
&\qquad\qquad \text{(Since Equation (3) tells us that } \frac{1}{n} \sum_{i=1}^{n} \|\mathbf{y}_i - \widehat{\boldsymbol{\mu}}\|_2^2 \leq \frac{1}{n} \sum_{i=1}^{n} \|\mathbf{y}_i - \boldsymbol{\mu}\|_2^2 ) \\
&= \frac{2}{n} \sum_{i=1}^{n} \left( (\boldsymbol{\mu} + \mathbf{g}_i)^\top (\widehat{\boldsymbol{\mu}} - \boldsymbol{\mu}) - \boldsymbol{\mu}^\top \widehat{\boldsymbol{\mu}} + \boldsymbol{\mu}^\top \boldsymbol{\mu} \right) && \text{(Since } \mathbf{y}_i = \boldsymbol{\mu} + \mathbf{g}_i ) \\
&= \frac{2}{n} \sum_{i=1}^{n} \left( \mathbf{g}_i^\top (\widehat{\boldsymbol{\mu}} - \boldsymbol{\mu}) \right) \\
&= \frac{2}{n} \sum_{i=1}^{n} \langle \mathbf{g}_i, \widehat{\boldsymbol{\mu}} - \boldsymbol{\mu} \rangle \\
&= \frac{2}{n} \langle \sum_{i=1}^{n} \mathbf{g}_i, \widehat{\boldsymbol{\mu}} - \boldsymbol{\mu} \rangle && \text{(Linearity of inner product)}
\end{aligned}
$$

establishing Equation (3) as desired.

# D. General covariance setting

In this section, we give our results for learning multivariate Gaussians with imperfect advice for the general covariance setting. We will later define analogs of $m(d, \alpha, \delta)$ and APPROXL1 from Section 2 to the unknown covariance setting: $m'(d, \alpha, \delta)$ and VECTORIZEDAPPROXL1 respectively. Then, after stating the guarantees of VECTORIZEDAPPROXL1, we show how to use them according to the strategy outlined in Section 1.2.2.

For the rest of this section, we assume that we get i.i.d. samples from $N(\mathbf{0}, \boldsymbol{\Sigma})$ and also that $\boldsymbol{\Sigma}$ is full rank. These are without loss of generality for the following reasons:

- Instead of a single sample from $N(\boldsymbol{\mu}, \boldsymbol{\Sigma})$, we will draw two samples $\boldsymbol{x}_1, \boldsymbol{x}_2 \sim N(\boldsymbol{\mu}, \boldsymbol{\Sigma})$ and consider $\boldsymbol{x}' = \frac{\boldsymbol{x}_1 + \boldsymbol{x}_2}{\sqrt{2}}$. One can check that $\boldsymbol{x}'$ is distributed according to $N(\mathbf{0}, \boldsymbol{\Sigma})$ and we only use a multiplicative factor of 2 additional samples, which is subsumed in the big-O.

- By Lemma A.13, the empirical covariance constructed from $d$ i.i.d. samples of $N(\mathbf{0}, \boldsymbol{\Sigma})$ will have the same rank as $\boldsymbol{\Sigma}$ itself, with probability at least $1 - \delta$. So, we can simply project and solve the problem on the full rank subspace of the empirical covariance matrix.

**Outline of this appendix section.** In Appendix D.1, we first elaborate on the adjustments mentioned in Section 1.2.2 to adapt the approach from the identity covariance setting to the unknown covariance setting, then show how to adapt the same approach as Section 2 to handle the general covariance setting in Appendix D.2. Appendix D.3 shows that optimization problem in Appendix D.2 can be reformulated as a semidefinite program (SDP) that is polynomial time solvable. Finally, Appendix D.4 presents the proof for Theorem 1.4.

## D.1. The adjustments

To begin, we elaborate on the adjustments mentioned in Section 1.2.2 to adapt the approach from the identity covariance setting to the unknown covariance setting.

The first adjustment relates to performing a suitable preconditioning process using an additional $d$ samples so that we can subsequently argue that $\lambda_{\min}(\mathbf{\Sigma}) \geq 1$. The idea is as follows: we will compute a preconditioning matrix $\mathbf{A}$ using $d$ i.i.d. samples such that $\mathbf{A\Sigma A}$ has eigenvalues at least 1, i.e. $\lambda_{\min}(\mathbf{A\Sigma A}) \geq 1$. That is, $\|(\mathbf{A\Sigma A})^{-1}\|_2 = \frac{1}{\lambda_{\min}(\mathbf{A\Sigma A})} \leq 1$. Then, we solve the problem treating $\mathbf{A\Sigma A}$ as our new $\mathbf{\Sigma}$. This adjustment succeeds with probability at least $1 - \delta$ for any given $\delta \in (0, 1)$ and is possible because, with probability 1, the empirical covariance $\widehat{\mathbf{\Sigma}}$ formed by using $d$ i.i.d. samples would have the same eigenspace as $\mathbf{\Sigma}$, and so we would have a bound on the ratios between the minimum eigenvalues between $\widehat{\mathbf{\Sigma}}$ and $\mathbf{\Sigma}$; see Lemma A.13.

**Lemma D.1.** *For any $\delta \in (0, 1)$, there is an explicit preconditioning process that uses $d$ i.i.d. samples from $N(\mathbf{0}, \mathbf{\Sigma})$ and succeeds with probability at least $1 - \delta$ in constructing a matrix $\mathbf{A} \in \mathbb{R}^{d \times d}$ such that $\lambda_{\min}(\mathbf{A\Sigma A}) \geq 1$. Furthermore, for any full rank PSD matrix $\widetilde{\mathbf{\Sigma}} \in \mathbb{R}^{d \times d}$, we have $\|(\mathbf{A}\widetilde{\mathbf{\Sigma}}\mathbf{A})^{-1/2}\mathbf{A\Sigma A}(\mathbf{A}\widetilde{\mathbf{\Sigma}}\mathbf{A})^{-1/2} - \mathbf{I}_d\| = \|\widetilde{\mathbf{\Sigma}}^{-1/2}\mathbf{\Sigma}\widetilde{\mathbf{\Sigma}}^{-1/2} - \mathbf{I}_d\|$.*

*Proof.* Suppose $\widehat{\mathbf{\Sigma}} \in \mathbb{R}^{d \times d}$ be the empirical covariance constructed from $n = d$ i.i.d. samples from $N(\mathbf{0}, \mathbf{\Sigma})$. Let $\lambda_1 \leq \ldots \leq \lambda_d$ and $\widehat{\lambda}_1 \leq \ldots \leq \widehat{\lambda}_d$ be the eigenvalues of $\mathbf{\Sigma}$ and $\widehat{\mathbf{\Sigma}}$ respectively. By Lemma A.13, we know that:

- With probability 1, we have that $\widehat{\mathbf{\Sigma}}$ and $\mathbf{\Sigma}$ share the same eigenspace.

- With probability at least $1 - \delta$, we have $\frac{\widehat{\lambda}_1}{\lambda_1} \leq 1 + c_0 \cdot \sqrt{\frac{d + \log 1/\delta}{d}}$ for some absolute constant $c_0$.

Let $\widehat{\mathbf{v}}_1, \ldots, \widehat{\mathbf{v}}_d$ be the eigenvectors corresponding to the eigenvalues $\widehat{\lambda}_1, \ldots, \widehat{\lambda}_d$. Define the following terms:

- $\mathbf{V}_{\text{small}} = \{i \in [d] : \widehat{\lambda}_i < 1\}$ and $\mathbf{V}_{\text{big}} = [d] \setminus \mathbf{V}_{\text{small}}$

- $\mathbf{\Pi}_{\text{small}} = \sum_{i \in \mathbf{V}_{\text{small}}} \widehat{\mathbf{v}}_i \widehat{\mathbf{v}}_i^\top$ and $\mathbf{\Pi}_{\text{big}} = \sum_{i \in \mathbf{V}_{\text{big}}} \widehat{\mathbf{v}}_i \widehat{\mathbf{v}}_i^\top$

- $\mathbf{A} = \sqrt{k}\mathbf{\Pi}_{\text{small}} + \mathbf{\Pi}_{\text{big}}$, where $k = \left(1 + c_0 \cdot \sqrt{\frac{d + \log 1/\delta}{n}}\right) \cdot \frac{1}{\widehat{\lambda}_1}$

We first argue that the smallest eigenvalue of $\mathbf{A\Sigma A}$ is at least 1, i.e. $\lambda_{\min}(\mathbf{A\Sigma A}) \geq 1$. To show this, it suffices to show that $\mathbf{u}^\top \mathbf{A\Sigma A}\mathbf{u} \geq 1$ for any unit vector $\mathbf{u} \in \mathbb{R}^d$. By definition,

$$\mathbf{u}^\top \mathbf{A\Sigma A}\mathbf{u} = k\mathbf{u}^\top \mathbf{\Pi}_{\text{small}}\mathbf{\Sigma}\mathbf{\Pi}_{\text{small}}\mathbf{u} + \mathbf{u}^\top \mathbf{\Pi}_{\text{big}}\mathbf{\Sigma}\mathbf{\Pi}_{\text{big}}\mathbf{u}$$

since the cross terms are zero because $\mathbf{u}^\top \mathbf{\Pi}_{\text{small}}\mathbf{\Sigma}\mathbf{\Pi}_{\text{big}}\mathbf{u} = \mathbf{u}^\top \mathbf{\Pi}_{\text{big}}\mathbf{\Sigma}\mathbf{\Pi}_{\text{small}}\mathbf{u} = 0$.

Now, observe that $\mathbf{u}^\top \mathbf{\Pi}_{\text{small}}\mathbf{\Sigma}\mathbf{\Pi}_{\text{small}}\mathbf{u} \geq \lambda_1 \cdot \|\mathbf{\Pi}_{\text{small}}\mathbf{u}\|_2^2$ and $\mathbf{u}^\top \mathbf{\Pi}_{\text{big}}\mathbf{\Sigma}\mathbf{\Pi}_{\text{big}}\mathbf{u} \geq \|\mathbf{\Pi}_{\text{big}}\mathbf{u}\|_2^2$. Meanwhile, by Pythagoras theorem, we know that $\|\mathbf{\Pi}_{\text{small}}\mathbf{u}\|_2^2 + \|\mathbf{\Pi}_{\text{big}}\mathbf{u}\|_2^2 = 1$. Therefore,

$$
\begin{aligned}
\mathbf{u}^\top \mathbf{A\Sigma A}\mathbf{u} =& k\mathbf{u}^\top \mathbf{\Pi}_{\text{small}}\mathbf{\Sigma}\mathbf{\Pi}_{\text{small}}\mathbf{u} + \mathbf{u}^\top \mathbf{\Pi}_{\text{big}}\mathbf{\Sigma}\mathbf{\Pi}_{\text{big}}\mathbf{u} \\
\geq& k\lambda_1 \cdot \|\mathbf{\Pi}_{\text{small}}\mathbf{u}\|_2^2 + \|\mathbf{\Pi}_{\text{big}}\mathbf{u}\|_2^2 \\
\geq& \left(\|\mathbf{\Pi}_{\text{small}}\mathbf{u}\|_2^2 + \|\mathbf{\Pi}_{\text{big}}\mathbf{u}\|_2^2\right) \\
=& 1
\end{aligned}
$$

where the last inequality is because $k = \left(1 + c_0 \cdot \sqrt{\frac{d + \log 1/\delta}{n}}\right) \cdot \frac{1}{\widehat{\lambda}_1} \geq \frac{1}{\lambda_1}$.

To complete the proof, note that for any full rank PSD matrix $\widetilde{\mathbf{\Sigma}} \in \mathbb{R}^{d \times d}$, we have

$$
\begin{aligned}
\|(\mathbf{A}\widetilde{\mathbf{\Sigma}}\mathbf{A})^{-1/2}\mathbf{A\Sigma A}(\mathbf{A}\widetilde{\mathbf{\Sigma}}\mathbf{A})^{-1/2} - \mathbf{I}_d\| &= \|(\mathbf{A}\widetilde{\mathbf{\Sigma}}\mathbf{A})^{-1}\mathbf{A\Sigma A} - \mathbf{I}_d\| \\
&= \|\mathbf{A}^{-1}\widetilde{\mathbf{\Sigma}}^{-1}\mathbf{\Sigma}\mathbf{A} - \mathbf{I}_d\|
\end{aligned}
$$

$$= \|\widetilde{\boldsymbol{\Sigma}}^{-1} \boldsymbol{\Sigma} \mathbf{A} \mathbf{A}^{-1} - \mathbf{I}_d\|$$

$$= \|\widetilde{\boldsymbol{\Sigma}}^{-1} \boldsymbol{\Sigma} - \mathbf{I}_d\|$$

$$= \|\widetilde{\boldsymbol{\Sigma}}^{-1/2} \boldsymbol{\Sigma} \widetilde{\boldsymbol{\Sigma}}^{-1/2} - \mathbf{I}_d\|$$

$\square$

The matrix $\mathbf{A}$ in Lemma 1.7 is essentially constructed by combining the eigenspace corresponding to "large eigenvalues" with a suitably upscaled eigenspace corresponding to "small eigenvalues" in the empirical covariance matrix obtained by $d$ i.i.d. samples and relying on Lemma A.13 for correctness arguments.

The second adjustment relates to showing that the partitioning idea also works for obtaining sample efficient $\ell_1$ estimates of $\mathrm{vec}(\boldsymbol{\Sigma} - \mathbf{I}_d)$. While an existence result suffices, we show that a simple probabilistic construction will in fact succeed with high probability.

**Lemma D.2.** *Fix dimension $d \geq 2$ and group size $k \leq d$. Consider the $q = 2$ setting where $\mathbf{T} \in \mathbb{R}^{d \times d}$ is a matrix. Define $w = \frac{10d(d-1)\log d}{k(k-1)}$. Pick sets $\mathbf{B}_1, \ldots, \mathbf{B}_w$ each of size $k$ uniformly at random (with replacement) from all the possible $\binom{d}{k}$ sets. With high probability in $d$, this is a $(q = 2, d, k, a = 1, b = \frac{30(d-1)\log d}{(k-1)})$-partitioning scheme.*

*Proof.* By definition, we have $|\mathbf{B}_1|, \ldots, |\mathbf{B}_w| = k$. Let us define $\mathcal{E}_{1,i,j}$ as the event that the cell $(i, j)$ of $\mathbf{T}$ *never* appears in any of the submatrices $\mathbf{T}_{\mathbf{B}_1}, \ldots, \mathbf{T}_{\mathbf{B}_w}$, and $\mathcal{E}_{2,i,j}$ as the event that the cell $(i, j)$ of $\mathbf{T}$ appears in strictly more than $b$ submatrices. In the rest of this proof, our goal is to show that $\Pr[\mathcal{E}_1]$ and $\Pr[\mathcal{E}_2]$ are small, where $\mathcal{E}_1 = \cup_{(i,j) \in [d] \times [d]} \mathcal{E}_{1,i,j}$ and $\mathcal{E}_2 = \cup_{(i,j) \in [d] \times [d]} \mathcal{E}_{2,i,j}$.

Fix any two *distinct* $i, j \in [d]$. For $\ell \in [w]$, let us define $X_\ell^{i,j}$ as the indicator event that the cell $(i, j)$ in $\mathbf{T}$ appears in the $\ell^{th}$ principal submatrix $\mathbf{T}_{\mathbf{B}_\ell}$ when $i, j \in \mathbf{B}_\ell$. By construction,

$$\Pr[X_\ell^{i,j} = 1] = \begin{cases} \frac{\binom{d-2}{k-2}}{\binom{d}{k}} = \frac{k(k-1)}{d(d-1)} & \text{if } i \neq j \\ \frac{\binom{d-1}{k-1}}{\binom{d}{k}} = \frac{k}{d} & \text{if } i = j \end{cases}$$

To analyze $\mathcal{E}_1$, we first consider $i, j \in [d]$ where $i \neq j$. We see that

$$\Pr[\mathcal{E}_{1,i,j}] = \prod_{\ell=1}^{w} \Pr[X_\ell^{i,j} = 0] = \left(1 - \frac{k(k-1)}{d(d-1)}\right)^w \leq \exp\left(-\frac{wk(k-1)}{d(d-1)}\right) = \exp(-10 \log d) = \frac{1}{d^{10}}$$

Meanwhile, when $i = j$,

$$\Pr[\mathcal{E}_{1,i,i}] = \prod_{\ell=1}^{w} \Pr[X_\ell^{i,i} = 0] = \left(1 - \frac{k}{d}\right)^w \leq \exp\left(-\frac{wk}{d}\right) \leq \exp(-10 \log d) = \frac{1}{d^{10}}$$

Taking union bound over $(i, j) \in [d] \times [d]$, we get

$$\Pr[\mathcal{E}_1] \leq \sum_{(i,j) \in [d] \times [d]} \Pr[\mathcal{E}_{1,i,j}] \leq \frac{d^2}{d^{10}} = \frac{1}{d^8}$$

To analyze $\mathcal{E}_2$, let us first define $Z^{i,j} = \sum_{\ell=1}^{w} X_\ell^{i,j}$ for any $i, j \in [d]$. Since the $X_\ell^{i,j}$ variables are indicators, linearity of expectations tells us that

$$\mathbb{E}[Z^{i,j}] = \sum_{\ell=1}^{w} \mathbb{E}[X_\ell^{i,j}] = \begin{cases} \sum_{\ell=1}^{w} \frac{k(k-1)}{d(d-1)} = \frac{wk(k-1)}{d(d-1)} & \text{if } i \neq j \\ \sum_{\ell=1}^{w} \frac{k}{d} = \frac{wk}{d} & \text{if } i = j \end{cases}$$

For $i \neq j$, applying Chernoff bound yields

$$\Pr[Z^{i,j} > (1+2) \cdot \mathbb{E}[Z^{i,j}]] \leq \exp\left(-\frac{\mathbb{E}[Z^{i,j}] \cdot 2^2}{2+2}\right) \leq \exp\left(-\mathbb{E}[Z^{i,j}]\right)$$

$$= \exp\left(-\frac{wk(k-1)}{d(d-1)}\right) = \exp\left(-10\log d\right) = \frac{1}{d^{10}}$$

Meanwhile, when $i = j$,

$$\Pr[Z^{i,i} > (1+2) \cdot \mathbb{E}[Z^{i,i}]] \leq \exp\left(-\frac{\mathbb{E}[Z^{i,i}] \cdot 2^2}{2+2}\right) \leq \exp\left(-\mathbb{E}[Z^{i,i}]\right) = \exp\left(-\frac{wk}{d}\right) \leq \exp\left(-10\log d\right) = \frac{1}{d^{10}}$$

By defining

$$b = 3 \cdot \max_{i,j \in [d]} \mathbb{E}[Z^{i,j}] = \frac{3wk}{d} = \frac{30(d-1)\log d}{(k-1)} \ ,$$

we see that $\Pr[E_{2,i,j}] = \Pr[Z^{i,j} > b] \leq \Pr[Z^{i,j} > (1+2) \cdot \mathbb{E}[Z^{i,j}]] \leq \frac{1}{d^{10}}$ and $\Pr[E_{2,i,i}] = \Pr[Z^{i,j} > b] \leq \Pr[Z^{i,i} > (1+2) \cdot \mathbb{E}[Z^{i,i}]] \leq \frac{1}{d^{10}}$. Therefore, taking union bound over $(i,j) \in [d] \times [d]$, we get

$$\Pr[\mathcal{E}_2] \leq \sum_{(i,j) \in [d] \times [d]} \Pr[\mathcal{E}_{2,i,j}] \leq \frac{d^2}{d^{10}} = \frac{1}{d^8}$$

In conclusion, this construction satisfy all 3 conditions of Definition 1.8 with high probability in $d$. $\square$

We can obtain a $(q = 2, d, k, a = 1, b = \mathcal{O}(\frac{d \log d}{k}))$-partitioning scheme by repeating the construction of Lemma 1.9 until it satisfies required conditions. Since it succeeds with high probability in $d$, we should not need many tries. The key idea behind utilizing partitioning schemes is that the marginal over a subset of indices $\mathbf{B} \subseteq [d]$ of a $d$-dimensional Gaussian with covariance matrix $\mathbf{\Sigma}$ has covariance matrix that is the principal submatrix $\mathbf{\Sigma_B}$ of $\mathbf{\Sigma}$. So, if we can obtain a multiplicative $\alpha$-approximation of a collection of principal submatrices $\mathbf{\Sigma_{B_1}}, \ldots \mathbf{\Sigma_{B_w}}$ such that all cells of $\mathbf{\Sigma}$ are present, then we can obtain a multiplicative $\alpha$-approximation of $\mathbf{\Sigma}$ just like in Section 2. Meanwhile, the $b$ parameter allows us to upper bound the overestimation factor due to repeated occurrences of any cell of $\mathbf{\Sigma}$.

### D.2. Following the approach from the identity covariance setting

We begin by defining a parameterized sample count $m'(d, \varepsilon, \delta)$, similar to Definition 2.1.

**Definition D.3.** Fix any $d \geq 1$, $\varepsilon > 0$, and $\delta \in (0, 1)$. We define $m'(d, \varepsilon, \delta) = n'_{d,\varepsilon} \cdot r_\delta$, where

$$n'_{d,\varepsilon} = \left\lceil 3200d \cdot \max\left\{\frac{1}{\varepsilon^2}, \frac{1}{\varepsilon}, 1\right\}\right\rceil \qquad \text{and} \qquad r_\delta = 1 + \left\lceil \log\left(\frac{12}{\delta}\right)\right\rceil$$

The VECTORIZEDAPPROXL1 algorithm corresponds to APPROXL1 in Section 2: it performs an exponential search to find the 2-approximation of the $\|\mathbf{\Sigma} - \mathbf{I}_d\|_F^2$ by repeatedly invoking the tolerant tester from Lemma 1.6 and then utilize a suitable partitioning scheme to bound $\|\text{vec}(\mathbf{\Sigma} - \mathbf{I}_d)\|_1$; see Lemma 1.9 and the discussions below it.

We now show that the VECTORIZEDAPPROXL1 algorithm has the following guarantees.

**Lemma D.4.** *Let* $\varepsilon$, $\delta$, $k$, $\alpha$, *and* $\zeta$ *be the input parameters to the* VECTORIZEDAPPROXL1 *algorithm (Algorithm 5). Given* $m(k, \alpha, \delta')$ *i.i.d. samples from* $N(\boldsymbol{\mu}, \mathbf{I}_d)$, *the* VECTORIZEDAPPROXL1 *algorithm succeeds with probability at least* $1 - \delta$ *and has the following properties:*

- *If* VECTORIZEDAPPROXL1 *outputs* Fail, *then* $\|\mathbf{\Sigma} - \mathbf{I}_d\|_F^2 > \zeta/2$.

- *If* VECTORIZEDAPPROXL1 *outputs* $\lambda \in \mathbb{R}$, *then*

$$\|\text{vec}(\mathbf{\Sigma} - \mathbf{I}_d)\|_1 \leq \lambda \leq 2\sqrt{k} \cdot \left(\frac{10d(d-1)\log d}{k(k-1)} \cdot \alpha + 2\|\text{vec}(\mathbf{\Sigma} - \mathbf{I}_d)\|_1\right)$$

---

**Algorithm 5** The VECTORIZEDAPPROXL1 algorithm.

1: **Input**: Error rate $\varepsilon > 0$, failure rate $\delta \in (0, 1)$, block size $k \in [d]$, lower bound $\alpha > 0$, upper bound $\zeta > 2\alpha$, and i.i.d. samples $\mathcal{S}$ from $N(\mathbf{0}, \mathbf{\Sigma})$
2: **Output**: Fail or $\lambda \in \mathbb{R}$
3: Define $w = \frac{10d(d-1)\log d}{k(k-1)}$, $\delta' = \frac{\delta}{w \cdot \lceil \log_2 \zeta/\alpha \rceil}$, and let $\mathbf{B}_1, \ldots, \mathbf{B}_w \subseteq [d]^2$ be a $(q = 2, d, k, a = 1, b = \mathcal{O}(\frac{d \log d}{k}))$-partitioning scheme as per Lemma 1.9
4: **for** $j \in \{1, \ldots, w\}$ **do**
5:     Define $\mathbf{S}_{\mathbf{B}_j} = \{\boldsymbol{x}_{\mathbf{B}_j} \in \mathbb{R}^{|\mathbf{B}_j|} : \boldsymbol{x} \in \mathbf{S}\}$ as the projected samples                        ▷ See Definition A.4
6:     Initialize $o_j = $ Fail
7:     **for** $i = 1, 2, \ldots, \lceil \log_2 \zeta/\alpha \rceil$ **do**
8:         Define $l_i = 2^{i-1} \cdot \alpha$
9:         Let Outcome be the output of the tolerant tester of Lemma 1.6 using sample set $\mathcal{S}_{\mathbf{B}_j}$ with $\varepsilon_1 = l_i$, $\varepsilon_2 = 2l_i$, and $\delta = \delta'$
10:         **if** Outcome is Accept **then**
11:             Set $o_j = l_i$ and **break**                             ▷ Escape inner loop for block $j$
12:         **end if**
13:     **end for**
14: **end for**
15: **if** there exists a Fail amongst $\{o_1, \ldots, o_w\}$ **then**
16:     **return** Fail
17: **else**
18:     **return** $\lambda = 2 \sum_{j=1}^{w} \sqrt{|\mathbf{B}_j|} \cdot o_j$                          ▷ $\lambda$ is an estimate for $\|vec(\Sigma - \mathbf{B}_d)\|_1$
19: **end if**

---

*Proof.* We begin by stating some properties of $o_1, \ldots, o_w$. Fix an arbitrary index $j \in \{1, \ldots, w\}$ and suppose $o_j$ is *not* a Fail, i.e. the tolerant tester of Lemma 1.6 outputs Accept for some $i^* \in \{1, 2, \ldots, \lceil \log_2 \zeta/\alpha \rceil\}$. Note that VECTORIZEDAP-PROXL1 sets $o_j = \ell_{i^*}$ and the tester outputs Reject for all smaller indices $i \in \{1, \ldots, i^* - 1\}$. Since the tester outputs Accept for $i^*$, we have that $\|\mathbf{\Sigma}_{\mathbf{B}_j} - \mathbf{I}_d\|_F \leq 2\ell_{i^*} = 2o_j$. Meanwhile, if $i^* > 1$, then $\|\mathbf{\Sigma}_{\mathbf{B}_j} - \mathbf{I}_d\|_F > \ell_{i^*-1} = \ell_{i^*}/2 = o_j/2$ since the tester outputs Reject for $i^* - 1$. Thus, we see that

- When $o_j$ is not Fail, we have $\|\mathbf{\Sigma}_{\mathbf{B}_j} - \mathbf{I}_d\|_F \leq 2o_j$.

- When $\|\mathbf{\Sigma}_{\mathbf{B}_j} - \mathbf{I}_d\|_F \leq 2\alpha$, we have $i^* = 1$ and $o_j = \ell_1 = \alpha$.

- When $\|\mathbf{\Sigma}_{\mathbf{B}_j} - \mathbf{I}_d\|_F > 2\alpha = 2\ell_1$, we have $i^* > 1$ and so $o_j < 2\|\mathbf{\Sigma}_{\mathbf{B}_j} - \mathbf{I}_d\|_F$.

**Success probability.** Fix an arbitrary index $i \in \{1, 2, \ldots, \lceil \log_2 \zeta/\alpha \rceil\}$ with $\ell_i = 2^{i-1}\alpha$, where $\ell_i \leq \ell_1 = \alpha$ for any $i$. We invoke the tolerant tester with $\varepsilon_2 = 2\ell_i = 2\varepsilon_1$, so the $i^{th}$ invocation uses at most $n'_{k,\varepsilon} \cdot r_\delta$ i.i.d. samples to succeed with probability at least $1 - \delta$; see Definition D.3 and Algorithm 3. So, with $m(k, \alpha, \delta')$ samples, *any* call to the tolerant tester succeeds with probability at least $1 - \delta'$, where $\delta' = \frac{\delta}{w \cdot \lceil \log_2 \zeta/\alpha \rceil}$. By construction, there will be at most $w \cdot \lceil \log_2 \zeta/\alpha \rceil$ calls to the tolerant tester. Therefore, by union bound, *all* calls to the tolerant tester jointly succeed with probability at least $1 - \delta$.

**Property 1.** When VECTORIZEDAPPROXL1 outputs Fail, there exists a Fail amongst $\{o_1, \ldots, o_w\}$. For any fixed index $j \in \{1, \ldots, w\}$, this can only happen when all calls to the tolerant tester outputs Reject. This means that $\|\mathbf{\Sigma}_{\mathbf{B}_j} - \mathbf{I}_d\|_F > \varepsilon_1 = \ell_i = 2^{i-1} \cdot \alpha$ for all $i \in \{1, 2, \ldots, \lceil \log_2 \zeta/\alpha \rceil\}$. In particular, this means that $\|\mathbf{\Sigma}_{\mathbf{B}_j} - \mathbf{I}_d\|_F > \zeta/2$.

**Property 2.** When VECTORIZEDAPPROXL1 outputs $\lambda = 2 \sum_{j=1}^{w} \sqrt{|\mathbf{B}_j|} \cdot o_j \in \mathbb{R}$, we can lower bound $\lambda$ as follows:

$$\lambda = 2 \sum_{j=1}^{w} \sqrt{|\mathbf{B}_j|} \cdot o_j$$

$$\geq 2 \sum_{j=1}^{w} \sqrt{|\mathbf{B}_j|} \cdot \frac{\|\mathbf{\Sigma}_{\mathbf{B}_j} - \mathbf{I}_d\|_F}{2} \hspace{3cm} \text{(since } \|\mathbf{\Sigma}_{\mathbf{B}_j} - \mathbf{I}_d\|_F \leq 2o_j)$$

$$= \sum_{j=1}^{w} \sqrt{|\mathbf{B}_j| \cdot \|\mathrm{vec}(\mathbf{\Sigma_{B}}_j - \mathbf{I}_d)\|_2^2} \qquad \text{(since } \|\mathbf{\Sigma_{B}}_j - \mathbf{I}_d\|_F^2 = \|\mathrm{vec}(\mathbf{\Sigma_{B}}_j - \mathbf{I}_d)\|_2^2)$$

$$\geq \sum_{j=1}^{w} \|\mathrm{vec}(\mathbf{\Sigma_{B}}_j - \mathbf{I}_d)\|_1 \qquad \text{(since } \|\mathrm{vec}(\mathbf{\Sigma_{B}}_j - \mathbf{I}_d)\|_1^2 \leq |\mathbf{B}_j| \cdot \|\mathrm{vec}(\mathbf{\Sigma_{B}}_j - \mathbf{I}_d)\|_2^2)$$

$$\geq \|\mathrm{vec}(\mathbf{\Sigma} - \mathbf{I}_d)\|_1 \qquad \text{(Since each cell in } \mathbf{\Sigma} \text{ appears at least } a = 1 \text{ times across all submatrices } \mathbf{\Sigma_{B}}_1, \dots, \mathbf{\Sigma_{B}}_w)$$

That is, $\lambda \geq \|\mathrm{vec}(\mathbf{\Sigma} - \mathbf{I}_d)\|_1$. Meanwhile, we can also upper bound $\lambda$ as follows:

$$\lambda = 2 \sum_{j=1}^{w} \sqrt{|\mathbf{B}_j|} \cdot o_j$$

$$\leq 2\sqrt{k} \cdot \sum_{j=1}^{w} o_j \qquad \text{(since } |\mathbf{B}_j| \leq k)$$

$$= 2\sqrt{k} \cdot \left( \sum_{\substack{j=1 \\ \|\mathbf{\Sigma_{B}}_j - \mathbf{I}_d\|_F \leq 2\alpha}}^{w} o_j + \sum_{\substack{j=1 \\ \|\mathbf{\Sigma_{B}}_j - \mathbf{I}_d\|_F > 2\alpha}}^{w} o_j \right) \qquad \text{(partitioning based on } \|\mathbf{\Sigma_{B}}_j - \mathbf{I}_d\|_F \text{ versus } 2\alpha)$$

$$= 2\sqrt{k} \cdot \left( \sum_{\substack{j=1 \\ \|\mathbf{\Sigma_{B}}_j - \mathbf{I}_d\|_F \leq 2\alpha}}^{w} \alpha + \sum_{\substack{j=1 \\ \|\mathbf{\Sigma_{B}}_j - \mathbf{I}_d\|_F > 2\alpha}}^{w} o_j \right) \qquad \text{(since } \|\mathbf{\Sigma_{B}}_j - \mathbf{I}_d\|_F \leq 2\alpha \text{ implies } o_j = \alpha)$$

$$\leq 2\sqrt{k} \cdot \left( \sum_{\substack{j=1 \\ \|\mathbf{\Sigma_{B}}_j - \mathbf{I}_d\|_F \leq 2\alpha}}^{w} \alpha + 2 \sum_{\substack{j=1 \\ \|\mathbf{\Sigma_{B}}_j - \mathbf{I}_d\|_F^2 \leq 2\alpha}}^{w} \|\mathbf{\Sigma_{B}}_j - \mathbf{I}_d\|_F \right)$$
$$\text{(since } \|\mathbf{\Sigma_{B}}_j - \mathbf{I}_d\|_F > 2\alpha \text{ implies } o_j \leq 2\|\mathbf{\Sigma_{B}}_j - \mathbf{I}_d\|_F)$$

$$= 2\sqrt{k} \cdot \left( \sum_{\substack{j=1 \\ \|\mathbf{\Sigma_{B}}_j - \mathbf{I}_d\|_F \leq 2\alpha}}^{w} \alpha + 2 \sum_{\substack{j=1 \\ \|\mathbf{\Sigma_{B}}_j - \mathbf{I}_d\|_F \leq 2\alpha}}^{w} \|\mathrm{vec}(\mathbf{\Sigma_{B}}_j - \mathbf{I}_d)\|_2 \right) \qquad \text{(since } \|\mathbf{\Sigma_{B}}_j - \mathbf{I}_d\|_F^2 = \|\mathrm{vec}(\mathbf{\Sigma_{B}}_j - \mathbf{I}_d)\|_2^2)$$

$$\leq 2\sqrt{k} \cdot \left( \sum_{\substack{j=1 \\ \|\mathbf{\Sigma_{B}}_j - \mathbf{I}_d\|_F \leq 2\alpha}}^{w} \alpha + 2 \sum_{\substack{j=1 \\ \|\mathbf{\Sigma_{B}}_j - \mathbf{I}_d\|_F \leq 2\alpha}}^{w} \|\mathrm{vec}(\mathbf{\Sigma_{B}}_j - \mathbf{I}_d)\|_1 \right)$$
$$\text{(since } \|\mathrm{vec}(\mathbf{\Sigma_{B}}_j - \mathbf{I}_d)\|_2 \leq \|\mathrm{vec}(\mathbf{\Sigma_{B}}_j - \mathbf{I}_d)\|_1)$$

$$\leq 2\sqrt{k} \cdot \left( w\alpha + 2 \sum_{\substack{j=1 \\ \|\mathbf{\Sigma_{B}}_j - \mathbf{I}_d\|_F^2 \leq 2\alpha}}^{w} \|\mathrm{vec}(\mathbf{\Sigma_{B}}_j - \mathbf{I}_d)\|_1 \right) \qquad \text{(since } |\{j \in [w] : \|\mathbf{\Sigma_{B}}_j - \mathbf{I}_d\|_F \leq 2\alpha\}| \leq w)$$

$$\leq 2\sqrt{k} \cdot (w\alpha + 2\|\mathrm{vec}(\mathbf{\Sigma} - \mathbf{I}_d)\|_1)$$
$$\text{(since } \sum_{\substack{j=1 \\ \|\mathbf{\Sigma_{B}}_j - \mathbf{I}_d\|_F \leq 2\alpha}}^{w} \|\mathrm{vec}(\mathbf{\Sigma_{B}}_j - \mathbf{I}_d)\|_1 \leq \sum_{j=1}^{w} \|\mathrm{vec}(\mathbf{\Sigma_{B}}_j - \mathbf{I}_d)\|_1 = \|\mathrm{vec}(\mathbf{\Sigma} - \mathbf{I}_d)\|_1)$$

That is, $\lambda \leq 2\sqrt{k} \cdot (w\alpha + 2\|\mathrm{vec}(\mathbf{\Sigma} - \mathbf{I}_d)\|_1)$, where $w = \frac{10d(d-1)\log d}{k(k-1)}$. The property follows by putting together both bounds. $\qquad \square$

Now, suppose VECTORIZEDAPPROXL1 tells us that $\|\mathrm{vec}(\mathbf{\Sigma} - \mathbf{I}_d)\|_1 \leq r$. We can then construct a SDP to search for a

candidate $\widehat{\boldsymbol{\Sigma}} \in \mathbb{R}^{d \times d}$ using $\mathcal{O}\left(\frac{r^2}{\varepsilon^4} \log \frac{1}{\delta}\right)$ samples from $N(\mathbf{0}, \boldsymbol{\Sigma})$.

**Lemma D.5.** *Fix $d \geq 1$, $r \geq 0$, and $\varepsilon, \delta > 0$. Given $\mathcal{O}\left(\frac{r^2}{\varepsilon^4} \log \frac{1}{\delta} + \frac{d + \sqrt{d \log(1/\delta)}}{\varepsilon^2}\right)$ samples from $N(\mathbf{0}, \boldsymbol{\Sigma})$ for some unknown $\boldsymbol{\Sigma} \in \mathbb{R}^{d \times d}$ with $\|\mathrm{vec}(\boldsymbol{\Sigma} - \mathbf{I}_d)\|_1 \leq r$, one can produce estimates $\widehat{\boldsymbol{\mu}} \in \mathbb{R}^d$ and $\widehat{\boldsymbol{\Sigma}} \in \mathbb{R}^{d \times d}$ in $\mathrm{poly}(n, d, \log(1/\varepsilon))$ time such that $\mathrm{d}_{\mathrm{TV}}(N(\boldsymbol{\mu}, \boldsymbol{\Sigma}), N(\widehat{\boldsymbol{\mu}}, \widehat{\boldsymbol{\Sigma}})) \leq \varepsilon$ with success probability at least $1 - \delta$.*

*Proof.* Suppose we get $n$ samples $\mathbf{y}_1, \ldots, \mathbf{y}_n \sim N(\mathbf{0}, \boldsymbol{\Sigma})$. For $i \in [n]$, we can re-express each $\mathbf{y}_i$ as $\mathbf{y}_i = \boldsymbol{\Sigma}^{1/2} \mathbf{g}_i$, for some $\mathbf{g}_i \sim N(\mathbf{0}, \mathbf{I}_d)$. Let us define $\mathbf{T} = \frac{1}{n} \sum_{i=1}^n \mathbf{g}_i \mathbf{g}_i^\top$ and $\mathbf{S} = \frac{1}{n} \sum_{i=1}^n \mathbf{y}_i \mathbf{y}_i^\top = \boldsymbol{\Sigma}^{1/2} \left(\frac{1}{n} \sum_{i=1}^n \mathbf{g}_i \mathbf{g}_i^\top\right) \boldsymbol{\Sigma}^{1/2} = \boldsymbol{\Sigma}^{1/2} \mathbf{T} \boldsymbol{\Sigma}^{1/2}$.

Let us define $\widehat{\boldsymbol{\Sigma}} \in \mathbb{R}^{d \times d}$ as follows:

$$\widehat{\boldsymbol{\Sigma}} = \underset{\substack{\mathbf{A} \in \mathbb{R}^{d \times d} \text{ is p.s.d.} \\ \|\mathrm{vec}(\mathbf{A} - \mathbf{I}_d)\|_1 \leq r \\ \lambda_{\min}(\mathbf{A}) \geq 1}}{\mathrm{argmin}} \sum_{i=1}^n \|\mathbf{A} - \mathbf{y}_i \mathbf{y}_i^\top\|_F^2 \tag{10}$$

Observe that $\boldsymbol{\Sigma}$ is a feasible solution to Equation (10). We show in Appendix D.3 that Equation (10) is a semidefinite program (SDP) that is polynomial time solvable.

Since $\boldsymbol{\Sigma}$ and $\widehat{\boldsymbol{\Sigma}}$ are symmetric p.s.d. matrices, observe that

$$\begin{aligned}
\sum_{i=1}^n \|\widehat{\boldsymbol{\Sigma}} - \mathbf{y}_i \mathbf{y}_i^\top\|_F^2 &= \sum_{i=1}^n \|\widehat{\boldsymbol{\Sigma}} - \boldsymbol{\Sigma}^{1/2} \mathbf{g}_i \mathbf{g}_i^\top \boldsymbol{\Sigma}^{1/2}\|_F^2 && \text{(Since } \mathbf{y}_i = \boldsymbol{\Sigma}^{1/2} \mathbf{g}_i\text{)} \\
&= \sum_{i=1}^n \mathrm{Tr}\left(\left(\widehat{\boldsymbol{\Sigma}} - \boldsymbol{\Sigma}^{1/2} \mathbf{g}_i \mathbf{g}_i^\top \boldsymbol{\Sigma}^{1/2}\right)^\top \left(\widehat{\boldsymbol{\Sigma}} - \boldsymbol{\Sigma}^{1/2} \mathbf{g}_i \mathbf{g}_i^\top \boldsymbol{\Sigma}^{1/2}\right)\right) \\
&&& \text{(Since } \|\mathbf{A}\|_F^2 = \mathrm{Tr}(\mathbf{A}^\top \mathbf{A}) \text{ for any matrix } \mathbf{A}\text{)} \\
&= \sum_{i=1}^n \mathrm{Tr}\left(\widehat{\boldsymbol{\Sigma}}^2 - 2 \mathbf{g}_i \mathbf{g}_i^\top \boldsymbol{\Sigma}^{1/2} \widehat{\boldsymbol{\Sigma}} \boldsymbol{\Sigma}^{1/2} + \mathbf{g}_i \mathbf{g}_i^\top \boldsymbol{\Sigma} \mathbf{g}_i \mathbf{g}_i^\top \boldsymbol{\Sigma}\right) \\
&&& \text{(Expanding and applying cyclic property of trace)}
\end{aligned}$$

Similarly, by replacing $\widehat{\boldsymbol{\Sigma}}$ with $\boldsymbol{\Sigma}$, we see that

$$\sum_{i=1}^n \|\boldsymbol{\Sigma} - \mathbf{y}_i \mathbf{y}_i^\top\|_F^2 = \sum_{i=1}^n \mathrm{Tr}\left(\boldsymbol{\Sigma}^2 - 2 \mathbf{g}_i \mathbf{g}_i^\top \boldsymbol{\Sigma}^2 + \mathbf{g}_i \mathbf{g}_i^\top \boldsymbol{\Sigma} \mathbf{g}_i \mathbf{g}_i^\top \boldsymbol{\Sigma}\right)$$

By standard SDP results (e.g. see (Vandenberghe & Boyd, 1996; Freund, 2004; Gärtner & Matousek, 2012)), Equation (10) can be solved optimally up to up to additive $\varepsilon$ in the objective function. We show explicitly in Appendix D.3 that our problem can be transformed into a SDP and be solved in $\mathrm{poly}(n, d, \log(1/\varepsilon))$ time. Since we solve up to additive $\varepsilon$ in the objective function, we have

$$\sum_{i=1}^n \|\widehat{\boldsymbol{\Sigma}} - \mathbf{y}_i \mathbf{y}_i^\top\|_F^2 \leq \varepsilon + \sum_{i=1}^n \|\boldsymbol{\Sigma} - \mathbf{y}_i \mathbf{y}_i^\top\|_F^2 \tag{11}$$

which implies that

$$\sum_{i=1}^n \mathrm{Tr}\left(\widehat{\boldsymbol{\Sigma}}^2 - 2 \mathbf{g}_i \mathbf{g}_i^\top \boldsymbol{\Sigma}^{1/2} \widehat{\boldsymbol{\Sigma}} \boldsymbol{\Sigma}^{1/2} + \mathbf{g}_i \mathbf{g}_i^\top \boldsymbol{\Sigma} \mathbf{g}_i \mathbf{g}_i^\top \boldsymbol{\Sigma}\right) \leq \varepsilon + \sum_{i=1}^n \mathrm{Tr}\left(\boldsymbol{\Sigma}^2 - 2 \mathbf{g}_i \mathbf{g}_i^\top \boldsymbol{\Sigma}^2 + \mathbf{g}_i \mathbf{g}_i^\top \boldsymbol{\Sigma} \mathbf{g}_i \mathbf{g}_i^\top \boldsymbol{\Sigma}\right)$$

Cancelling the common $\mathbf{g}_i \mathbf{g}_i^\top \boldsymbol{\Sigma} \mathbf{g}_i \mathbf{g}_i^\top \boldsymbol{\Sigma}$ term and rearranging, we get

$$\mathrm{Tr}\left(\widehat{\boldsymbol{\Sigma}}^2 - \boldsymbol{\Sigma}^2\right) \leq \frac{\varepsilon}{n} + \frac{2}{n} \sum_{i=1}^n \mathrm{Tr}\left(\mathbf{g}_i \mathbf{g}_i^\top \left(\boldsymbol{\Sigma}^{1/2} \widehat{\boldsymbol{\Sigma}} \boldsymbol{\Sigma}^{1/2} - \boldsymbol{\Sigma}^2\right)\right) \tag{12}$$

Therefore,

$$\|\widehat{\boldsymbol{\Sigma}} - \boldsymbol{\Sigma}\|_F^2 = \mathrm{Tr}\left(\left(\widehat{\boldsymbol{\Sigma}} - \boldsymbol{\Sigma}\right)^\top \left(\widehat{\boldsymbol{\Sigma}} - \boldsymbol{\Sigma}\right)\right)$$

$$= \mathrm{Tr}\left(\widehat{\boldsymbol{\Sigma}}^2 - 2\widehat{\boldsymbol{\Sigma}}\boldsymbol{\Sigma} + \boldsymbol{\Sigma}^2\right)$$

$$\leq \frac{\varepsilon}{n} + \frac{2}{n}\sum_{i=1}^n \mathrm{Tr}\left(\mathbf{g}_i\mathbf{g}_i^\top \left(\boldsymbol{\Sigma}^{1/2}\widehat{\boldsymbol{\Sigma}}\boldsymbol{\Sigma}^{1/2} - \boldsymbol{\Sigma}^2\right) - \widehat{\boldsymbol{\Sigma}}\boldsymbol{\Sigma} + \boldsymbol{\Sigma}^2\right)$$

(Add $2\boldsymbol{\Sigma}^2 - 2\widehat{\boldsymbol{\Sigma}}\boldsymbol{\Sigma}$ to both sides of Equation (12))

$$= \frac{\varepsilon}{n} + \frac{2}{n}\sum_{i=1}^n \mathrm{Tr}\left(\left(\mathbf{g}_i\mathbf{g}_i^\top - \mathbf{I}_d\right)\cdot\left(\boldsymbol{\Sigma}^{1/2}\widehat{\boldsymbol{\Sigma}}\boldsymbol{\Sigma}^{1/2} - \boldsymbol{\Sigma}^2\right)\right) \qquad \text{(Since } \mathrm{Tr}(\widehat{\boldsymbol{\Sigma}}\boldsymbol{\Sigma}) = \mathrm{Tr}(\boldsymbol{\Sigma}^{1/2}\widehat{\boldsymbol{\Sigma}}\boldsymbol{\Sigma}^{1/2})\text{)}$$

$$= \frac{\varepsilon}{n} + 2\cdot\mathrm{Tr}\left(\left(\boldsymbol{\Sigma}^{1/2}\widehat{\boldsymbol{\Sigma}} - \boldsymbol{\Sigma}^{1/2}\boldsymbol{\Sigma}\right)\cdot\boldsymbol{\Sigma}^{1/2}\cdot\left(\left(\frac{1}{n}\sum_{i=1}^n \mathbf{g}_i\mathbf{g}_i^\top\right) - \mathbf{I}_d\right)\right)$$

(Rearranging with cyclic property of trace)

$$\leq \frac{\varepsilon}{n} + 2\cdot\left\|\mathrm{vec}\left(\boldsymbol{\Sigma}\widehat{\boldsymbol{\Sigma}} - \boldsymbol{\Sigma}^2\right)\right\|_1\cdot\left\|\left(\frac{1}{n}\sum_{i=1}^n \mathbf{g}_i\mathbf{g}_i^\top\right) - \mathbf{I}_d\right\|_2$$

(By Lemma A.5 with $\mathbf{A} = \boldsymbol{\Sigma}^{1/2}\widehat{\boldsymbol{\Sigma}} - \boldsymbol{\Sigma}^{1/2}\boldsymbol{\Sigma}$, $\mathbf{B} = \boldsymbol{\Sigma}^{1/2}$, and $\mathbf{C} = \left(\frac{1}{n}\sum_{i=1}^n \mathbf{g}_i\mathbf{g}_i^\top\right) - \mathbf{I}_d$)

Recall that $\mathbf{T} = \frac{1}{n}\sum_{i=1}^n \mathbf{g}_i\mathbf{g}_i^\top$ and Lemma A.11 tells us that $\Pr\left(\|\mathbf{T} - \mathbf{I}_d\|_2 > \varepsilon\right) \leq 2\exp(-t^2 d)$ when the number of samples $n = \frac{c_0}{\varepsilon^2}\log\frac{2}{\delta}$, for some absolute constant $c_0$. So, to complete the proof, it suffices to upper bound $\left\|\mathrm{vec}\left(\boldsymbol{\Sigma}\widehat{\boldsymbol{\Sigma}} - \boldsymbol{\Sigma}^2\right)\right\|_1$. Consider the following:

$$\left\|\mathrm{vec}\left(\boldsymbol{\Sigma}\widehat{\boldsymbol{\Sigma}} - \boldsymbol{\Sigma}^2\right)\right\|_1 = \left\|\mathrm{vec}\left((\mathbf{I}_d - \boldsymbol{\Sigma})(\boldsymbol{\Sigma} - \widehat{\boldsymbol{\Sigma}}) - \boldsymbol{\Sigma} + \widehat{\boldsymbol{\Sigma}}\right)\right\|_1$$

$$\leq \|\mathrm{vec}(\mathbf{I}_d - \boldsymbol{\Sigma})\|_1\cdot\left\|\mathrm{vec}(\boldsymbol{\Sigma} - \widehat{\boldsymbol{\Sigma}})\right\|_1 + \left\|\mathrm{vec}(\widehat{\boldsymbol{\Sigma}} - \boldsymbol{\Sigma})\right\|_1 \qquad \text{(By Lemma A.6)}$$

$$= (\|\mathrm{vec}(\mathbf{I}_d - \boldsymbol{\Sigma})\|_1 + 1)\cdot\left\|\mathrm{vec}(\widehat{\boldsymbol{\Sigma}} - \mathbf{I}_d + \mathbf{I}_d - \boldsymbol{\Sigma})\right\|_1 \qquad \text{(Rearranging and adding 0)}$$

$$\leq (\|\mathrm{vec}(\mathbf{I}_d - \boldsymbol{\Sigma})\|_1 + 1)\cdot\left(\|\mathrm{vec}(\widehat{\boldsymbol{\Sigma}} - \mathbf{I}_d)\|_1 + \|\mathrm{vec}(\mathbf{I}_d - \boldsymbol{\Sigma})\|_1\right) \qquad \text{(By Lemma A.6)}$$

$$\leq (r+1)\cdot 2r \qquad \text{(Since } \|\mathrm{vec}(\mathbf{I}_d - \boldsymbol{\Sigma})\|_1 \leq r \text{ and } \left\|\mathrm{vec}(\widehat{\boldsymbol{\Sigma}} - \mathbf{I}_d)\right\|_1 \leq r\text{)}$$

When $\frac{2}{\varepsilon} \leq n$ and $n \in \mathcal{O}\left(\frac{r^2}{\varepsilon^4}\log\frac{1}{\delta}\right)$, the following holds with probability at least $1 - \delta$:

$$\|\widehat{\boldsymbol{\Sigma}} - \boldsymbol{\Sigma}\|_F^2 \leq \frac{\varepsilon}{n} + 2\cdot\left\|\mathrm{vec}\left(\boldsymbol{\Sigma}\widehat{\boldsymbol{\Sigma}} - \boldsymbol{\Sigma}^2\right)\right\|_1\cdot\|\mathbf{T} - \mathbf{I}_d\|_2 \leq \frac{\varepsilon}{n} + 4r(r+1)\cdot\|\mathbf{T} - \mathbf{I}_d\|_2 \leq \frac{\varepsilon}{n} + \frac{\varepsilon^2}{2} \leq \varepsilon^2$$

Now, Lemma A.12 tells us that the empirical mean $\widehat{\boldsymbol{\mu}}$ formed using $\mathcal{O}\left(\frac{d + \sqrt{d\log(1/\delta)}}{\varepsilon^2}\right)$ samples satisfies $(\widehat{\boldsymbol{\mu}} - \boldsymbol{\mu})^\top\boldsymbol{\Sigma}^{-1}(\widehat{\boldsymbol{\mu}} - \boldsymbol{\mu}) \leq \varepsilon^2$, with failure probability at most $\delta$. So,

$$d_{\mathrm{KL}}(N(\widehat{\boldsymbol{\mu}}, \widehat{\boldsymbol{\Sigma}}), N(\boldsymbol{\mu}, \boldsymbol{\Sigma}))$$

$$= \frac{1}{2}\cdot\left(\mathrm{Tr}(\boldsymbol{\Sigma}^{-1}\widehat{\boldsymbol{\Sigma}}) - d + (\boldsymbol{\mu} - \widehat{\boldsymbol{\mu}})^\top\boldsymbol{\Sigma}^{-1}(\boldsymbol{\mu} - \widehat{\boldsymbol{\mu}}) + \ln\left(\frac{\det\boldsymbol{\Sigma}}{\det\widehat{\boldsymbol{\Sigma}}}\right)\right)$$

$$\leq \frac{1}{2}\cdot\left((\boldsymbol{\mu} - \widehat{\boldsymbol{\mu}})^\top\boldsymbol{\Sigma}^{-1}(\boldsymbol{\mu} - \widehat{\boldsymbol{\mu}}) + \|\boldsymbol{\Sigma}^{-1/2}\widehat{\boldsymbol{\Sigma}}\boldsymbol{\Sigma}^{-1/2} - \mathbf{I}_d\|_F^2\right) \qquad \text{(By Lemma A.8)}$$

$$= \frac{1}{2}\cdot\left((\boldsymbol{\mu} - \widehat{\boldsymbol{\mu}})^\top\boldsymbol{\Sigma}^{-1}(\boldsymbol{\mu} - \widehat{\boldsymbol{\mu}}) + \|\widehat{\boldsymbol{\Sigma}}\boldsymbol{\Sigma}^{-1} - \mathbf{I}_d\|_F^2\right) \qquad \text{(By Lemma A.2)}$$

$$\leq \frac{1}{2} \cdot \left( \varepsilon^2 + \|\widehat{\boldsymbol{\Sigma}} \boldsymbol{\Sigma}^{-1} - \mathbf{I}_d\|_F^2 \right) \qquad \text{(Since } (\widehat{\boldsymbol{\mu}} - \boldsymbol{\mu})^\top \boldsymbol{\Sigma}^{-1} (\widehat{\boldsymbol{\mu}} - \boldsymbol{\mu}) \leq \varepsilon, \text{ with probability at least } 1 - \delta)$$

$$\leq \frac{1}{2} \cdot \left( \varepsilon^2 + \|\boldsymbol{\Sigma}^{-1}\|_2^2 \cdot \|\widehat{\boldsymbol{\Sigma}} - \boldsymbol{\Sigma}\|_F^2 \right) \qquad \text{(Submultiplicativity of Frobenius norm)}$$

$$\leq \frac{1}{2} \cdot \left( \varepsilon^2 + \|\widehat{\boldsymbol{\Sigma}} - \boldsymbol{\Sigma}\|_F^2 \right) \qquad \text{(Since } \|\boldsymbol{\Sigma}^{-1}\|_2 = \frac{1}{\lambda_{\min}(\boldsymbol{\Sigma})} \leq 1)$$

$$\leq \frac{1}{2} \cdot \left( \varepsilon^2 + \varepsilon^2 \right) \qquad \text{(From above, with probability at least } 1 - \delta)$$

$$= \varepsilon^2$$

By union bound, the above events jointly hold with probability at least $1 - 2\delta$. Thus, by symmetry of TV distance and Theorem A.10, we see that

$$d_{\text{TV}}(N(\boldsymbol{\mu}, \mathbf{I}_d), N(\widehat{\boldsymbol{\mu}}, \mathbf{I}_d)) = d_{\text{TV}}(N(\widehat{\boldsymbol{\mu}}, \mathbf{I}_d), N(\boldsymbol{\mu}, \mathbf{I}_d)) \leq \sqrt{\frac{1}{2} d_{\text{KL}}(N(\widehat{\boldsymbol{\mu}}, \mathbf{I}_d), N(\boldsymbol{\mu}, \mathbf{I}_d))} \leq \sqrt{\varepsilon^2} = \varepsilon$$

The claim holds by repeating the same argument after scaling $\delta$ by an appropriate constant. $\qquad \square$

---

**Algorithm 6** The TESTANDOPTIMIZECOVARIANCE algorithm.

---

1: **Input**: Error rate $\varepsilon > 0$, failure rate $\delta \in (0, 1)$, parameter $\eta \in [0, 1]$, and sample access to $N(\mathbf{0}, \boldsymbol{\Sigma})$
2: **Output**: $\widehat{\boldsymbol{\Sigma}} \in \mathbb{R}^{d \times d}$
3: Define $k = \lceil d^\eta \rceil$, $\alpha = \varepsilon d^{\eta-1}$, $\zeta = 4\varepsilon d$, and $\delta' = \frac{\delta}{w \cdot \lceil \log_2 \zeta/\alpha \rceil}$ $\qquad \triangleright$ Note: $\zeta > 2\alpha$
4: Draw $m'(k, \alpha, \delta')$ i.i.d. samples from $N(\mathbf{0}, \boldsymbol{\Sigma})$ and store it into a set $\mathcal{S}$ $\qquad \triangleright$ See Definition D.3
5: Let Outcome be the output of the VECTORIZEDAPPROXL1 algorithm given $\varepsilon$, $\delta$, $k$, $\alpha$, $\zeta$, and $\mathbf{S}$ as inputs
6: **if** Outcome is $\lambda \in \mathbb{R}$ and $\lambda < \varepsilon d$ **then**
7: $\quad$ Draw $n \in \widetilde{\mathcal{O}}(\lambda^2/\varepsilon^4)$ i.i.d. samples $\mathbf{y}_1, \ldots, \mathbf{y}_n \in \mathbb{R}^d$ from $N(\mathbf{0}, \mathbf{I}_d)$
8: $\quad$ **return** $\widehat{\boldsymbol{\Sigma}} = \operatorname{argmin}_{\substack{\mathbf{A} \in \mathbb{R}^{d \times d} \text{ is p.s.d.} \\ \|\text{vec}(\mathbf{A} - \mathbf{I}_d)\|_1 \leq \lambda \\ \lambda_{\min}(\mathbf{A}) \geq 1}} \sum_{i=1}^n \|\mathbf{A} - \mathbf{y}_i \mathbf{y}_i^\top\|_F^2$ $\qquad \triangleright$ See Equation (10)
9: **else**
10: $\quad$ Draw $2n \in \widetilde{\mathcal{O}}(d^2/\varepsilon^2)$ i.i.d. samples $\mathbf{y}_1, \ldots, \mathbf{y}_{2n} \in \mathbb{R}^d$ from $N(\mathbf{0}, \mathbf{I}_d)$
11: $\quad$ **return** $\widehat{\boldsymbol{\Sigma}} = \frac{1}{2n} \sum_{i=1}^{2n} (\mathbf{y}_{2i} - \mathbf{y}_{2i-1})(\mathbf{y}_{2i} - \mathbf{y}_{2i-1})^\top$ $\qquad \triangleright$ Empirical covariance
12: **end if**

---

**Theorem 1.2.** *For any given $\varepsilon, \delta \in (0, 1)$, $\eta \in [0, 1]$ and $\widetilde{\boldsymbol{\Sigma}} \in \mathbb{R}^{d \times d}$, TESTANDOPTIMIZECOVARIANCE uses $n \in \widetilde{\mathcal{O}} \left( \frac{d^2}{\varepsilon^2} \cdot \left( d^{-\eta} + \min \left\{ 1, f(\boldsymbol{\Sigma}, \widetilde{\boldsymbol{\Sigma}}, d, \eta, \varepsilon) \right\} \right) \right)$ where*

$$f(\boldsymbol{\Sigma}, \widetilde{\boldsymbol{\Sigma}}, d, \eta, \varepsilon) = \frac{\|\text{vec}(\widetilde{\boldsymbol{\Sigma}}^{-1/2} \boldsymbol{\Sigma} \widetilde{\boldsymbol{\Sigma}}^{-1/2} - \mathbf{I}_d)\|_1^2}{d^{2-\eta} \varepsilon^2}$$

*i.i.d. samples from $N(\boldsymbol{\mu}, \boldsymbol{\Sigma})$ for some unknown mean $\boldsymbol{\mu}$ and unknown covariance $\boldsymbol{\Sigma}$, and can produce $\widehat{\boldsymbol{\mu}}$ and $\widehat{\boldsymbol{\Sigma}}$ in $\text{poly}(n, d, \log(1/\varepsilon))$ time such that $d_{\text{TV}}(N(\boldsymbol{\mu}, \boldsymbol{\Sigma}), N(\widehat{\boldsymbol{\mu}}, \widehat{\boldsymbol{\Sigma}})) \leq \varepsilon$ with success probability at least $1 - \delta$.*

*Proof.* Without loss of generality, we may assume that $\widetilde{\boldsymbol{\Sigma}} = \mathbf{I}_d$. This is because we can pre-process all samples by pre-multiplying $\widetilde{\boldsymbol{\Sigma}}^{-1/2}$ each of them to yield i.i.d. samples from $N(\boldsymbol{\mu}, \widetilde{\boldsymbol{\Sigma}}^{-1/2} \boldsymbol{\Sigma} \widetilde{\boldsymbol{\Sigma}}^{-1/2})$ and then post-process the estimated $\widehat{\boldsymbol{\Sigma}}$ by outputting $\widetilde{\boldsymbol{\Sigma}}^{1/2} \widehat{\boldsymbol{\Sigma}} \widetilde{\boldsymbol{\Sigma}}^{1/2}$ instead.

**Correctness of $\widehat{\boldsymbol{\Sigma}}$ output.** Consider the TESTANDOPTIMIZECOVARIANCE algorithm given in Algorithm 6. Using the empirical mean $\widehat{\boldsymbol{\mu}} = \frac{1}{n} \sum_{i=1}^n \mathbf{y}_i$ formed by $\mathcal{O} \left( \frac{d + \sqrt{d \log(1/\delta)}}{\varepsilon^2} \right) \subseteq \widetilde{\mathcal{O}}(d/\varepsilon^2)$ samples, Lemma A.12 tells us that $(\widehat{\boldsymbol{\mu}} - \boldsymbol{\mu})^\top \boldsymbol{\Sigma}^{-1} (\widehat{\boldsymbol{\mu}} - \boldsymbol{\mu}) \leq \varepsilon$ with probability at least $1 - \delta$. There are two possible outputs for $\widehat{\boldsymbol{\Sigma}}$:

1. $\widehat{\boldsymbol{\Sigma}} = \operatorname{argmin}_{\substack{\mathbf{A} \in \mathbb{R}^{d \times d} \text{ is p.s.d.} \\ \|\text{vec}(\mathbf{A} - \mathbf{I}_d)\|_1 \leq r \\ \lambda_{\min}(\mathbf{A}) \geq 1 \leq 1}} \sum_{i=1}^n \|\mathbf{A} - \mathbf{y}_i \mathbf{y}_i^\top\|_F^2$, which can only happen when Outcome is $\lambda \in \mathbb{R}$

2. $\widehat{\boldsymbol{\Sigma}} = \frac{1}{2n}\sum_{i=1}^{2n}(\mathbf{y}_{2i}-\mathbf{y}_{2i-1})(\mathbf{y}_{2i}-\mathbf{y}_{2i-1})^{\top}$

Conditioned on VECTORIZEDAPPROXL1 succeeding, with probability at least $1-\delta$, we will now show that $d_{\text{TV}}(N(\boldsymbol{\mu},\boldsymbol{\Sigma}),N(\widehat{\boldsymbol{\mu}},\widehat{\boldsymbol{\Sigma}}))\leq\varepsilon$ and failure probability at most $2\delta$ in each of these cases, which implies the theorem statement as we can repeat the argument by scaling $\varepsilon$ and $\delta$ by appropriate constants.

**Case 1:** Using $r=\lambda$ as the upper bound, Lemma D.5 tells us that $d_{\text{TV}}(N(\boldsymbol{\mu},\boldsymbol{\Sigma}),N(\widehat{\boldsymbol{\mu}},\widehat{\boldsymbol{\Sigma}}))\leq\varepsilon$ with failure probability at most $\delta$ when $\widetilde{\mathcal{O}}(\frac{\lambda^2}{\varepsilon^4}+\frac{d}{\varepsilon^2})$ i.i.d. samples are used.

**Case 2:** With $\widetilde{\mathcal{O}}(d^2/\varepsilon^2)$ samples, Lemma A.12 tells us that $d_{\text{TV}}(N(\boldsymbol{\mu},\boldsymbol{\Sigma}),N(\widehat{\boldsymbol{\mu}},\widehat{\boldsymbol{\Sigma}}))\leq\varepsilon$ with failure probability at most $\delta$.

**Sample complexity used.** By Definition D.3, VECTORIZEDAPPROXL1 uses $|\mathbf{S}|=m'(k,\alpha,\delta')\in\widetilde{\mathcal{O}}(k/\alpha^2)$ samples to produce Outcome. Then, VECTORIZEDAPPROXL1 further uses $\widetilde{\mathcal{O}}(\lambda^2/\varepsilon^4)$ samples or $\widetilde{\mathcal{O}}(d^2/\varepsilon^2)$ samples depending on whether $\lambda<\varepsilon d$. So, TESTANDOPTIMIZECOVARIANCE has a total sample complexity of

$$\widetilde{\mathcal{O}}\left(\frac{k}{\alpha^2}+\min\left\{\frac{\lambda^2}{\varepsilon^4}+\frac{d}{\varepsilon^2},\frac{d^2}{\varepsilon^2}\right\}\right)\subseteq\widetilde{\mathcal{O}}\left(\frac{k}{\alpha^2}+\frac{d}{\varepsilon^2}+\min\left\{\frac{\lambda^2}{\varepsilon^4},\frac{d^2}{\varepsilon^2}\right\}\right) \tag{13}$$

Meanwhile, Lemma D.4 states that

$$\|\text{vec}(\boldsymbol{\Sigma}-\mathbf{I}_d)\|_1\leq\lambda\leq 2\sqrt{k}\cdot\left(\frac{10d(d-1)\log d}{k(k-1)}\cdot\alpha+2\|\text{vec}(\boldsymbol{\Sigma}-\mathbf{I}_d)\|_1\right)$$

whenever Outcome is $\lambda\in\mathbb{R}$. Since $(a+b)^2\leq 2a^2+2b^2$ for any two real numbers $a,b\in\mathbb{R}$, we see that

$$\frac{\lambda^2}{\varepsilon^4}\in\widetilde{\mathcal{O}}\left(\frac{k}{\varepsilon^4}\cdot\left(\frac{d^4\alpha^2}{k^4}+\|\text{vec}(\boldsymbol{\Sigma}-\mathbf{I}_d)\|_1^2\right)\right)\subseteq\widetilde{\mathcal{O}}\left(\frac{d^2}{\varepsilon^2}\cdot\left(\frac{d^2\alpha^2}{\varepsilon^2 k^3}+\frac{k\cdot\|\text{vec}(\boldsymbol{\Sigma}-\mathbf{I}_d)\|_1^2}{d^2\varepsilon^2}\right)\right) \tag{14}$$

Putting together Equation (13) and Equation (14), we see that the total sample complexity is

$$\widetilde{\mathcal{O}}\left(\frac{k}{\alpha^2}+\frac{d}{\varepsilon^2}+\frac{d^2}{\varepsilon^2}\cdot\min\left\{1,\frac{d^2\alpha^2}{\varepsilon^2 k^3}+\frac{k\cdot\|\text{vec}(\boldsymbol{\Sigma}-\mathbf{I}_d)\|_1^2}{d^2\varepsilon^2}\right\}\right)$$

Recalling that TESTANDOPTIMIZECOVARIANCE sets $k=\lceil d^\eta\rceil$, $\alpha=\varepsilon d^{\eta-1}$, with $0\leq\eta\leq 1$. So, the above expression simplifies to

$$\widetilde{\mathcal{O}}\left(\frac{d^{2-\eta}}{\varepsilon^2}+\frac{d}{\varepsilon^2}+\frac{d^2}{\varepsilon^2}\cdot\min\left\{1,d^{-\eta}+\frac{\|\text{vec}(\boldsymbol{\Sigma}-\mathbf{I}_d)\|_1^2}{d^{2-\eta}\varepsilon^2}\cdot\right\}\right)\subseteq\widetilde{\mathcal{O}}\left(\frac{d^2}{\varepsilon^2}\cdot\left(d^{-\eta}+\min\left\{1,\frac{\|\text{vec}(\boldsymbol{\Sigma}-\mathbf{I}_d)\|_1^2}{d^{2-\eta}\varepsilon^2}\cdot\right\}\right)\right)$$

To conclude, recall that $\boldsymbol{\Sigma}$ in the analysis above actually refers to the pre-processed $\widetilde{\boldsymbol{\Sigma}}^{-1/2}\boldsymbol{\Sigma}\widetilde{\boldsymbol{\Sigma}}^{-1/2}$. $\qquad\square$

**Remark on setting upper bound $\zeta$.** As $\zeta$ only affects the sample complexity logarithmically, one may be tempted to use a larger value than $\zeta=4\varepsilon d$. However, observe that running VECTORIZEDAPPROXL1 with a larger upper bound than $\zeta=4\varepsilon\sqrt{d}$ would not be helpful since $\|\boldsymbol{\Sigma}-\mathbf{I}_d\|_F^2>\zeta/2$ whenever VECTORIZEDAPPROXL1 currently returns Fail and we have $\|\text{vec}(\boldsymbol{\Sigma}-\mathbf{I}_d)\|_1\leq\lambda$ whenever VECTORIZEDAPPROXL1 returns $\lambda\in\mathbb{R}$. So, $\varepsilon d=\zeta/4<\|\boldsymbol{\Sigma}-\mathbf{I}_d\|_F^2=\|\text{vec}(\boldsymbol{\Sigma}-\mathbf{I}_d)\|_2\leq\|\text{vec}(\boldsymbol{\Sigma}-\mathbf{I}_d)\|_1\leq\lambda$ and TESTANDOPTIMIZEMEAN would have resorted to using the empirical mean anyway.

**Remark about early termination without the optimization step.** If there is no Fail amongst $\{o_1,\ldots,o_w\}$ and $4b\sum_{j=1}^{w}o_j^2\leq\varepsilon^2$ after Line 9 of VECTORIZEDAPPROXL1, then we could have just output $\widehat{\boldsymbol{\Sigma}}=\mathbf{I}_d$ without running the optimization step. This ie because since $4b\sum_{j=1}^{w}o_j^2\leq\varepsilon^2$ would imply $\|\boldsymbol{\Sigma}-\mathbf{I}_d\|_F^2\leq\varepsilon^2$ via

$$\|\boldsymbol{\Sigma}-\mathbf{I}_d\|_F^2\leq b\cdot\sum_{j=1}^{w}\|\boldsymbol{\Sigma}_{\mathbf{B}_j}-\mathbf{I}_d\|_F^2\leq b\cdot\sum_{j=1}^{w}(2o_j)^2\leq\varepsilon^2$$

Meanwhile, Lemma A.12 tells us that $(\widehat{\boldsymbol{\mu}} - \boldsymbol{\mu})^{\top} \boldsymbol{\Sigma}^{-1} (\widehat{\boldsymbol{\mu}} - \boldsymbol{\mu}) \leq \varepsilon^2$. Therefore, we see that

$$
\begin{aligned}
& d_{\mathrm{KL}}(N(\widehat{\boldsymbol{\mu}}, \widehat{\boldsymbol{\Sigma}}), N(\boldsymbol{\mu}, \boldsymbol{\Sigma})) \\
&= \frac{1}{2} \cdot \left( \mathrm{Tr}(\boldsymbol{\Sigma}^{-1} \widehat{\boldsymbol{\Sigma}}) - d + (\boldsymbol{\mu} - \widehat{\boldsymbol{\mu}})^{\top} \boldsymbol{\Sigma}^{-1} (\boldsymbol{\mu} - \widehat{\boldsymbol{\mu}}) + \ln \left( \frac{\det \boldsymbol{\Sigma}}{\det \widehat{\boldsymbol{\Sigma}}} \right) \right) \\
&\leq \frac{1}{2} \cdot \left( (\boldsymbol{\mu} - \widehat{\boldsymbol{\mu}})^{\top} \boldsymbol{\Sigma}^{-1} (\boldsymbol{\mu} - \widehat{\boldsymbol{\mu}}) + \| \boldsymbol{\Sigma}^{-1/2} \widehat{\boldsymbol{\Sigma}} \boldsymbol{\Sigma}^{-1/2} - \mathbf{I}_d \|_F^2 \right) && \text{(By Lemma A.8)} \\
&= \frac{1}{2} \cdot \left( (\boldsymbol{\mu} - \widehat{\boldsymbol{\mu}})^{\top} \boldsymbol{\Sigma}^{-1} (\boldsymbol{\mu} - \widehat{\boldsymbol{\mu}}) + \| \boldsymbol{\Sigma} - \mathbf{I}_d \|_F^2 \right) && \text{(Since } \widehat{\boldsymbol{\Sigma}} = \mathbf{I}_d) \\
&\leq \frac{1}{2} \cdot \left( \varepsilon^2 + \| \boldsymbol{\Sigma} - \mathbf{I}_d \|_F^2 \right) && \text{(Since } (\widehat{\boldsymbol{\mu}} - \boldsymbol{\mu})^{\top} \boldsymbol{\Sigma}^{-1} (\widehat{\boldsymbol{\mu}} - \boldsymbol{\mu}) \leq \varepsilon, \text{ with probability at least } 1 - \delta) \\
&\leq \frac{1}{2} \cdot \left( \varepsilon^2 + \alpha^2 \right) && \text{(Since } \| \boldsymbol{\Sigma} - \mathbf{I}_d \|_F^2 \leq \alpha^2, \text{ with probability at least } 1 - \delta) \\
&\leq \frac{1}{2} \cdot \left( \varepsilon^2 + \varepsilon^2 \right) && \text{(since } \alpha = \frac{\varepsilon k}{d} \leq \varepsilon \text{ as } k \leq d) \\
&= \varepsilon^2
\end{aligned}
$$

Thus, by symmetry of TV distance and Theorem A.10, we see that

$$
d_{\mathrm{TV}}(N(\boldsymbol{\mu}, \boldsymbol{\Sigma}), N(\widehat{\boldsymbol{\mu}}, \widehat{\boldsymbol{\Sigma}})) = d_{\mathrm{TV}}(N(\widehat{\boldsymbol{\mu}}, \widehat{\boldsymbol{\Sigma}}), N(\boldsymbol{\mu}, \boldsymbol{\Sigma})) \leq \sqrt{\frac{1}{2} d_{\mathrm{KL}}(N(\widehat{\boldsymbol{\mu}}, \widehat{\boldsymbol{\Sigma}}), N(\boldsymbol{\mu}, \boldsymbol{\Sigma}))} \leq \sqrt{\varepsilon^2} = \varepsilon
$$

### D.3. Polynomial running time of Equation (10)

In this section, we show that Equation (10) in Lemma D.5 can be reformulated as a semidefinite program (SDP) that is polynomial time solvable. Recall that we are given $n$ samples $\mathbf{y}_1, \ldots, \mathbf{y}_n \sim N(\mathbf{0}, \boldsymbol{\Sigma})$ under the assumption that $\| \mathrm{vec}(\boldsymbol{\Sigma} - \mathbf{I}_d) \|_1 \leq r$ for some $r > 0$, and Equation (10) was defined as follows:

$$
\widehat{\boldsymbol{\Sigma}} = \underset{\substack{\mathbf{A} \in \mathbb{R}^{d \times d} \text{ is p.s.d.} \\ \| \mathrm{vec}(\mathbf{A} - \mathbf{I}_d) \|_1 \leq r \\ \lambda_{\min}(\mathbf{A}) \geq 1}}{\operatorname{argmin}} \sum_{i=1}^{n} \| \mathbf{A} - \mathbf{y}_i \mathbf{y}_i^{\top} \|_F^2
$$

To convert our optimization problem to the standard SDP form, we "blow up" the problem dimension into some integer $n' \in \mathrm{poly}(d)$. Let $m$ be the number of constraints and $n'$ be the problem dimension. For symmetric matrices $\mathbf{C}, \mathbf{D}_1, \ldots, \mathbf{D}_m \in \mathbb{R}^{n' \times n'}$ and values $b_1, \ldots, b_m \in \mathbb{R}$, the standard form of a SDP is written as follows:

$$
\begin{aligned}
\min_{\mathbf{X} \in \mathbb{R}^{n' \times n'}} \quad & \langle \mathbf{C}, \mathbf{X} \rangle \\
\text{subject to} \quad & \langle \mathbf{D}_1, \mathbf{X} \rangle = b_1 \\
& \qquad \vdots \\
& \langle \mathbf{D}_m, \mathbf{X} \rangle = b_m \\
& \quad \mathbf{X} \succeq 0
\end{aligned}
\tag{15}
$$

where the inner product between two matrices $\mathbf{A}, \mathbf{B} \in \mathbb{R}^{n' \times n'}$ is written as

$$
\langle \mathbf{A}, \mathbf{B} \rangle = \sum_{i=1}^{n'} \sum_{j=1}^{n'} \mathbf{A}_{i,j} \mathbf{B}_{i,j}
$$

For further expositions about SDPs, we refer readers to (Vandenberghe & Boyd, 1996; Boyd & Vandenberghe, 2004; Freund, 2004; Gärtner & Matousek, 2012). In this section, we simply rely on the following known result to argue that our optimization problem will be polynomial time (in terms of $n$, $d$, and $r$) after showing how to frame Equation (10) in the standard SDP form.

**Theorem D.6** (Implied by (Huang et al., 2022)). *Consider an SDP instance of the form Equation (15). Suppose it has an optimal solution $\mathbf{X}^* \in \mathbb{R}^{n' \times n'}$ and any feasible solution $\mathbf{X} \in \mathbb{R}^{n' \times n'}$ satisfies $\| \mathbf{X} \|_2 \leq R$ for some $R > 0$. Then, there is an algorithm that produces $\widehat{\mathbf{X}}$ in $\mathcal{O}(\mathrm{poly}(n, d, \log(1/\varepsilon)))$ time such that $\langle \mathbf{C}, \widehat{\mathbf{X}} \rangle \leq \langle \mathbf{C}, \mathbf{X}^* \rangle + \varepsilon R \cdot \| \mathbf{C} \|_2$.*

*Remark* D.7. Apart from notational changes, Theorem 8.1 of (Huang et al., 2022) actually deals with the maximization problem but here we transform it to our minimization setting. They also guarantee additional bounds on the constraints with respect to $\widehat{\mathbf{X}}$, which we do not use.

In the following formulation, for any indices $i$ and $j$, we define $\delta_{i,j} \in \{0, 1\}$ as the indicator indicating whether $i = j$. This will be useful for representation of the identity matrix.

### D.3.1. RE-EXPRESSING THE OBJECTIVE FUNCTION

Observe that for any $i \in [n]$, we have

$$\|\mathbf{A} - \mathbf{y}_i\mathbf{y}_i^\top\|_F^2 = \mathrm{Tr}\left((\mathbf{A} - \mathbf{y}_i\mathbf{y}_i^\top)^\top(\mathbf{A} - \mathbf{y}_i\mathbf{y}_i^\top)\right)$$
$$= \mathrm{Tr}\left(\mathbf{A}^\top\mathbf{A}\right) - 2\mathrm{Tr}\left(\mathbf{y}_i\mathbf{y}_i^\top\mathbf{A}\right) + \mathrm{Tr}\left(\mathbf{y}_i\mathbf{y}_i^\top\mathbf{y}_i\mathbf{y}_i^\top\right)$$

Since $\mathbf{y}_1, \ldots, \mathbf{y}_n \in \mathbb{R}^d$ are constants with respect to the optimization problem, we can ignore the $\mathrm{Tr}\left(\mathbf{y}_i\mathbf{y}_i^\top\mathbf{y}_i\mathbf{y}_i^\top\right)$ term and instead minimize $n\mathrm{Tr}\left(\mathbf{A}^\top\mathbf{A}\right) - 2\sum_{i=1}^n \mathrm{Tr}\left(\mathbf{y}_i\mathbf{y}_i^\top\mathbf{A}\right)$. As $\mathbf{A}^\top\mathbf{A}$ is a quadratic expression, let us define an auxiliary matrix $\mathbf{B} \in \mathbb{R}^{d \times d}$ which we will later enforce $\mathrm{Tr}(\mathbf{B}) \geq \mathrm{Tr}(\mathbf{A}^T\mathbf{A})$. Defining a symmetric matrix $\mathbf{Y} = \sum_{i=1}^n \mathbf{y}_i\mathbf{y}_i^\top \in \mathbb{R}^{d \times d}$, the minimization objective becomes

$$n\mathrm{Tr}\left(\mathbf{B}\right) - 2\mathrm{Tr}\left(\mathbf{YA}\right) = n\mathbf{B}_{1,1} + \ldots + n\mathbf{B}_{d,d} - 2\langle\mathbf{Y}, \mathbf{A}\rangle \tag{16}$$

### D.3.2. DEFINING THE VARIABLE MATRIX X

Let $n' = 2d^2 + 3d + 2$ and let us define the SDP variable matrix $\mathbf{X} \in \mathbb{R}^{n' \times n'}$ as follows:

$$\mathbf{X} = \begin{bmatrix} \mathbf{B} & \mathbf{A}^\top & & & & & \\ \mathbf{A} & \mathbf{I}_d & & & & & \\ & & \mathbf{A} - \mathbf{I}_d & & & & \\ & & & \mathbf{U} & & & \\ & & & & \mathbf{S} & & \\ & & & & & s_{\mathbf{U}} & \\ & & & & & & s_{\mathbf{B}} \end{bmatrix} \in \mathbb{R}^{n' \times n'}$$

where the empty parts of $\mathbf{X}$ are zero matrices of appropriate sizes, $\mathbf{B} \in \mathbb{R}^{d \times d}$ is an auxiliary matrix aiming to capture $\mathbf{A}^\top\mathbf{A}$, and $\mathbf{U}$ and $\mathbf{S}$ are diagonal matrices of size $d^2$:

$$\mathbf{U} = \mathrm{diag}(u_{1,1}, u_{1,2}, \ldots, u_{1,d}, \ldots, u_{d,1}, \ldots, u_{d,d}) \in \mathbb{R}^{d^2 \times d^2}$$
$$\mathbf{S} = \mathrm{diag}(s_{1,1}, s_{1,2}, \ldots, s_{1,d}, \ldots, s_{d,1}, \ldots, s_{d,d}) \in \mathbb{R}^{d^2 \times d^2}$$

For convenience, we define

$$\mathbf{M} = \begin{bmatrix} \mathbf{B} & \mathbf{A}^\top \\ \mathbf{A} & \mathbf{I}_d \end{bmatrix} \in \mathbb{R}^{2d \times 2d}$$

so we can write

$$\mathbf{X} = \begin{bmatrix} \mathbf{M} & & & & & \\ & \mathbf{A} - \mathbf{I}_d & & & & \\ & & \mathbf{U} & & & \\ & & & \mathbf{S} & & \\ & & & & s_{\mathbf{U}} & \\ & & & & & s_{\mathbf{B}} \end{bmatrix} \in \mathbb{R}^{n' \times n'} \tag{17}$$

In the following subsections, we explain how to ensure that submatrices in $\mathbf{X}$ model the desired notions and constraints on $\mathbf{A}, \mathbf{B}$, and so on. For instance, we will use $\mathbf{U}$ to enforce $\|\mathrm{vec}(\mathbf{A} - \mathbf{I}_d)\|_1 \leq r$ in an element-wise fashion and use $\mathbf{S}$ and $s_{\mathbf{U}}$ for slack variables to transform inequality constraints to equality ones. The slack variable $s_{\mathbf{B}}$ is used for upper bounding the norm of $\mathbf{B}$ later, so that we can argue that the feasible region is bounded.

### D.3.3. DEFINING THE COST MATRIX $\mathbf{C}$

To capture the objective function Equation (16), let us define a symmetric cost matrix $\mathbf{C} \in \mathbb{R}^{n' \times n'}$ as follows:

$$\mathbf{C} = \begin{bmatrix} \mathrm{diag}(n, \ldots, n) & -\mathbf{Y} & \\ -\mathbf{Y} & \mathbf{0}_{d \times d} & \\ & & \mathbf{0}_{(2d^2+d+2) \times (2d^2+d+2)} \end{bmatrix} \in \mathbb{R}^{n' \times n'} \tag{18}$$

One can check that $\langle \mathbf{C}, \mathbf{X} \rangle = n\mathbf{B}_{1,1} + \ldots + n\mathbf{B}_{d,d} - 2\langle \mathbf{Y}, \mathbf{A} \rangle$.

### D.3.4. ENFORCING ZEROES, ONES, AND LINKING $\mathbf{A}$ ENTRIES WITH $\mathbf{A} - \mathbf{I}_d$

To enforce that the empty parts of $\mathbf{X}$ always solves to zeroes, we can define a symmetric constraint matrix $\mathbf{D}_{i,j}^{zero} \in \mathbb{R}^{n' \times n'}$ such that

$$(\mathbf{D}_{i,j}^{zero})_{i',j'} = \begin{cases} 1 & \text{if } i' = i \text{ and } j' = j \\ 0 & \text{otherwise} \end{cases}$$

and $b_{i,j}^{zero} = 0$. Then, $\langle \mathbf{D}_{i,j}^{zero}, \mathbf{X} \rangle = b_{i,j}^{zero}$ resolves to $\mathbf{X}_{i,j} = \langle \mathbf{D}_{i,j}^{zero}, \mathbf{X} \rangle = b_{i,j}^{zero} = 0$. We can similarly enforce that the appropriate part of $\mathbf{X}$ in $\mathbf{M}$ resolves to $\mathbf{I}_d$.

Now, to ensure that the $\mathbf{A}$ submatrices within $\mathbf{M}$ are appropriately linked to $\mathbf{A} - \mathbf{I}_d$, we can define a symmetric constraint matrix $\mathbf{D}_{i,j}^{\mathbf{A}} \in \mathbb{R}^{n' \times n'}$ such that

$$\mathbf{D}_{i,j}^{\mathbf{A}} = \begin{bmatrix} \mathbf{0}_{d \times d} & * & & & & \\ * & \mathbf{0}_{d \times d} & & & & \\ & & \dagger & & & \\ & & & \mathbf{0}_{d^2 \times d^2} & & \\ & & & & \mathbf{0}_{d^2 \times d^2} & \\ & & & & & 0 \\ & & & & & & 0 \end{bmatrix} \in \mathbb{R}^{n' \times n'}$$

and $b_{i,j}^{\mathbf{A}} = 0$, where $*$ contains $\frac{1}{4}$ at the $(i,j)$-th and $(j,i)$-th entries and $\dagger$ contains $\delta_{i,j} - \frac{1}{2}$ at the $(i,j)$-th and $(j,i)$-th entries, with 0 everywhere else; if $i = j$, we double the value. So, $\langle \mathbf{D}_{i,j}^{\mathbf{A}}, \mathbf{X} \rangle = b_{i,j}^{\mathbf{A}}$ would enforce that the $(i,j)$-th and $(j,i)$-th entries between the $\mathbf{A}$ submatrices within $\mathbf{M}$ and those in $\mathbf{A} - \mathbf{I}_d$ are appropriately linked.

### D.3.5. MODELING THE $\ell_1$ CONSTRAINT

To encode $\|\mathrm{vec}(\mathbf{A} - \mathbf{I}_d)\|_1 \le r$ in SDP form, let us define auxiliary variables $\{u_{i,j}\}_{i,j \in [d]}$ and define the linear constraints:

- $-A_{i,j} - u_{i,j} \le -\delta_{i,j}$, for all $i, j \in [d]$

- $A_{i,j} - u_{i,j} \le \delta_{i,j}$, for all $i, j \in [d]$

- $\sum_{i=1}^{d} \sum_{j=1}^{d} u_{i,j} \le r$

The first two constraints effectively encode $|A_{i,j} - \delta_{i,j}| \le u_{i,j}$ and so the third constraint captures $\|\mathrm{vec}(\mathbf{A} - \mathbf{I}_d)\|_1 \le r$ as desired. To convert the inequality constraint to an equality one, we use the slack variables $\{s_{i,j}\}_{i,j \in [d]}$ in $\mathbf{S}$. For instance, we can define symmetric constraint matrices $\mathbf{D}_{i,j}^{+} \in \mathbb{R}^{n' \times n'}$, $\mathbf{D}_{i,j}^{-} \in \mathbb{R}^{n' \times n'}$, and $\mathbf{D}_{i,j}^{r} \in \mathbb{R}^{n' \times n'}$ with $b_{i,j}^{+} = b_{i,j}^{-} = 0$ and $b^r = r$ as follows:

$$\mathbf{D}_{i,j}^{+} = \begin{bmatrix} \mathbf{0}_{d \times d} & * & & & & \\ * & \mathbf{0}_{d \times d} & & & & \\ & & \mathbf{0}_{d \times d} & & & \\ & & & \dagger & & \\ & & & & \ddagger & \\ & & & & & 0 \\ & & & & & & 0 \end{bmatrix} \qquad \mathbf{D}_{i,j}^{-} = \begin{bmatrix} \mathbf{0}_{d \times d} & -* & & & & \\ -* & \mathbf{0}_{d \times d} & & & & \\ & & \mathbf{0}_{d \times d} & & & \\ & & & \dagger & & \\ & & & & \ddagger & \\ & & & & & 0 \\ & & & & & & 0 \end{bmatrix}$$

$$\mathbf{D}_{i,j}^{r} = \begin{bmatrix} \mathbf{0}_{2d \times 2d} & & & & & \\ & \mathbf{0}_{d \times d} & & & & \\ & & \mathbf{1}_{d^2 \times d^2} & & & \\ & & & \mathbf{0}_{d^2 \times d^2} & & \\ & & & & 1 & \\ & & & & & 0 \end{bmatrix}$$

where $*$ contains $\frac{\delta_{i,j}-1}{4}$ at the $(i,j)$-th and $(j,i)$-th entries, $\dagger$ contains $-\frac{1}{2}$ at the $(i,j)$-th and $(j,i)$-th entries, and $\ddagger$ contains $\frac{1}{2}$ at the $(i,j)$-th and $(j,i)$-th entries, with 0 everywhere else; if $i = j$, we double the value. So, $\langle \mathbf{D}_{i,j}^{+}, \mathbf{X} \rangle = b_{i,j}^{+}$ models $\delta_{i,j} - A_{i,j} - u_{i,j} + s_{i,j} = 0$, $\langle \mathbf{D}_{i,j}^{-}, \mathbf{X} \rangle = b_{i,j}^{-}$ models $A_{i,j} - \delta_{i,j} - u_{i,j} + s_{i,j} = 0$, and $\langle \mathbf{D}_{i,j}^{r}, \mathbf{X} \rangle = b_{i,j}^{r}$ models $s_{\mathbf{S}} + \sum_{i=1} \sum_{j=1} u_{i,j} = r$.

### D.3.6. POSITIVE SEMIDEFINITE CONSTRAINTS

By known properties of the (generalized) Schur complement (see Section 1.4 and Section 1.6 of (Zhang, 2005)), it is known that $\mathbf{X} \succeq \mathbf{0}$ if and only if the following properties hold simultaneously:

1. $\mathbf{M} \succeq \mathbf{0}$

2. $\mathbf{A} - \mathbf{I}_d \succeq \mathbf{0} \iff \mathbf{A} \succeq \mathbf{I}_d \iff \lambda_{\min}(\mathbf{A}) \geq 1$, which also implies that $\mathbf{A}$ is psd

3. $\mathbf{U} \succeq \mathbf{0} \iff u_{1,1}, u_{1,2}, \ldots, u_{1,d}, \ldots, u_{d,1}, \ldots, u_{d,d} \geq 0$

4. $\mathbf{S} \succeq \mathbf{0} \iff s_{1,1}, s_{1,2}, \ldots, s_{1,d}, \ldots, s_{d,1}, \ldots, s_{d,d} \geq 0$

5. $s_{\mathbf{U}} \geq 0$

6. $s_{\mathbf{B}} \geq 0$

For the first property, since $\mathbf{I}_d \succ \mathbf{0}$, Schur complement tells us that $\mathbf{M} = \begin{bmatrix} \mathbf{B} & \mathbf{A}^{\top} \\ \mathbf{A} & \mathbf{I}_d \end{bmatrix} \succeq \mathbf{0}$ if and only if $\mathbf{B} \succeq \mathbf{A}^{\top}\mathbf{A}$.

Observe that $\mathbf{B} \succeq \mathbf{A}^{\top}\mathbf{A}$ implies $\mathrm{Tr}(\mathbf{B}) \geq \mathrm{Tr}(\mathbf{A}^{\top}\mathbf{A})$, which aligns with our intention of modeling $\mathbf{A}^{\top}\mathbf{A}$ by $\mathbf{B}$. Note that the objective function is $n\mathrm{Tr}(\mathbf{B}) - 2\mathrm{Tr}(\mathbf{YA})$ and we have that $\mathrm{Tr}(\mathbf{B}) \geq \mathrm{Tr}(\mathbf{A}^{\top}\mathbf{A})$ for all feasible matrices $\mathbf{B}$. Thus, for any pair $(\mathbf{A}^*, \mathbf{B}^*)$ that minimizes of the objective function, it has to be that $\mathrm{Tr}(\mathbf{B}^*) = \mathrm{Tr}((\mathbf{A}^*)^{\top}\mathbf{A}^*)$, since otherwise, the pair $(\mathbf{A}^*, \mathbf{B}^{**} = (\mathbf{A}^*)^{\top}\mathbf{A}^*)$ would have a smaller value.

### D.3.7. ENFORCING AN UPPER BOUND ON $\|\mathbf{B}\|_2$

To apply Theorem D.6, we need to argue that the feasible region of our SDP is bounded and non-empty, so that $\|\mathbf{X}\|_2$ is upper bounded. To do so, we need to enforce an upper bound on $\|\mathbf{B}\|_2$.

Since $\|\mathrm{vec}(\mathbf{A} - \mathbf{I}_d)\|_1 \leq r$, by triangle inequality and standard norm inequalities, we see that

$$\|\mathbf{A}\|_2 \leq \|\mathbf{A} - \mathbf{I}_d\|_2 + \|\mathbf{I}_d\|_2 \leq \|\mathbf{A} - \mathbf{I}_d\|_F + \|\mathbf{I}_d\|_2 = \|\mathrm{vec}(\mathbf{A} - \mathbf{I}_d)\|_2 + d \leq \|\mathrm{vec}(\mathbf{A} - \mathbf{I}_d)\|_1 + d \leq r + d \quad (19)$$

As $\mathbf{B}$ is supposed to model $\mathbf{A}^T\mathbf{A}$ and is constrained only by $\mathbf{B} \succeq \mathbf{A}^T\mathbf{A}$, it is feasible to enforce $\mathrm{Tr}(\mathbf{B}) \leq \|\mathbf{B}\|_F^2 \leq d\cdot(r+d)^4$ because

$$\|\mathbf{A}^T\mathbf{A}\|_F^2 \leq d \cdot \|\mathbf{A}^T\mathbf{A}\|_2^2 = d \cdot \|\mathbf{A}\|_2^4 \leq d \cdot (r+d)^4$$

To this end, let us define a symmetric constraint matrix $\mathbf{D}_{i,j}^{\mathbf{B}} \in \mathbb{R}^{n' \times n'}$ such that

$$\mathbf{D}^{\mathbf{B}} = \begin{bmatrix} \mathbf{I}_d & & \\ & \mathbf{0}_{(2d^2+2d+1) \times (2d^2+2d+1)} & \\ & & 1 \end{bmatrix} \in \mathbb{R}^{n' \times n'}$$

and $b^{\mathbf{B}} = d \cdot (r+d)^4$. Then, $\langle \mathbf{D}^{\mathbf{B}}, \mathbf{X} \rangle = b^{\mathbf{B}}$ resolves to $\mathrm{Tr}(\mathbf{B}) + s_{\mathbf{B}} = \langle \mathbf{D}^{\mathbf{B}}, \mathbf{X} \rangle = b^{\mathbf{B}} = d \cdot (r+d)^4$. In other words, since the slack variable $s_{\mathbf{B}}$ is non-negative, i.e. $s_{\mathbf{B}} \geq 0$, we have

$$\|\mathbf{B}\|_2 \leq \mathrm{Tr}(\mathbf{B}) \leq \|\mathbf{B}\|_F^2 \leq d \cdot (r+d)^4 \quad (20)$$

### D.3.8. BOUNDING $\|\mathbf{C}\|_2$ AND $\|\mathbf{X}\|_2$

Recalling the definition of $\mathbf{C}$ in Equation (18), we see that

$$\|\mathbf{C}\|_2 \leq \left\| \begin{bmatrix} \mathrm{diag}(n,\ldots,n) & -\mathbf{Y} \\ -\mathbf{Y} & \mathbf{0}_{d\times d} \end{bmatrix} \right\|_2 \leq n + \|\mathbf{Y}\|_2$$

Meanwhile, we know from Lemma A.13 that

$$\|\mathbf{Y}\|_2 \leq \|\boldsymbol{\Sigma}\|_2 \cdot \left( 1 + \mathcal{O}\left( \sqrt{\frac{d + \log 1/\delta}{n}} \right) \right)$$

with probability at least $1 - \delta$.

Recall from Algorithm 6 that when we solve the optimization problem of Equation (10), we have that $\|\mathrm{vec}(\boldsymbol{\Sigma} - \mathbf{I})\|_1 \leq r$. So, by a similar chain of arguments as Equation (19), we see that

$$\|\boldsymbol{\Sigma}\|_2 \leq \|\boldsymbol{\Sigma} - \mathbf{I}_d\|_2 + \|\mathbf{I}_d\|_2 \leq \|\boldsymbol{\Sigma} - \mathbf{I}_d\|_F + \|\mathbf{I}_d\|_2 = \|\mathrm{vec}(\boldsymbol{\Sigma} - \mathbf{I}_d)\|_2 + d \leq \|\mathrm{vec}(\boldsymbol{\Sigma} - \mathbf{I}_d)\|_1 + d = r + d$$

Therefore,

$$\|\mathbf{C}\|_2 \leq n + \|\boldsymbol{\Sigma}\|_2 \cdot \left( 1 + \mathcal{O}\left( \sqrt{\frac{d + \log 1/\delta}{n}} \right) \right) \leq n + (r + d) \cdot \left( 1 + \mathcal{O}\left( \sqrt{\frac{d + \log 1/\delta}{n}} \right) \right) \in \mathrm{poly}(n, d, r)$$

Meanwhile, recalling definition of $\mathbf{X}$ from Equation (17), we see that for *any* feasible solution $\mathbf{X}$,

$$\|\mathbf{X}\|_2 \leq \max\left\{ \|\mathbf{M}\|_2, \|\mathbf{A} - \mathbf{I}_d\|_2, \|\mathbf{U}\|_2, \|\mathbf{S}\|_2, s_{\mathbf{U}}, s_{\mathbf{B}} \right\}$$

By Equation (20), we have that $\|\mathbf{B}\|_2 \leq \sqrt{d} \cdot (r + d)^2$. So,

$$\|\mathbf{M}\|_2 \leq \|\mathbf{B}\|_2 + \|\mathbf{A}\|_2 + 1 \leq d \cdot (r + d)^4 + r + d + 1 \in \mathrm{poly}(d, r)$$

Also, all the remaining terms are in $\mathrm{poly}(r, d)$ since $\|\mathrm{vec}(\mathbf{A} - \mathbf{I}_d)\|_1 \leq r$. Therefore, $\|\mathbf{X}\|_2 \in \mathrm{poly}(d, r)$ with probability $1 - \delta$. So, $\|\mathbf{X}\|_2 \leq R$ for some $R \in \mathrm{poly}(d, r)$.

### D.3.9. PUTTING TOGETHER

Suppose we aim for an additive error of $\varepsilon' > 0$ in Equation (11) when we solve Equation (10). From above, we have that $\|\mathbf{C}\|_2, R \in \mathrm{poly}(n, d, r)$. Let us define $\varepsilon = \frac{\varepsilon'}{R \cdot \|\mathbf{C}\|_2}$ in Theorem D.6. Then, the algorithm of Theorem D.6 produces $\widehat{\mathbf{X}} \in \mathbb{R}^{n' \times n'}$ in $\mathrm{poly}(n, d, \log(1/\varepsilon)) \subseteq \mathrm{poly}(n, d, \log(\frac{R \cdot \|\mathbf{C}\|_2}{\varepsilon'})) \subseteq \mathrm{poly}(n, d, r, \log(1/\varepsilon'))$ time such that $\langle \mathbf{C}, \widehat{\mathbf{X}} \rangle \leq \langle \mathbf{C}, \mathbf{X}^* \rangle + \varepsilon R \cdot \|\mathbf{C}\|_2 = \langle \mathbf{C}, \mathbf{X}^* \rangle + \varepsilon'$ as desired.

## D.4. Hardness results

**Theorem 1.4.** *Suppose we are given a symmetric and positive-definite $\widetilde{\boldsymbol{\Sigma}} \in \mathbb{R}^{d \times d}$ as advice with only the guarantee that $\|\mathrm{vec}\left( \widetilde{\boldsymbol{\Sigma}}^{-\frac{1}{2}} \boldsymbol{\Sigma} \widetilde{\boldsymbol{\Sigma}}^{-\frac{1}{2}} - \mathbf{I}_d \right)\|_1 \leq \Delta$. Then, any algorithm that $(\varepsilon, \frac{2}{3})$-PAC learns $N(\mathbf{0}, \boldsymbol{\Sigma})$ requires $\Omega\left( \frac{\min\{d^2, \Delta^2/\varepsilon^2\}}{\varepsilon^2 \log(1/\varepsilon)} \right)$ samples in the worst case.*

*Proof.* Without loss of generality, we can assume $\widetilde{\boldsymbol{\Sigma}} = \mathbf{I}_d$ since, we can transform the input samples from $N(\mathbf{0}, \boldsymbol{\Sigma})$ as $\boldsymbol{x} \mapsto \widetilde{\boldsymbol{\Sigma}}^{-\frac{1}{2}} \boldsymbol{x}$ to get samples from $N\left( \mathbf{0}, \widetilde{\boldsymbol{\Sigma}}^{-\frac{1}{2}} \boldsymbol{\Sigma} \widetilde{\boldsymbol{\Sigma}}^{-\frac{1}{2}} \right)$, so that the advice quality in the transformed space (with advice taken to be $\mathbf{I}_d$) would be $\|\mathrm{vec}\left( \mathbf{I}_d \left( \widetilde{\boldsymbol{\Sigma}}^{-\frac{1}{2}} \boldsymbol{\Sigma} \widetilde{\boldsymbol{\Sigma}}^{-\frac{1}{2}} \right) \mathbf{I}_d - \mathbf{I}_d \right)\|_1$, which is equal to the original advice quality $\|\mathrm{vec}\left( \widetilde{\boldsymbol{\Sigma}}^{-\frac{1}{2}} \boldsymbol{\Sigma} \widetilde{\boldsymbol{\Sigma}}^{-\frac{1}{2}} - \mathbf{I}_d \right)\|_1$.

To use Lemma 3.1, we need to construct a set of $M$ distributions $f_1, \ldots, f_M$ with $f_i \triangleq N(\mathbf{0}, \boldsymbol{\Sigma}_i)$ such that

(i) Advice quality $\|\mathrm{vec}\left( \boldsymbol{\Sigma}_i - \mathbf{I}_d \right)\|_1 \leq \Delta$ for each $i \in [M]$,

(ii) the pairwise KL divergence $d_{KL}(f_i \| f_j) \leq \mathcal{O}(\varepsilon^2)$,

(iii) the the pairwise TV distance $d_{TV}(f_i, f_j) \geq \Omega(\varepsilon)$, and

(iv) $\log M \geq \Omega\left(\min\left(d^2, \frac{\Delta^2}{\varepsilon^2}\right)\right)$.

If we can construct such a family, Lemma 3.1 would give us a sample complexity lower bound of $\Omega\left(\min\left(\frac{d^2}{\varepsilon^2 \log(1/\varepsilon)}, \frac{\Delta^2}{\varepsilon^4 \log(1/\varepsilon)}\right)\right)$ to $(\varepsilon, 2/3)$-PAC learn the true disitribution, even given advice with quality $\leq \Delta$.

The following lemma is a Gilbert-Varshamov like bound on the existence of large sets of $s$-tuples of $[N]$ with pairwise distance $\geq (1 - \frac{1}{40})s$.

**Lemma D.8.** *For any $N \geq 200$ and $s > 0$, there exists $A = \{A_1, \ldots, A_M\} \subseteq [N]^s$ with $M \geq N^{\Omega(s)}$ such that for all pairs $i \neq j \in [M]$, $A_i$ and $A_j$ agree on $\leq s/40$ coordinates.*

And the following lemma follows from (Ashtiani et al., 2020), Lemma 6.4.

**Lemma D.9.** *For $p \geq 10$, there exist $N \geq 2^{\Omega(p^2)}$ matrices $\mathbf{U}_1, \ldots, \mathbf{U}_N \in \mathbb{R}^{p \times (p/10)}$ such that the columns of each $\mathbf{U}_i$ are the first $p \times 10$ columns of a $p \times p$ orthogonal matrix, and for each pair $i \neq j \in [N]$, $\|\mathbf{U}_i^\top \mathbf{U}_j\|_F^2 \leq p/20$.*

Let $d$ be a positive integer such that $d$ is a multiple of 10, and either $d^2$ is a multiple of $10 \left\lceil \frac{\Delta^2}{\varepsilon^2} \right\rceil$ or $d^2 < 10 \left\lceil \frac{\Delta^2}{\varepsilon^2} \right\rceil$. For every $\varepsilon > 0$ and $\Delta \geq \varepsilon$, there exist infinitely many choices of $d$ that satisfy these criteria. Take $p = \min\left(d, \frac{10}{d} \left\lceil \frac{\Delta^2}{\varepsilon^2} \right\rceil\right)$. Then, we will have $d = s \cdot p$ for some integer $s \geq 1$, and $p$ will be a multiple of 10. Also take $\mu = \frac{\Delta}{d}\sqrt{\frac{10}{p}} \lesssim \varepsilon/\sqrt{d}$ (using $p \leq (10/d)\lceil \Delta^2/\varepsilon^2 \rceil$).

Let $\mathbf{U}_1, \ldots, \mathbf{U}_N \in \mathbb{R}^{p \times (p/10)}$ be the $N \geq 2^{\Omega(p^2)}$ matrices as in Lemma D.9.

Also let $A_1, \ldots, A_M$ denote the $M \geq 2^{\Omega(p^2 s)} = 2^{\Omega(\min(d^2, \Delta^2/\varepsilon^2))}$ tuples in $[N]^s$ which agree pairwise only on $\leq s/40$ coordinates as guaranteed by Lemma D.8.

Then, we use the construction in Theorem 6.3 of (Ashtiani et al., 2020) block-wise to construct each covariance matrix $\mathbf{\Sigma}_i, i \in [M]$. We construct each $\mathbf{\Sigma}_i = \begin{bmatrix} \mathbf{\Sigma}_{i,1} & 0 & \cdots & 0 \\ 0 & \mathbf{\Sigma}_{i,2} & \cdots & 0 \\ 0 & 0 & \cdots & \mathbf{\Sigma}_{i,s} \end{bmatrix} \in \mathbb{R}^{d \times d}$, where each $\mathbf{\Sigma}_{i,j} = \mathbf{I}_p + \mu \mathbf{U}_{A_i(j)} \mathbf{U}_{A_i(j)}^\top \in \mathbb{R}^{p \times p}$.

By Lemma D.9, each $\mathbf{\Sigma}_{i,j} - \mathbf{I}_p = \mu \mathbf{U}_{A_i(j)} \mathbf{U}_{A_i(j)}^\top$ has $p/10$ eigenvalues which are equal to $\mu$ and the remaining $p - p/10$ eigenvalues equal to 0. Thus, we have $\|\mathbf{\Sigma}_i - \mathbf{I}_d\|_1 = \sum_{j=1}^s \|\mathbf{\Sigma}_{i,j} - I_p\|_1$ (decomposing the sum in the $\ell_1$ norm definition) $\leq \sum_{j=1}^s p \cdot \|\mathbf{\Sigma}_{i,j} - \mathbf{I}_p\|_F$ (by Cauchy-Schwarz) $\leq s \cdot p \cdot \sqrt{\frac{p}{10}\mu^2}$ (since Frobenius norm = Schatten-2 norm) $\leq d\mu\sqrt{p/10} \leq \Delta$ (substituting $sp = d$ and $\mu = (\Delta/d)\sqrt{10/p}$).

We have $\mathbf{\Sigma}_{i,j}^{-1} = \mathbf{I}_p - \frac{\mu}{1+\mu}\mathbf{U}_{A_i(j)}\mathbf{U}_{A_i(j)}^\top$ by construction of $\mathbf{U}_1, \ldots, \mathbf{U}_N$. By a similar calculation as in Theorem 6.3 of (Ashtiani et al., 2020), we have $d_{KL}(f_i, f_j) = \frac{1}{2}\mathrm{Tr}(\mathbf{\Sigma}_i^{-1}\mathbf{\Sigma}_j - \mathbf{I}_d) = \sum_{r=1}^s \frac{1}{2}\mathrm{Tr}(\mathbf{\Sigma}_{i,r}^{-1}\mathbf{\Sigma}_{j,r} - \mathbf{I}_p) \leq s\mu^2 \frac{p}{10} \leq \frac{d}{10}\mu^2 \leq \mathcal{O}(\varepsilon^2)$ (using $\mu \lesssim \varepsilon/\sqrt{d}$).

By using a similar argument as in Lemma 6.6 of (Ashtiani et al., 2020), we can lower bound the pairwise TV distance. By Theorem 1.1 in (Devroye et al., 2018), we have $d_{TV}(f_i, f_j) \geq \Theta\left(\min\{1, \|\mathbf{\Sigma}_i^{-1/2}\mathbf{\Sigma}_j\mathbf{\Sigma}_i^{-1/2} - \mathbf{I}_d\|_F\}\right)$. Since $\sigma_{\min}(\mathbf{\Sigma}_i^{-1/2}) = (1 + \mu)^{-1/2} = \Theta(1)$ when $\varepsilon \leq \sqrt{d}$, we have $d_{TV}(f_i, f_j) \geq \Omega(\varepsilon)$ when $\|\mathbf{\Sigma}_i - \mathbf{\Sigma}_j\|_F \geq \Omega(\varepsilon)$. We then

have

$$\|\mathbf{\Sigma}_i - \mathbf{\Sigma}_j\|_{\mathrm{F}}^2 = \sum_{r=1}^{s} \|\mathbf{\Sigma}_{i,r} - \mathbf{\Sigma}_{j,r}\|_{\mathrm{F}}^2 = \sum_{r=1}^{s} \mu^2 \|\mathbf{U}_{A_i(r)} \mathbf{U}_{A_i(r)}^\top - \mathbf{U}_{A_j(r)} \mathbf{U}_{A_j(r)}^\top\|_{\mathrm{F}}^2$$

$$= \sum_{r=1}^{s} \mu^2 \mathrm{Tr}\left(\left(\mathbf{U}_{A_i(r)} \mathbf{U}_{A_i(r)}^\top - \mathbf{U}_{A_j(r)} \mathbf{U}_{A_j(r)}^\top\right)\left(\mathbf{U}_{A_i(r)} \mathbf{U}_{A_i(r)}^\top - \mathbf{U}_{A_j(r)} \mathbf{U}_{A_j(r)}^\top\right)\right)$$

$$= \sum_{r=1}^{s} \mu^2 \left(\mathrm{Tr}(\mathbf{U}_{A_i(r)} U_{A_i(r)}^\top) + \mathrm{Tr}(\mathbf{U}_{A_j(r)} U_{A_j(r)}^\top) - 2\|\mathbf{U}_{A_i(r)}^\top \mathbf{U}_{A_j(r)}\|_{\mathrm{F}}^2\right)$$

(using $\mathbf{U}_{A_i(r)}^\top \mathbf{U}_{A_i(r)} = \mathbf{I}_{p/10}$, cyclic property of trace, and $\|A\|_{\mathrm{F}}^2 = \mathrm{Tr}(A^\top A)$)

$$= \cdot\frac{2\mu^2 d}{10} - 2\mu^2 \sum_{r=1}^{s} \|\mathbf{U}_{A_i(r)}^\top \mathbf{U}_{A_j(r)}\|_{\mathrm{F}}^2 \text{ (using } \mathrm{Tr}(\mathbf{U}_n \mathbf{U}_n^\top) = \tfrac{p}{10} \, \forall \, n \in [N], d = sp)$$

$$\geq \frac{2\mu^2 d}{10} - 2\mu^2 \left(\#\{A_i(r) = A_j(r)\}\frac{p}{10} + \#\{A_i(r) \neq A_j(r)\}\frac{p}{20}\right)$$

(using $\mathbf{U}_n^\top \mathbf{U}_n = \mathbf{I}_{p/10}$ and $\|\mathbf{U}_m^\top \mathbf{U}_n\|_{\mathrm{F}}^2 \leq p/20$ for $m \neq n$ by Lemma D.9)

$$\geq \frac{2\mu^2 d}{10} - 2\mu^2 \left(\frac{sp}{40} - \frac{sp}{20}\right) \geq \frac{9\mu^2 d}{40} \geq \Omega(\varepsilon^2) \text{ (using Lemma D.8).}$$

This concludes the proof that all requirements of Leamma 3.1 are met and we get the desired result by applying it.

$\square$

# E. Additional experiments

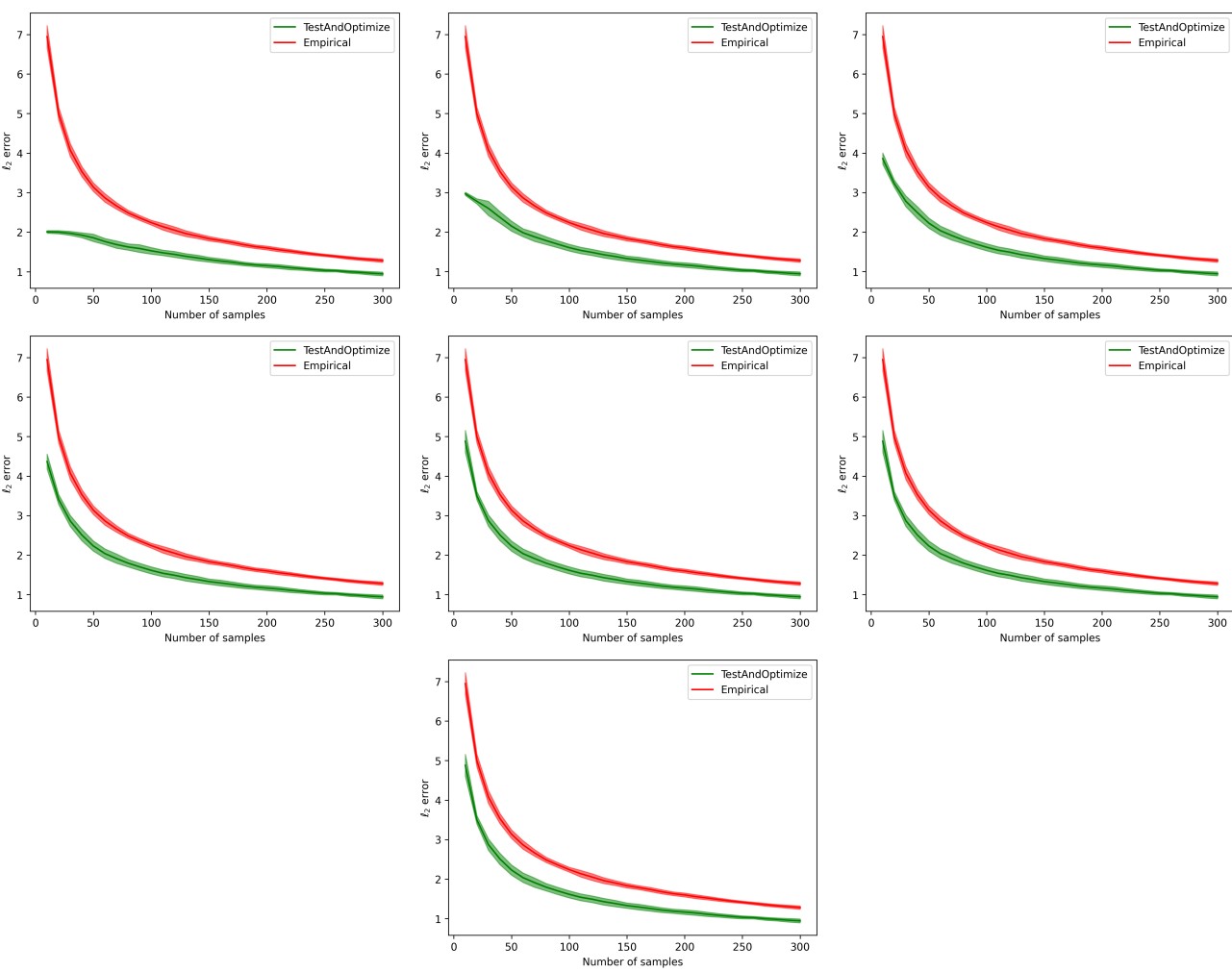

Figure 4: Here, $d = 500$, $s = 100$, and $q = \|\boldsymbol{\mu} - \widetilde{\boldsymbol{\mu}}\|_1 \in \{20, 30, 40, 50, 1000, 10000, 100000\}$. Error bars show standard deviation over 10 runs. Observe that the slope of the green line looks the same for all $q \geq 1000$ instances.

