# OpenReview forum: "Learning multivariate Gaussians with imperfect advice"
_ICML.cc/2025/Conference — ICML 2025 poster_

### Official Review · Reviewer_Mqzt · 2025-03-08

**Overall Recommendation:** 3

**Summary:**

This paper studies distribution learning within the framework of learning-augmented algorithms. Specifically, the authors study how to leverage potentially inaccurate "advice" about the true distribution to achieve lower sample complexity for learning multivariate Gaussian distributions.

For Gaussians with identity covariance, when given mean advice μ̃, the authors develop an algorithm achieving better sample complexity when the advice is accurate. Similar bounds were given for general covariance Gaussian. The paper also gives lower bound showing that the algorithms are optimal.

**Claims And Evidence:**

The main sample complexity claims are well-supported by the analysis.

The efficiency claims for their algorithms are supported by showing how to formulate the estimation problems as LASSO or SDP optimization problems.

The experimental section provides empirical validation of the theoretical claims, showing the performance advantages when advice quality is good. I find the evidence presented thorough.

**Essential References Not Discussed:**

No

**Experimental Designs Or Analyses:**

The experimental section only explores the sample complexity gains in the identity
covariance setting when one is given high quality advice. It'd be great if the work also studies the covariance case and see if the algorithm proposed is practical

**Methods And Evaluation Criteria:**

The paper uses LASSO-based method for mean estimation and SDP for covariance estimation. The latter may be somewhat inefficient in practice.

In terms of evaluation, the paper provides synthetic experiments, it'd be stronger if the experiments consider real data.

**Other Comments Or Suggestions:**

No.

**Other Strengths And Weaknesses:**

No

**Questions For Authors:**

No

**Relation To Broader Scientific Literature:**

The topic broadly lies in the literature on learning-augmented algorithm. This particular work also intersects with algorithmic high dimensional statistics. Both have a long line of works in the past ~10 years in the theory community.

**Theoretical Claims:**

I checked the main proofs in the main paper.

---

> ### Author Rebuttal · Authors · 2025-04-01
>
> Thank you for thoughtful and constructive review!
>
> You are right that while the SDP formulation used in TestAndOptimizeCovariance can be solved in polynomial time from a theoretical perspective (as we show in Appendix C.3), it is definitely impractical in practice. The focus of this work is to establish theoretical achievable sample complexity bounds and we hope that our work inspires future work that develops practical solutions matching our sample complexity bound.

---

### Official Review · Reviewer_M9iy · 2025-03-10

**Overall Recommendation:** 4

**Summary:**

One of the fundamental tasks in statistics is learning a Gaussian in total variation (TV) distance $\epsilon$. It is well known that the sample complexity for this task is on the order of $d/\epsilon^2$ when the covariance is the identity and $d^2/\epsilon^2$ for general Gaussians. This paper investigates whether the learning process can be made more efficient when given advice (or a warm start). For the first case (identity covariance), the advice consists of a vector $\tilde \mu$ that is somewhat close to $\mu$, while for the second case (general Gaussians), it consists of a matrix $\tilde \Sigma$ that is somewhat close to $\Sigma$.

The main results show that in the identity covariance case, if $\|\tilde \mu - \mu\|_1 < \epsilon d^{(1-\gamma)/2}$, then $O(d^{1-\Omega(\gamma)}/\epsilon^2)$ samples suffice. For general Gaussians, if $\|\text{vec}(\tilde{\Sigma}^{-1/2}\Sigma \tilde{\Sigma} - I )\|_1 < \epsilon d^{1-\gamma}$, then $O(d^{2-\gamma}/\epsilon^2)$ samples suffice. Here, $\|\cdot\|_1$ denotes the $\ell_1$-norm of vectors. Both algorithms run in polynomial time with respect to $d$ and the number of samples. These upper bounds are complemented by lower bounds showing that, up to constant factors in $\gamma$, no algorithm can achieve the task with fewer samples. Notably, the algorithm does not require knowledge of the quality of advice (i.e., $\|\tilde \mu - \mu\|_1$ and $\|\text{vec}(\tilde{\Sigma}^{-1/2}\Sigma \tilde{\Sigma} - I )\|_1$), and the lower bounds hold even when this quality is known.


The algorithmic approach consists of two main steps: (i) using a Gaussian tester combined with a search procedure to determine the quality of the given advice and (ii) designing an estimator that effectively utilizes the advice. The authors initially attempt to formulate the advice using the $\ell_2$-norm, since existing Gaussian testing methods rely on it. Ignoring computational constraints for a moment, a natural approach is to use a standard tournament argument along with a discretization of the unit ball to search for a vector that is $\epsilon$-close to $\mu$ in $\ell_2$-norm, which would imply a TV-distance guarantee. However, this approach requires at least $d$ samples, which is suboptimal. Instead, the argument is adapted to use the $\ell_1$-norm, leading to improved sample complexity. The paper further modifies the testers from (i) to be $\ell_1$-norm-based by partitioning the vector into chunks and using relationships between $\ell_1$ and $\ell_2$ norms for each chunk. While the tournament method has exponential computational complexity, the paper circumvents this by reformulating it as an appropriate optimization problem. This explains the use of the $\ell_1$-norm and outlines the proof of the first result.



The second result follows a similar high-level strategy, but the proof requires additional complexity to handle the matrix setting instead of the vector setting. Specifically, the partitioning of coordinates and the optimization program are more intricate. Additionally, an initial preconditioning step, inspired by differential privacy techniques, is introduced to ensure that $\|\Sigma^{-1}\|_2 \leq 1$.

The lower bound is derived using the Fano method, specifically a lemma from Ashtiani et al. (2020). This approach involves constructing a large family of Gaussians whose pairwise TV distance is at least $c_1 \epsilon$ while maintaining a KL divergence of at most $c_2 \epsilon$. The lemma is applied to Gaussians with means $\mu_i$ such that all $\mu_i$ satisfy the same upper bound on $\|\mu_i - \tilde \mu\|_1$ (advice) and have pairwise distances bounded by $\epsilon$. Such means can be constructed using codewords from an error-correcting code, as guaranteed by the Gilbert-Varshamov bound.

Finally, some numerical experiments are provided to demonstrate the sample complexity improvements when advice is used.

**Claims And Evidence:**

The paper contains an extensive proof sketch in the main body which describes the main ideas.

**Essential References Not Discussed:**

Regarding the reference at the end of the first page, there is a much simpler testing algorithm in Ilias Diakonikolas, Daniel M. Kane, and Ankit Pensia. “Gaussian Mean Testing Made Simple.” SOSA, 2023.

**Experimental Designs Or Analyses:**

See above.

**Methods And Evaluation Criteria:**

See below for theoretical methods. I haven't looked into detail on the experiments, but they seem to be clear.

**Other Comments Or Suggestions:**

I suggest explicitly stating the polylogarithmic factors in the theorem statements, as $\tilde{O}(\cdot)$ typically hides factors involving the variables inside the parentheses. In the current notation, it is unclear whether the sample complexity depends on $\delta$.

Additionally, does the lower bound hold for any $\Delta$ or only for a specific value of $\Delta$? It would be helpful to clarify this in the statement or in the text that follows.

**Other Strengths And Weaknesses:**

The paper is clearly written, and it provides lower bounds, which strengthen its theoretical contributions. However, there remains a gap between the upper and lower bounds. Specifically, when $\Delta^2 = \epsilon^2 d^{1-5\eta}$, the upper bound (ignoring logarithmic factors) is $d^{1-\eta}/\epsilon^2$, while the lower bound is $d^{1-5\eta}/\epsilon^2$.

Overall, I do not have any major concerns with the paper and would recommend its acceptance.

**Questions For Authors:**

Included above.

**Relation To Broader Scientific Literature:**

The paper notes that advice has been studied in various problems within the context of online algorithms. While I have not seen this specific problem explicitly stated as an open question in prior work, I find it reasonable and believe it is a valuable addition to the literature.

**Theoretical Claims:**

I read the proof sketch of the introduction and the proofs presented in the main body. I don't have any major issues with these technical parts.

---

> ### Author Rebuttal · Authors · 2025-04-01
>
> Thank you for thoughtful and constructive review!
>
> Thank you for the reference of "Gaussian Mean Testing Made Simple", we will add it in our revision.
>
> You are also indeed correct that there is a fundamental difference between our upper and lower bounds in terms of whether the advice quality is known. Our upper bounds (and the problem formulation) assume that the advice quality ($\\| \widetilde{\mu} - \mu \\|_1$ and $\\| \widetilde{\Sigma}^{-1/2} \Sigma \widetilde{\Sigma}^{-1/2} - I_d \\|_1$ respectively) is not known to the learning-augmented algorithms, even as an upper bound. Whereas the lower bounds apply even for the *weaker* problem where (an upper bound of) the advice quality is known to the algorithms.

---

### Official Review · Reviewer_qviZ · 2025-03-14

**Overall Recommendation:** 4

**Summary:**

This paper studied the problem of learning high dimensional Gaussians given imperfect advice. In particular, the authors show that given imperfect advice that is close to the true statistic quantity (in $l_1$ norm), both the following tasks have polynomial improvements with respect to the dependence of the dimension $d$ in sample complexity compared with the algorithms without using imperfect advice. The tasks are: given an imperfect advice of Gaussian mean, learn the true Gaussian mean of $N(\mu, I)$; given imperfect advice of Gaussian mean and covariance, learn the true mean and covariance of $N(\mu, \Sigma)$.

**Claims And Evidence:**

Yes, they are supported by clear and convincing evidence.

**Essential References Not Discussed:**

N/A

**Experimental Designs Or Analyses:**

Yes

**Methods And Evaluation Criteria:**

Yes.

**Other Comments Or Suggestions:**

N/A

**Other Strengths And Weaknesses:**

Strengths:
1. I think this paper is clearly written with very useful explanation of the algorithm and motivation.
2. I think the results are interesting and strong. It is interesting to see that given imperfect advice one can break the sample complexity lower bounds.
3. The paper is solid with highly technical contributions.

Question/Weakness
I seems the SDP algorithm blows up the dimension to n' which is a polynomial of $d$. I think it would be a nice improvement to have more efficient algorithm for this.

**Questions For Authors:**

N/A

**Relation To Broader Scientific Literature:**

The paper contributes well to the vast thread research of gaussian testing/learning, providing interesting results and improving the sample complexity of gaussian distribution learning. For example, the prior result on gaussian mean learning requires O(d) sample, however, the authors showed that given some accurate knowledge of the mean $\mu$, the sample complexity can be improved to $O(d^{1-\eta})$.

**Theoretical Claims:**

Checked the proof in the main body.

---

> ### Author Rebuttal · Authors · 2025-04-01
>
> Thank you for thoughtful and constructive review!
>
> You are right that while the SDP formulation used in TestAndOptimizeCovariance can be solved in polynomial time from a theoretical perspective (as we show in Appendix C.3), it is definitely impractical in practice. The focus of this work is to establish theoretical achievable sample complexity bounds and we hope that our work inspires future work that develops practical solutions matching our sample complexity bound.

---

### Official Review · Reviewer_nuex · 2025-03-17

**Overall Recommendation:** 3

**Summary:**

Authors study bounds on sample complexity of mean and covariance estimation of multivariate gaussians
in learning-augmented setting. Here, the algorithm receives an untrusted advice in the form
of estimates of the mean and the covariance matrix.
The goal is to improve the sample complexity beyond the classical bounds of $O(d/\epsilon^2)$ and $O(d^2/\epsilon^2)$ respectively if the advice is close to correct.

The authors are motivated by the fact that there is an algorithm by Diakonikolas et al. which, given
precise estimate of the mean, can certify that this estimate is correct in time only $O(\sqrt{d}/epsilon^2)$
which suggests that an improvement might be possible also with an approximately correct estimate.
Indeed, they propose an algorithm which, given an estimate $\tilde \mu$ of the mean $\mu$ requires
only $\tilde O(d^{1-\eta}/\epsilon^2)$ samples if $\lVert \mu -\tilde \mu\rVert_1 < \epsilon\sqrt{d}\cdot d^{-5\eta/2}$. They provide a similar improvement in the case of covariance estimation.

**Claims And Evidence:**

their claims are well supported with proofs and experiments

**Essential References Not Discussed:**

I am not aware of any.

**Experimental Designs Or Analyses:**

I did not try to replicate the experiments

**Methods And Evaluation Criteria:**

performance metrics studied are well established and the benchmark datasets are suitable for a theoretical study.

**Other Comments Or Suggestions:**

none

**Other Strengths And Weaknesses:**

Strengths:

* nice and promising results on an important problem

* performance of their algorithm with known prediction error is almost optimal, as supported by their lower bounds

Weaknesses:

* performance with perfect advice does not seem to match the certification algorithm by Diakonikolas et al. (2017).

* not clear whether the performance of their algorithm with unknown prediction error is optimal: lower bounds provided only for the case of known prediction error.

**Questions For Authors:**

can you please comment on the performance of your algorithm with $\lVert \eta - \tilde \eta\rVert_1$ approaching 0 and how does it match Diakonikolas et al.?

**Relation To Broader Scientific Literature:**

Authors study an important problem and their study fits well within the literature on learning-augmented algorithms.

**Theoretical Claims:**

I did not check fully any proof. However, their approach seems viable.

---

> ### Author Rebuttal · Authors · 2025-04-01
>
> Thank you for thoughtful and constructive review!
>
> For the mean setting, one can trivially obtain a sample complexity of $\widetilde{O}(\sqrt{d}/\varepsilon^2)$ when the advice quality is "good enough" by first running the tolerant tester (see Lemam 1.5) with $k = d$ and $\alpha = \varepsilon$, and returning the advice mean directly if the tolerant tester accepts (since this would imply that using the advice mean only incurs a KL error of at most $\varepsilon^2$). However, we agree that there is a discontinguity in the sample complexity between this "good enough" advice quality and our Theorem 1.1. This is in contrast to the covariance setting where our parameters in Theorem 1.2 allow us to recover $\widetilde{O}(d^2 / \varepsilon^2)$ directly. We are currently unsure if such brittleness/discontiguity is inherent for the mean problem setting but it would be interesting future work to pursue this inquiry.
>
> You are also indeed correct that there is a fundamental difference between our upper and lower bounds in terms of whether the advice quality is known. Our upper bounds (and the problem formulation) assume that the advice quality ($\\| \widetilde{\mu} - \mu \\|_1$ and $\\| \widetilde{\Sigma}^{-1/2} \Sigma \widetilde{\Sigma}^{-1/2} - I_d \\|_1$ respectively) is not known to the learning-augmented algorithms, even as an upper bound. Whereas the lower bounds apply even for the \emph{weaker} problem where (an upper bound of) the advice quality is known to the algorithms.
>
> As a side note, there is a typo in the definition of $\alpha$ in Algorithm 6 and on Line 1847. It should be $\alpha = \varepsilon d^{\eta - 1}$ and our subsequent calculations, theorem statement, and conclusions about the covariance setting remains unchanged. There are also some missing tildes to capture the $\log d$ factors in the proof. We will fix these errors in the revision.

---

> > ### Comment · Reviewer_nuex · 2025-04-02
> >
> > thank you for your careful response. I understand that your lower bound is for an easier problem and therefore it is a stronger result. But still it does not show the tightness of your upper bound. I maintain my (positive) score.

---

### Decision · Program_Chairs · 2025-05-01

**Decision:**

Accept (poster)

**Comment:**

The paper analyzes the sample complexity of mean and covariance estimation for multivariate Gaussians in a learning-augmented setting, where the algorithm receives untrusted advice in the form of estimated parameters. The authors show how accurate advice can reduce the required sample complexity below the classical bounds for mean and covariance estimation.

The main concern by multiple reviewers is the gap between the upper bounds achieved in this paper compared to the upper bounds achieved by Diakonikolas et. al. (2018), which should be discussed in more detail. Although the initial review scores were positive, these concerns were not resolved during the subsequent external or internal discussion phases, reflecting more borderline evaluations.